# The DNA methylation landscape of primary triple-negative breast cancer

Mattias Aine [1], Deborah F. Nacer [1,2], Elsa Arbajian [1], Srinivas Veerla [1,2], Anna Karlsson [1,2], Jari Häkkinen [1], Henrik J. Johansson [3], Frida Rosengren[1], Johan Vallon-Christersson [1], Åke Borg [1] & Johan Staaf [1,2] ✉

Triple-negative breast cancer (TNBC) is a clinically challenging and molecularly heterogenous breast cancer subgroup. Here, we investigate the DNA methylation landscape of TNBC. By analyzing tumor methylome profiles and accounting for the genomic context of CpG methylation, we divide TNBC into two epigenetic subtypes corresponding to a Basal and a non-Basal group, in which characteristic transcriptional patterns are correlated with DNA methylation of distal regulatory elements and epigenetic regulation of key steroid response genes and developmental transcription factors. Further subdivision of the Basal and non-Basal subtypes identifies subgroups transcending genetic and proposed TNBC mRNA subtypes, demonstrating widely differing immunological microenvironments, putative epigenetically-mediated immune evasion strategies, and a specific metabolic gene network in older patients that may be epigenetically regulated. Our study attempts to target the epigenetic backbone of TNBC, an approach that may inform future studies regarding tumor origins and the role of the microenvironment in shaping the cancer epigenome.

Triple-negative breast cancer (TNBC) is negative for the three current treatment-predictive markers (estrogen receptor, ER, progesterone receptor, PR, and *ERBB2/HER2* gene amplification) and constitutes ~10–20% of patients with primary disease[1,2]. While TNBC is a clinical subgroup of breast cancer, TNBC tumors display great molecular heterogeneity with respect to somatic alterations, the general tumor microenvironment (TME), and the specific tumor immune microenvironment (TIME)[3,4]. Based on genetics, gene expression of bulk tumor tissue, or proteomic analyses, attempts have been made to derive a TNBC molecular taxonomy, clinically relevant molecular subgroups, and genetic alterations associated with DNA repair deficiency[5–11]. A key distinction in transcriptional studies of TNBC has been a subgroup of tumors with a luminal-like expression pattern[5,8,10]. Lehmann et al. termed this group of tumors luminal androgen receptor (LAR) as they displayed increased expression of genes involved in steroid synthesis, androgen receptor (AR)/estrogen, and porphyrin

metabolism[8]. LAR tumors have been further shown to display specific metabolic differences connected to sphingolipid metabolism and ceramides[12]. In addition to LAR, other TNBC mRNA subtypes have been proposed in different studies related to, e.g., immune and mesenchymal features, and investigated for correlation with other genomic and TME/TIME alterations[13,14]. On a general level, bulk tissue mRNA studies represented the key first steps toward a detailed molecular phenotyping of TNBC tumors and their TME/TIME characteristics, and bulk tissue profiling still probably represents the most easily standardizable and clinically amenable approach to tumor profiling.

DNA-based TNBC studies have focused on the strong association between TNBC and DNA repair deficiency, typically manifested as homologous recombination deficiency (HRD) caused predominantly by mutational or epigenetic alterations in *BRCA1*[6,7,15,16], shaping tumor genomes over time through a mutational process that confers genomic instability. Importantly, these studies have shown that while HRD

[1]Division of Oncology, Department of Clinical Sciences Lund, Lund University, Medicon Village, SE 22381 Lund, Sweden. [2]Division of Translational Cancer Research, Department of Laboratory Medicine, Lund University, Medicon Village, SE 22381 Lund, Sweden. [3]Department of Oncology-Pathology, Science for Life Laboratory, Karolinska Institutet, Solna, Sweden. ✉e-mail: johan.staaf@med.lu.se

typically arises in a basal-like mRNA phenotype, the basal-like phenotype also includes HRD-negative cases, and HRD-positive tumors are found in all proposed TNBC-specific subtypes[6]. In contrast to the mRNA and DNA levels, less attention has been given to the global epigenetic landscape of TNBC. Epigenetic regulation is a powerful mechanism whereby cells control local and global gene expression patterns using both promoter hypermethylation and non-promoter (distal) regulatory regions often marked by differentially open chromatin and enhancer-linked histone modifications. Histone modifications, chromatin remodeling, and DNA methylation represent fundamental processes underlying epigenetic regulation and are commonly altered during tumorigenesis (see ref. 17 for review). DNA methylation involves the addition of a methyl group to a cytosine base in the context of a CpG dinucleotide. DNA methylation has a vital role in transcriptional regulation, gene imprinting, X-chromosome inactivation, and suppression of repeat elements. Three studies have suggested the existence of either three[18,19] or four[20] DNA methylation subtypes in TNBC based on analysis of Illumina DNA methylation data from bulk tumor tissue. Briefly, Stirzaker et al. reported three TNBC subgroups after analyzing 73 tumors from The Cancer Genome Atlas (TCGA) project, and only differing prognoses were reported as group-specific characteristics[18]. Lin et al. also reported three subgroups based on 44 analyzed tumors: one subgroup was characterized by older patients with typically low-proliferative tumors with low immune infiltration, and an overrepresentation of *PIK3CA* mutations; another subgroup comprised younger patients with tumors typically of higher grade, with higher levels of immune infiltration, and an over-representation of *BRCA1/2* mutations; and the third subgroup lacked defining characteristics[19]. Lastly, DiNome et al. reported four subgroups based on 79 analyzed TCGA tumors, two of which showed enrichment of two TNBC mRNA subtypes (mesenchymal and LAR), and another subgroup showed slightly more immune-warm tumors[20]. However, none of these studies provided validation of the proposed DNA methylation subtypes in independent cohorts and there has been no systematic comparison of reported subgroups.

Notably, most subtype studies in TNBC (mRNA and DNA methylation-based[18-20]) have been conducted without accounting or adjusting for the mixture of patterns originating from malignant and non-malignant cells in bulk tumor tissue. Importantly, tumor heterogeneity and tumor purity have been highlighted as sources of bias in unsupervised analyses of both mRNA and DNA methylation studies[21,22], and cellular composition may explain much of the observed variability if left unadjusted. Thus, it remains unclear whether proposed TNBC molecular subtypes are tumor intrinsic or reflections of the composition of the bulk tumor tissue analyzed. In addition, expected DNA methylation patterns and tumor-specific shifts are context-dependent in relation to genes and local CpG density. Principally, gene promoters reside in either a high (HCP) or a low CpG density (LCP) context with respect to the local observed vs expected (O/E) CpG content[23], but the existence of an intermediate CpG density (ICP) promoter class enriched for dynamic methylation changes has also been proposed[24]. HCP promoters usually colocalize with CpG islands (CGI), which are short genomic regions of above expected CpG content in the genome[25]. In addition, CGI shores (small regions adjacent to CGIs) have been established as being enriched for tissue- and cancer-specific differences in methylation[26]. Extending the analogy of the island-shore paradigm, regions distal to these features have been called oceans and are mainly characterized by a low O/E CpG metric and a methylated ground state. Given the link between gene promoters and CGIs, shores in turn largely overlap with regions proximal to promoters, and oceans with intra- or intergenic regions, representing different ways to conceptualize CpGs in the genome considered here as contexts. Moreover, CpGs can reside in regions with different chromatin accessibility, normally referred to as open (accessible) or closed (inaccessible) chromatin as measured for instance by the assay for transposase-

accessible chromatin using sequencing (ATAC-seq). A hallmark of active DNA regulatory elements is open chromatin with specific binding of transcription factors (TFs) and specific histone modifications including, e.g., histone H3 lysine 27 acetylation. DNA methylation at distal enhancer regions has been implicated in gene regulation by, e.g., interfering with TF binding to enhancer regions[27-30]. Together, this provides an additional layer of complexity when analyzing DNA methylation typically not accounted for in previous studies[18-20].

In this work, we delineate the DNA methylation landscape of TNBC and link epigenetic patterns with clinicopathological and other high-dimensional omics data seeking to resolve the question of intrinsic subtypes and potential epigenetic regulation of key transcriptional drivers in TNBC. Two key methodological features in this undertaking are the usage of DNA methylation data adjusted per CpG for tumor purity and the consideration of the genomic context of CpGs. Through an unsupervised analysis of purity-adjusted DNA methylation data we identify two main epitypes (epigenetic subtypes), a basal and a non-basal. These are further subdivided into three basal and two non-basal subgroups associated with distinct characteristics regarding clinicopathological variables, transcriptional patterns, genetic alterations, and TIME/TME patterns. Taken together, findings in this study provide a detailed molecular picture of TNBC and highlights the existence of distinct and large-scale epigenetic alterations underlying both broad expression phenotypes and potentially specific immune-evasion strategies.

## Results

The outline of the study is shown in Fig. 1. Clinicopathological characteristics for the Sweden Cancerome Analysis Network - Breast (SCAN-B) discovery and validation cohorts are provided in Table 1, with detailed patient characteristics available in Supplementary Data 1.

### The genomic context of CpG methylation in TNBC

CpG methylation measured in bulk tumor tissue is characterized by two main features, tissue heterogeneity and genomic context. To address bulk tissue heterogeneity, here represented by varying amounts of tumor cells in each analyzed sample, we employed a method that seeks to correct observed beta values on CpG-specific levels using tumor cell content estimated from whole-genome sequencing (WGS) data[31], generating what we refer to as purity-adjusted beta values. Such adjusted values have been shown to improve the epigenetic separation between the well-established PAM50 Basal and Luminal gene expression subtypes of breast cancer, as well as to enhance DNA methylation levels of individual genes like *BRCA1* in samples[31,32]. As shown in Supplementary Fig. S1, adjusted tumor data show less intermediate beta values, as would be theoretically expected given a binary methylated/not methylated state of individual cells. To address genomic context, we capitalized on the Illumina EPIC array's coverage of CpG methylation across both genic and non-genic regions of the human genome, including a substantial amount of CpGs overlapping with reported ATAC-peaks (i.e., peaks of ATAC-seq signal indicating regions of open chromatin) in breast cancer (Fig. 2a). Based on the premise that variability across samples indicates presence of biological information, we analyzed the variance in DNA methylation across samples using both a gene-centric (promoter, proximal, distal) and a CpG-centric (CGI, shore, ocean) context, further stratified by ATAC-peak overlap, to identify contexts that hold most information (Fig. 2b). With respect to gene-centric features, promoter and proximal CpGs (which are typically associated with a higher O/E CpG metric) exhibited lower aggregate DNA methylation beta variance when overlapping ATAC-peaks, while distal CpGs (with low baseline O/E metric) showed increased variance, especially in ATAC-peaks. Regarding the CpG-centric paradigm, we found that CGIs and shores exhibit the same characteristics as promoters and proximal regions (i.e., lower aggregate variance when overlapping ATAC-peaks),

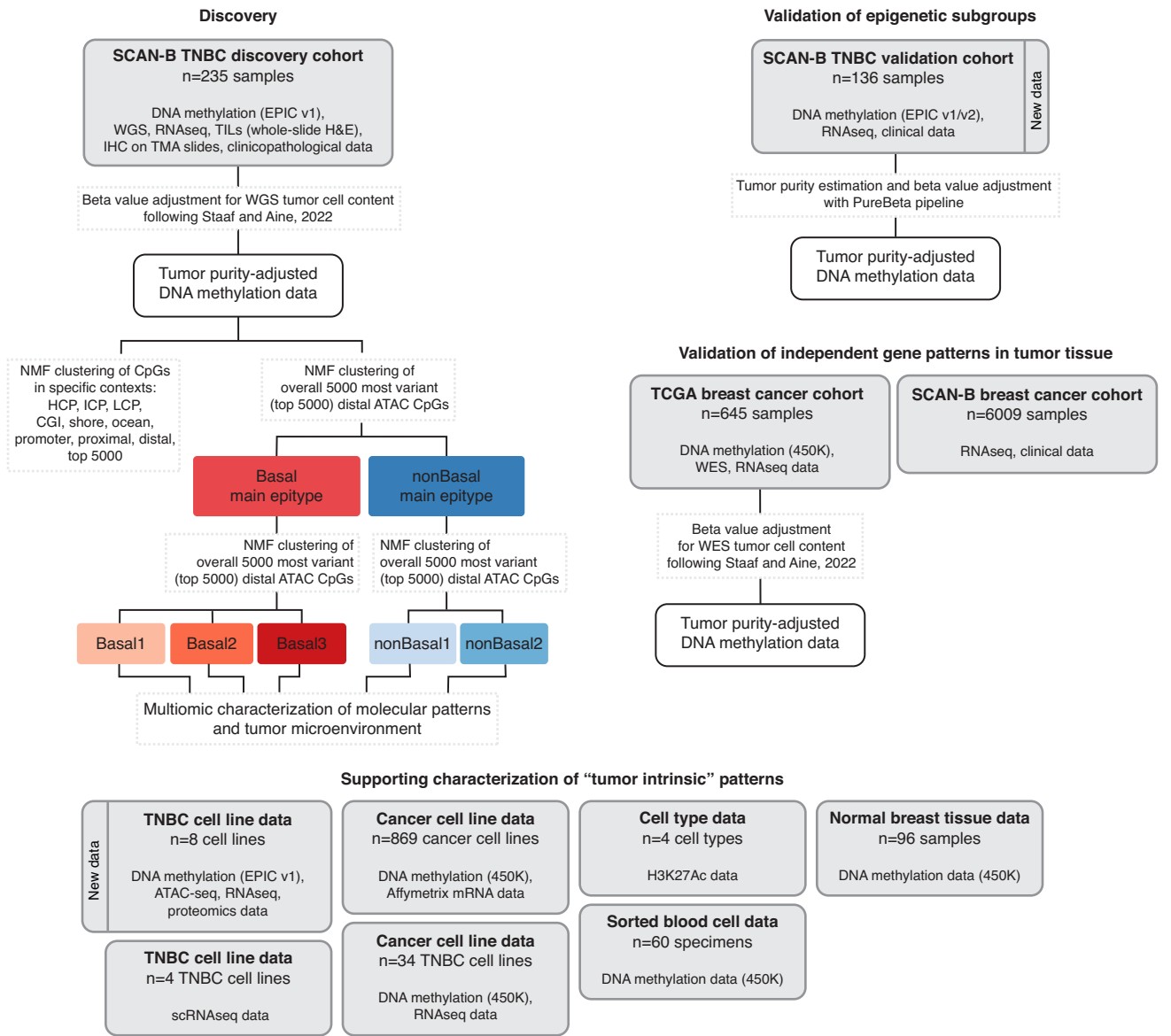

**Fig. 1 | Overview of study with included cohorts and analysis steps.** WGS: whole genome sequencing. TILs: tumor infiltrating lymphocytes. WES: whole exome sequencing. scRNAseq: single cell RNA-sequencing. NMF: non-negative matrix factorization. ATAC-seq: assay for transposase-accessible chromatin using sequencing.

while CpGs residing in a low-density context (i.e., oceans or LCP/ICP regions), exhibited increased variability also in ATAC-peaks.

Another important hallmark of putative regulatory function is the presence or enrichment of transcription factor binding sites (TFBSs) and histone modifications associated with gene regulation. Using ENCODE TFBS data for 340 TFs derived from 130 cell lines, we found that CpGs located within ATAC-peaks were associated with a higher number of TFs (Fig. 2c). Enrichment analysis and clustering of the ENCODE TFBSs in relation to both the gene- and CpG-centric genomic contexts stratified by ATAC-peak overlaps highlighted three broad enrichment patterns largely corresponding to a promoter/CGI, proximal/shore, and distal/ocean context (Supplementary Fig. S2). While CpGs in promoters and CGIs showed universal enrichment of TFBSs irrespective of ATAC-peak overlaps, consistent with their localization in constitutive euchromatin, distal/ocean sites displayed a pattern consistent with facultative euchromatin wherein a smaller subset of TFBSs, including members of the known pioneering factor families FOX and GATA, were specifically enriched in CpGs with ATAC-peak overlaps (Supplementary Fig. S2). Together, the DNA methylation

variance and TFBS overlap analyses indicate that CpGs in a distal/ocean context with overlapping ATAC-peaks capture a subset of the gene-regulatory landscape in TNBC with higher biological information content (variance) and specificity (TF binding), representing a promising DNA methylation context within this breast cancer subtype.

## Two main DNA methylation phenotypes in TNBC
After exploring characteristics of different CpG genomic contexts, we proceeded to analyze their potential impact on TNBC subgroup identification. To evaluate the effects of number of CpGs used and of CpG genomic context on sample clustering, we performed non-negative matrix factorization (NMF) clustering of nine different CpG sets representing both CpG density- and gene-centric contexts, as well as one CpG set representing the commonly used approach of selecting top varying CpGs by standard deviation (SD) genome-wide ($n = 10$ CpG sets in total). We evaluated two-class clustering similarity between all ten contexts using the 1000, 2500, or 5000 most variant CpGs within each context (allowing for investigation of the impact of different feature numbers on the clustering), as well as stratifying by overlaps

with, without, or agnostic to ATAC-peaks (i.e., chromatin accessibility). Given that different CpG sets and varying number of CpGs used for clustering did not result in materially different classifications

## Table 1 | Clinicopathological and molecular characteristics of the SCAN-B discovery and validation cohorts

| | Discovery cohort[a] | Validation cohort[a] |
|---|---|---|
| Number of patients | | |
| | 235 | 136 |
| Patients assessable for DRFI after adjuvant therapy | | |
| | 144 | Not applicable |
| Age (median and range, in years) | | |
| | 62 (26–91) | 62.5 (30–95) |
| Lymph node status | | |
| Node-negative (N0) | 152 (65%) | 87 (69%) |
| Node-positive (N+) | 83 (35%) | 39 (31%) |
| Not available | 0 | 10 |
| Nottingham Histologic Grade | | |
| G1 | 0 (0%) | 0 (0%) |
| G2 | 26 (11%) | 24 (23%) |
| G3 | 206 (89%) | 81 (77%) |
| Not available | 3 | 31 |
| PAM50 subtype | | |
| Basal-like | 187 (80%) | 95 (70%) |
| non-Basal-like | 48 (20%) | 41 (30%) |

[a]Percentages computed excluding missing data. *DRFI*: distant relapse-free interval.

(Supplementary Fig. S3a), we proceeded with sets of 5000 CpGs as these provided the largest number of informative data points.

To further evaluate clustering performance across contexts, we used concordance between NMF two-group splits and gene expression-derived PAM50 Basal vs non-Basal (nonBasal) classification as the latter constitutes a dominant biological signal even in the context of TNBC. Overall, we observed that methylation clusters derived using NMF corresponded well to PAM50 Basal/nonBasal gene expression classifications across analyzed CpG sets (Supplementary Fig. S3b). When 5000 CpGs were used, the accuracy for separating PAM50 nonBasal and Basal TNBC was similar across all context sets (mean = 0.84, range = 0.70–0.92) and CpGs with ATAC-peak overlaps showed highest concordance with the gene expression classification (mean = 0.896 vs 0.816 for the rest, two-sided *t*-test, *p* < 0.001). Importantly, purity-adjusted CpG data showed consistently more stable and similar accuracies for this split across tested CpG contexts compared with conventional unadjusted CpG data (Supplementary Fig. S3c). Together, these findings show that the PAM50 Basal and nonBasal gene expression phenotypes largely correspond to the dominant epigenetic subtypes in TNBC, that this split is well reflected in most genomic contexts, and that the concordance is highest in CpGs overlapping ATAC-seq peaks. These observations, combined with the ones made in the previous section regarding biological information potential, led to the hypothesis that DNA methylation profiling of distal CpGs purportedly involved in regulatory activity due to their overlap with ATAC regions of open chromatin (referred to as distal-ATAC from here on) would constitute a promising background for investigating TNBC DNA methylation phenotypes.

To test this hypothesis, we performed NMF clustering of the 5000 most varying distal-ATAC CpGs with number of groups to identify (k)

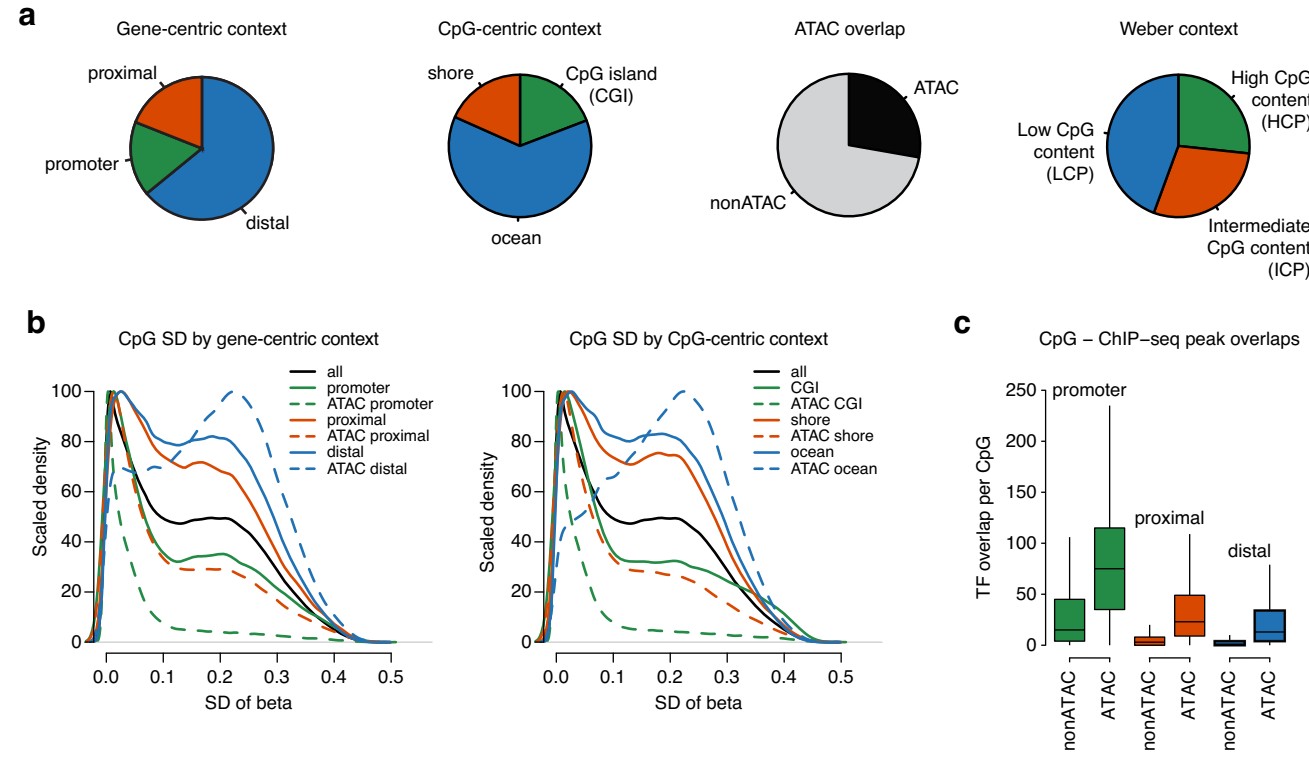

**Fig. 2 | CpG contexts and DNA methylation variance in TNBC. a** Proportions of custom Illumina EPIC CpG annotations related to different CpG contexts explored in the study. **b** CpGs in distal/low CpG-density regions overlapping ATAC-peaks show increased variability in DNA methylation (beta standard deviation, SD), whereas this is lower in CpGs in promoters and proximal regions with ATAC-peak overlaps. **c** The number of bound transcription factors at promoters is relatively higher than that of proximal and distal regions and is less affected by the presence of overlapping ATAC-peaks based on ENCODE TFBS data for 340 transcription factors (TFs) derived from 130 cell lines. Boxplot elements correspond to: (i) center line = median, (ii) box limits = upper and lower quartiles, (iii) whiskers = 1.5x interquartile range. Outliers are not displayed. Source data are provided as a Source Data file.

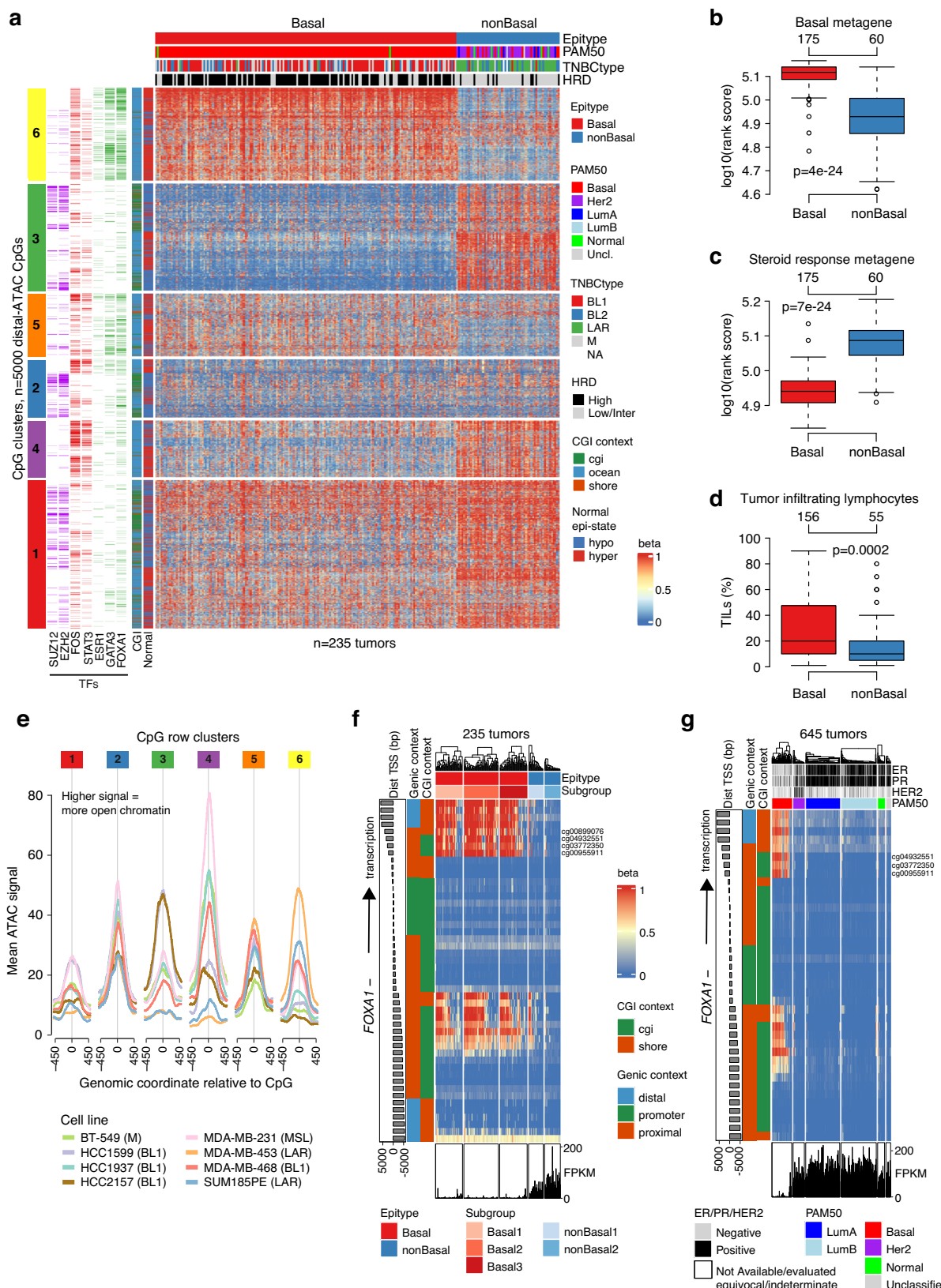

ranging from 2 to 5. Irrespective of k, clustering captured two main patterns that corresponded to samples classified as PAM50 Basal or nonBasal, and increasing k did not affect clustering of these two main subgroups but rather resulted in further splitting of the larger Basal cluster (Supplementary Fig. S4a, b). We therefore opted for a k = 2 NMF solution with two clusters comprising 175 and 60 tumors, respectively,

as the basis for further analysis (Fig. 3a). With respect to molecular and clinical features, the larger cluster broadly corresponded to PAM50 Basal tumors (referred to as the Basal epigenetic subtype, or epitype, hereon) while the smaller cluster was dominated by PAM50 nonBasal tumors (referred to as the nonBasal epitype hereon) (Supplementary Fig. S4c). Regarding TNBCtype mRNA subtypes, the nonBasal epitype

**Fig. 3 | Two main epigenetic subtypes of TNBC correspond to Basal and non-Basal tumor phenotypes. a** NMF clustering of DNA methylation values for 235 TNBCs using the 5000 most variant CpGs located in distal ATAC-peaks and a two-group solution, defining a Basal and nonBasal main epitypes. Sample (columns) annotation tracks show PAM50 subtypes, TNBCtype, and homologous recombination deficiency (HRD) status. CpG (rows) annotation tracks show different overlaps with transcription factor (TF) binding sites (color = overlap, white = no overlap), CGI context, and beta values seen in normal breast samples. Row clustering performed using Euclidean distance and ward.D linkage. **b, c** Rank scores for (**b**) a basal gene expression metagene and (**c**) a steroid response metagene (including *FOXA1* and *GATA3*) versus NMF clusters. **d** Pathology-estimated TIL scores versus NMF clusters. **e** Mean ATAC signal at distal-ATAC CpGs in eight TNBC cell lines across the six row clusters highlighted in (**a**). ATAC signal peaks indicate

open chromatin in that genomic region. Association of cell lines with TNBCtype mRNA subtypes are indicated. **f** Methylation heatmap of CpGs in the *FOXA1* promoter region (rows) in 235 SCAN-B TNBC tumors (columns). Bottom panel shows corresponding *FOXA1* mRNA expression (fragments per kilobase per million reads, FPKM). Arrow indicates direction of transcription for the gene (minus strand). Dist TSS (bp) = distance between CpG and transcription start site of gene in base pairs. The Subgroup track corresponds to subdivisions of the Basal and nonBasal epitypes described further below. **g** Same plot as in (**f**), but for purity-adjusted beta values of 645 TCGA tumors of all clinical subgroups and PAM50 subtypes. In panels b-d, two-sided *p*-values were calculated using Wilcoxon's test. Boxplot elements correspond to: (i) center line = median, (ii) box limits = upper and lower quartiles, (iii) whiskers = 1.5x interquartile range. Top-axis in boxplots reports group sizes. Source data are provided as a Source Data file.

contained 90% of the LAR tumors in the cohort (36/40), while the Basal epitype was a mix of the remaining TNBCtypes (Supplementary Fig. S4c). HRD-positive tumors were mainly found in the Basal cluster (Fig. 3a, Supplementary Fig. S4c).

The two epitypes also showed distinct expression of basal-related and steroid response-associated genes (including *GATA3* and *FOXA1*) as assessed by reported biological metagenes (i.e., groups of highly connected, co-expressed genes in breast cancer that have been associated to specific biological traits)[33], as well as a general difference of the TIME assessed by tumor infiltrating lymphocytes (TILs) (Fig. 3b–d). NonBasal tumors also showed differences in *TP53* and PI3K pathway mutation rates, higher patient age at diagnosis, and poorer patient prognosis after chemotherapy using distant relapse-free interval (DRFI) as endpoint when compared to Basal tumors/patients (Supplementary Fig. S4d–f).

Hierarchical cluster analysis identified six distinct methylation patterns (CpG clusters) within the 5000 CpGs used to define the two NMF clusters (Fig. 3a). Enrichments of ENCODE TF overlaps were calculated for each CpG cluster (Supplementary Data 2a), revealing that CpG clusters 1 and 3 (associated with hypermethylation in nonBasal tumors) displayed more TFs usually enriched at promoters and other high CpG-density regions like RNA-polymerase II subunits and members of the polycomb silencing complex. While CpG cluster 2 showed a similar TF enrichment to CpG cluster 3, the methylation profile of the former was less distinct between Basal and nonBasal tumors. CpG cluster 4, also hypermethylated in nonBasal samples, was enriched for *FOS*, *STAT3*, and TFs in the JUN family. CpG clusters 5 and 6, both hypomethylated in nonBasal TNBC, were enriched for overlaps with *GATA3* and TFs in the forkhead family (*FOXA1*/*FOXA2*). Of these two, cluster 6 displayed stronger overall enrichment *p*-values and additionally robust enrichment of estrogen receptor binding. Aggregated ATAC-seq peak data from eight TNBC cell lines mapped to CpGs in the respective clusters showed that CpG positions in clusters 5 and 6 (hypomethylated in nonBasal) have higher peak signal (representing open chromatin) in LAR cell lines (Fig. 3e). We also mapped the 5000 distal-ATAC CpGs to acetylation of the H3K27 histone residue (H3K27Ac), a marker of active enhancer regions, in breast luminal progenitor cells and three common immune cell types using publicly available data. This analysis showed a pattern of specifically elevated local H3K27Ac in breast luminal progenitor cells (in particular for CpG clusters 3-4) and lower, unspecific signal in immune cells (Supplementary Fig. S4g). Together, the extended analyses in TNBC cell lines and different non-malignant cell types argue in favor of this set of distal-ATAC CpGs being involved in regulation of gene expression in tumor and/or epithelial cells and not in infiltrating immune cells.

## Promoter region methylation of steroid response genes

Differential methylation analysis identified 28565 CpGs with significantly differing methylation levels between Basal and nonBasal tumors based on all ~740 k profiled CpGs (Bonferroni-adjusted, two-sided Wilcoxon test, *p* < 0.01, absolute beta difference between groups

>0.25) (Supplementary Data 2b). Among these, 17752 CpGs showed higher methylation in nonBasal samples and 10813 displayed lower methylation. Of the 28565 CpGs, 6.3% were among the 5000 most variant distal-ATAC CpGs in Fig. 2a, mainly in CpG clusters 3, 4, and 6 (Supplementary Fig. S4h). Conversely, 70% of CpGs in cluster 3, 56% of CpGs in cluster 6, and 42% of CpGs in cluster 4 were among the 28565 significant CpGs (Supplementary Fig. S4h). A difference in CpG density distribution was evident between the two sets, wherein the majority of CpGs with higher methylation in Basal samples were in low density/ocean regions, while CpGs with higher methylation in nonBasal samples were more frequently in high-density CpG island regions (Supplementary Fig. S4i). To identify genes correlated with CpG methylation patterns, we mapped all 28565 CpGs to the expression of nearby genes in a 500 kbp genomic window and calculated correlation coefficients between CpG methylation and FPKM (fragments per kilobase per million reads) gene expression considering all tumors. This analysis identified 21082 significant CpG-gene pairs mapping to 12517 unique CpGs (Supplementary Data 2c), 664 of which were present among the most-varying distal-ATAC CpGs in Fig. 3a, mainly in CpG clusters 3 and 6 (43.5% and 23.5% of CpGs, respectively).

Consistent with the relevance of CpG cluster 6, enriched for overlaps with the *GATA3* and *FOXA1*/*FOXA2* transcription factors, a key discriminator between the Basal and nonBasal tumors is the elevated mRNA expression of genes included in a steroid response metagene, like *FOXA1* and *GATA3* (Fig. 3c), but not *ESR1* or *PGR* (Supplementary Fig. S4j). To investigate potential epigenetic regulation of steroid response genes between Basal and nonBasal tumors, we first focused on CpGs near the transcription start sites (TSSs ± 6 kbp) of the nine genes present in the cited metagene (*ABAT*, *AGR2*, *CA12*, *DNALI1*, *FOXA1*, *GATA3*, *SLC44A4*, *TBC1D9*, and *XBP1*) with a significant negative correlation between DNA methylation and gene expression. These constraints identified twelve CpGs for *FOXA1*, nine for *SLC44A4*, and one CpG each for *XBP1* and *CA12*. We created detailed promoter region plots for all nine genes, finding that *FOXA1* and *SLC44A4* show potential differential methylation between Basal and nonBasal tumors (Supplementary Fig. S4k).

More specifically, we observed an evident pattern of CpG shore/CGI methylation for *FOXA1* that aligned almost perfectly with the Basal (hypermethylated shore/CGI CpGs) and nonBasal (hypomethylated shore/CGI CpGs) epigenetic stratification of the 235 SCAN-B tumors and their mRNA expression of *FOXA1* (Fig. 3f, with labeled CpGs shown in detail in Supplementary Fig. S4l). Importantly, the same set of CpGs are hypomethylated in normal breast tissue specimens, i.e., a similar methylation state to nonBasal tumors (Supplementary Fig. S4m). A corresponding analysis in the eight TNBC cell lines showed that LAR cell lines appeared to be consistently fully hypomethylated, whereas proposed basal-like 1 (BL1) cell lines were typically hypermethylated in the same region (Supplementary Fig. S4n). Supporting a gene-regulatory association, ATAC-seq signals in the *FOXA1* promoter region indicate clear differences in chromatin accessibility between LAR and non-LAR TNBC cell lines in addition to the DNA methylation

pattern (Supplementary Fig. S4o). Finally, shore/CGI hypermethylation and downregulated mRNA expression of the *FOXA1* gene seem to be unique to Basal-like tumors irrespective of ER, PR and HER2 status through analysis of 645 TCGA breast cancers of all PAM50 subtypes with matched DNA methylation and RNA-sequencing (RNAseq) data, consistent with the pattern observed in SCAN-B TNBC tumors (Fig. 3g). A similar analysis performed on non-CGI CpGs in the TSS region of *SLC44A4* resulted in an opposite finding to that of *FOXA1* (albeit more heterogenous), with selective hypomethylation in nonBasal-like tumors irrespective of ER, PR, and HER2 status, compared to hypermethylation in Basal-like tumors and in normal breast tissue (Supplementary Fig. S4p, r). Together, this analysis identified potential epigenetic regulation of two out of nine genes in the steroid response metagene.

## Selective epigenetic regulation in Basal/nonBasal tumors

To expand the steroid response gene analysis to more genes, we first performed a differential gene expression analysis between Basal and nonBasal tumors identifying 5943 significant genes. Next, we filtered the 21082 significantly correlated CpG-gene pairs obtained from the 28565 differentially methylated CpGs between Basal and nonBasal tumors by requiring: i) a computed negative CpG-expression correlation, ii) a CpG distance to TSS < 7 kbp (as resolution of the EPIC platform varies per gene) for at least three CpGs per gene, iii) a minimum absolute median FPKM difference >5 between Basal and nonBasal tumors, and iv) presence of the gene in the list of 5943 differentially expressed genes. These filters identified 137 candidate genes for potential epigenetic regulation differing between epitypes and included transcription factors like *FOXC1* (present in the PAM50 gene signature[34]), *TFAP2B*, and *ELF5* (involved in mammary gland development), as well as the already mentioned *FOXA1* and *SLC44A4* (Supplementary Data 2d). Pathway analysis using hallmark signatures from MSigDB[35] identified only the estrogen late response hallmark as significant (adjusted two-sided $p = 0.03$). To further refine the gene list, we required a minimum absolute fold change difference of >10 in FPKM between Basal and nonBasal tumors, leaving 27 genes. For each gene, we created promoter methylation and expression maps to examine support for distinct epigenetic regulation between Basal and nonBasal tumors in the SCAN-B discovery cohort and between Basal and nonBasal tumors in the 645-sample TCGA breast cancer cohort. Three genes (*FOXC1*, *EN1*, and *ELF5*) stood out by displaying a pattern opposite to that of *FOXA1*, with hypomethylation and expression in Basal tumors, but hypermethylation and repressed expression in all other breast cancer subtypes (Supplementary Fig. S4s).

## Three epigenetic subgroups within the Basal epitype

To investigate whether the Basal epitype could be further subdivided, we performed NMF clustering of the 175 Basal tumors using two- and three-group solutions and purity-adjusted distal-ATAC CpG beta values ($n = 5000$ most variant). Sample cluster agreement between solutions was high (two-sided Chi-square test, $p < 2e{-}16$), and there was a 57% overlap of differentially methylated CpGs identified between groups using the different solutions ($n = 1087$ CpGs for two-group, $n = 2845$ CpGs for three-group), indicating that the same biology was captured (Supplementary Fig. S5a). Based on this, we proceeded to characterize the three-group NMF solution. Figure 4a shows the 2845 differentially methylated CpGs clustered into six groups for the three basal subgroups, Basal1 ($n = 53$ samples), Basal2 ($n = 68$), and Basal3 ($n = 54$). CpG clusters 3 and 5 did not show strong enrichment of any tested TF, while CpG cluster 1 (associated with hypermethylation in Basal2 tumors) displayed weaker enrichment of TFs usually associated with promoters and other high CpG-density regions like RNA-polymerase II subunits and members of the polycomb silencing complex. CpG cluster 2 resembled CpG cluster 1 in TF enrichment but appeared more associated with hypermethylation in both Basal1 and

Basal2 tumors compared to CpG cluster 1. Finally, CpG clusters 4 and 6 were enriched for *FOS* and *STAT3* but showed different methylation patterns across Basal subgroups, mainly for Basal3 tumors (hypomethylation of cluster 6 while hypermethylation of cluster 4). TNBCtype subtypes and HRD-positive tumors appeared in all subgroups. Consistently, there was no subgroup difference for rearrangement signatures characteristic of *BRCA1*- or *BRCA2*[7] nor for tumor mutational burden (two-sided Kruskal-Wallis test, $p > 0.05$). Moreover, based on mutational driver analysis of single nucleotide variants and indels in 102 genes from a previous WGS study performed on the same tumors[6], we found no significant enrichment of alterations for any gene in the subgroups (two-sided Chi-square test, $p > 0.05$). While *MYC* and *AKT3* amplifications were more frequent in Basal1 tumors (two-sided Chi-square test, $p = 0.05$ and $p = 0.02$, respectively), significance was not retained after multiple testing adjustment (two-sided False Discovery Rate test, FDR, $p > 0.05$).

The Basal subgroups showed different immune cell infiltration, with high values in Basal3 tumors, comparatively low values in Basal2 tumors, and intermediate values in Basal1 tumors as shown by TIL estimates and scores of an immune response metagene[33] (Fig. 4b). Kaplan-Meier analysis demonstrated a nonsignificant trend towards better prognosis (DRFI) for the Basal3 subgroup (Fig. 4c). Based on the other proposed metagenes, Basal2 tumors showed highest scores of both the basal (Kruskal-Wallis test, $p = 2e{-}7$) and stroma metagenes (two-sided Kruskal-Wallis test, $p = 6e{-}7$) (Supplementary Fig. S5b). In contrast, Basal1 tumors were not strongly associated with any metagene and may constitute a mixed phenotype. The three subgroups showed a considerable number of differentially expressed genes ($n = 5005$) between them (Supplementary Fig. S5c). The Basal3 subgroup displayed the highest number of differentially expressed genes ($n = 2099$), followed by Basal2 ($n = 1015$) and Basal1 ($n = 578$) (Fig. 4d, Supplementary Data 3a). Gene set analysis using hallmark signatures of differential genes in Basal3 tumors confirmed a strong association with multiple immune pathways (Supplementary Data 3b). Basal2 differential genes were related to, among others, epithelial-mesenchymal transition (EMT), hypoxia, and transforming growth-factor beta (TGFB) signaling signatures. Consistently, Basal2 tumors also showed higher stroma content compared to the other subgroups as estimated by CIBERSORTx cell deconvolution data obtained from[16] (Kruskal-Wallis test, $p = 4e{-}10$). In contrast, Basal1 differential genes were not associated with any hallmark signatures, but KEGG and Reactome analyses identified different metabolic, respiration, and oxidative processes among them. Further analysis of genes associated with Basal1 using the GAM (Genes And Metabolism) web service[36] did not, however, identify any specific metabolic pathways.

Given the slightly lower tumor cell content and prominent immune gene expression differences between the Basal subgroups, we sought to probe whether the observed epigenetic and expression phenotypes were intrinsically linked or represented a composite of an intrinsic epigenetic phenotype and an extrinsic expression phenotype confounded by infiltrating immune cells. To address this, we performed a CpG-gene correlation analysis between the 2845 CpGs differentially methylated across the three Basal subgroups (Fig. 4a) and the expression of nearby genes. In addition, we performed an analogous analysis for the same CpGs in the eight TNBC cell lines with matching DNA methylation and RNAseq data to capture CpG-gene associations that are present in both data sets. Out of 14464 pairwise CpG-gene correlations, 142 were significant in both data sets with predominantly negative correlations (Fig. 4e). Importantly, these 142 pairs show significant correlations between DNA methylation and gene expression data in both bulk tumor tissue and a pure TNBC cell line context devoid of non-malignant cells. As expected from the CpG-gene associations, the aggregated 142 CpGs showed different chromatin accessibility across the eight profiled cell lines (Fig. 4f). We also

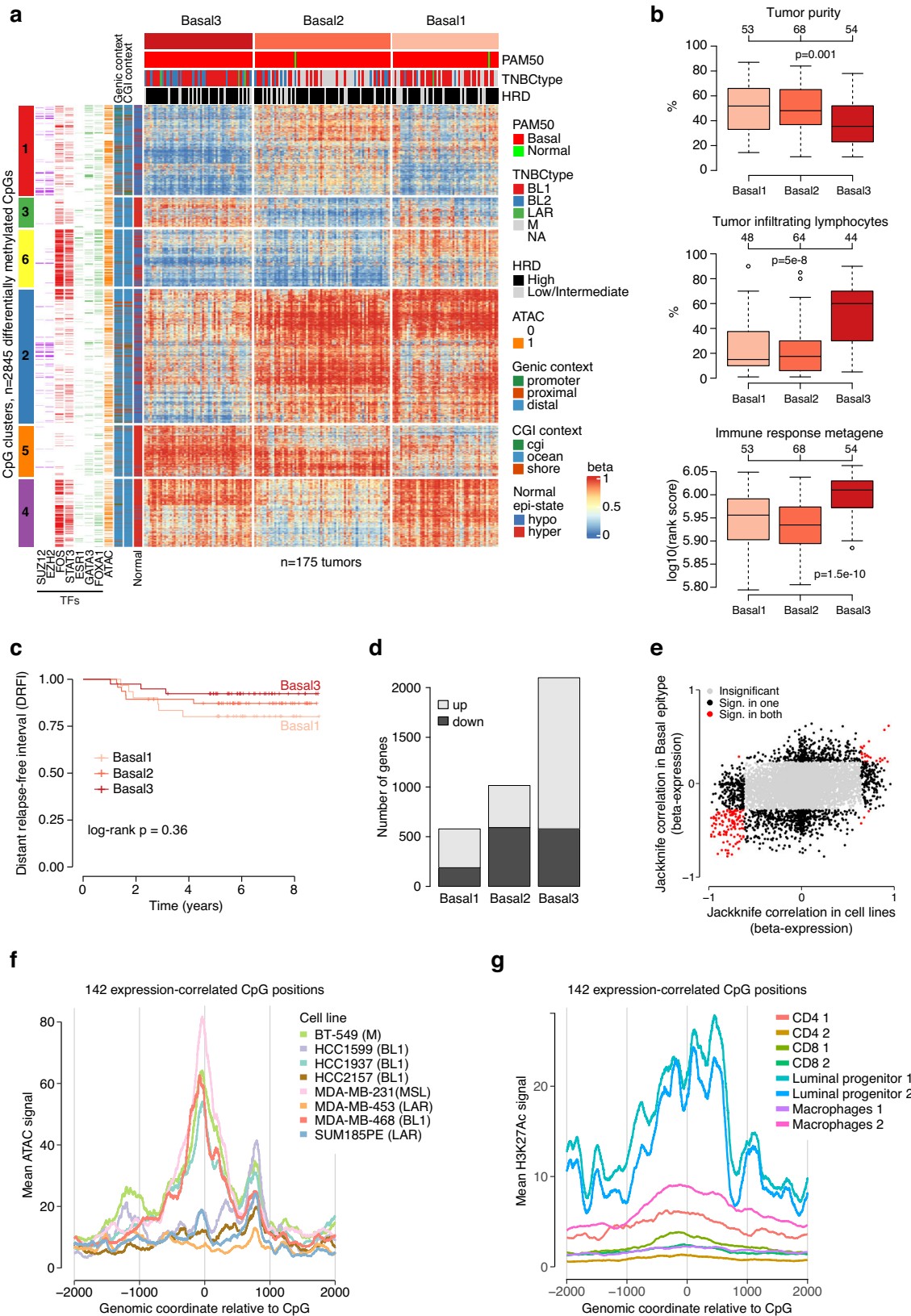

investigated H3K27Ac for this CpG set in breast luminal progenitor cells and three common immune cell types, noticing a pattern of locally elevated acetylation in the breast luminal progenitors and lower, unspecific signal in immune cells (Fig. 4g). This indicates that this set of CpGs are involved in gene expression regulation in tumor and/or epithelial cells rather than in infiltrating immune cells. Of the

142 CpG-gene pairs, 138 agreed also in correlation direction between cell lines and tumors (Supplementary Data 3c). A hallmark signature analysis of the 97 unique genes in the 138-set identified four significant terms related to interferon alpha response, interferon gamma response, TNFA signaling via NFKB, and EMT (adjusted two-sided test, $p < 0.05$).

**Fig. 4 | Epigenetic subgroups within the main Basal epitype. a** NMF clustering of 175 TNBCs from the main Basal epitype defined three subgroups: Basal1, Basal2, and Basal3. Heatmap shows beta values for 2845 differentially methylated CpGs between the three Basal subgroups. Sample annotation tracks show PAM50 subtypes, TNBCtype, and homologous recombination deficiency (HRD) status. CpGs (rows) were clustered using Euclidean distance and ward.D linkage. CpG annotation tracks also show if CpGs are located within transcription factor (TF) binding sites (color = yes, white = no), different CpG contexts, and beta values seen in normal breast samples. **b** Boxplots of tumor purity (as estimated from whole genome sequencing data), tumor infiltrating lymphocyte counts, and gene expression rank scores for the immune response metagene for Basal1-3 tumors. Two-sided *p*-values calculated using the Kruskal-Wallis test. **c** Kaplan-Meier plot using DRFI as endpoint for patients treated with adjuvant chemotherapy stratified by Basal1-3 subgroups. *P*-value calculated using the log-rank test. **d** Number of differentially expressed genes per Basal subgroup divided into whether the gene is up or down in expression compared to the other two groups. **e** Comparison of correlation between CpG-gene expression pairs in Basal epitype tumors (y axis) and in TNBC cell lines, demonstrating that most significant correlations with same direction in both data sets (red) are negative (e.g., hypermethylation and reduced gene expression). **f** Mean ATAC-signal in eight TNBC cell lines aggregated for 142 CpGs with correlated gene expression and methylation in cell line and tumor data. The x axis is centered at the CpG position showing e.g., that nonBasal cell lines (e.g., SUM185PE) do not have a signal for these positions (i.e., closed chromatin). **g** Similar plot as in (**f**), but for mean H3K27Ac signal across different cell types, showing e.g., that the signal is present in luminal progenitor cell lines but not immune cells. Boxplot elements correspond to: (i) center line = median, (ii) box limits = upper and lower quartiles, (iii) whiskers = 1.5x interquartile range. Top-axis in boxplots reports group sizes. Source data are provided as a Source Data file.

## Epigenetic patterns in immune-infiltrated Basal TNBC

Cross-referencing the 142 CpG-gene pairs to genes specifically upregulated in Basal3 tumors identified the immune-suppressive cell-surface proteins PD-L1 (*CD274* gene) and PD-L2 (*PDCD1LG2* gene), co-located in chromosome 9p24, as linked to differential methylation of the CpGs cg07211259 (*CD274* and *PDCD1LG2*) and cg19724470 (*PDCD1LG2*) in both tumors and TNBC cell lines (Fig. 5a–c). The two genes are highly co-expressed, with a Spearman correlation of 0.84 in the 175 Basal tumors, 0.86 in all 235 SCAN-B tumors, and 0.8 in a set of 34 TNBC cell lines. Notably, cg19724470 (promoter/shore context) is located 411 bp away from the TSS of *PD-L1*, while cg07211259 is approximately 60 kbp away from *PD-L1*, but only 73 bp away from the *PD-L2* TSS. Consequently, a correlation between cg19724470 and *PD-L2* and between cg07211259 and *PD-L1* stems from the two genes being within the same 500 kbp genomic window and co-expressed. In fact, cg19724470 had a significant correlation of -0.43 to *PD-L1* and of -0.36 to *PD-L2* in the discovery cohort. Limited by the number of included CpGs in the EPIC platform for both *PD-L1* and *PD-L2*, detailed promoter region maps for the two genes did not show a similar apparent pattern of shore/CGI methylation as seen for *FOXA1* (Supplementary Fig. S5d, e). A hypomethylated background state for both cg19724470 and cg07211259 is observed in normal breast tissue, as well as in sorted immune cell populations (Supplementary Fig. S5f, g), suggesting somatic hypermethylation in Basal1 and Basal2 tumors at these CpG positions.

To further investigate if hypomethylation of the two mentioned CpGs was associated with elevated *PD-L1/PD-L2* mRNA expression specifically in tumor cells, we analyzed 34 TNBC cell lines with matched RNAseq and DNA methylation data, finding that the cell lines with highest *PD-L1* or *PD-L2* mRNA expression showed hypomethylation of these CpGs (Fig. 5d), consistent with our own data. Our observations from the two TNBC cell line cohorts were further supported by similar findings in a set of 869 cancer cell lines representing several different malignancies (Supplementary Fig. S5h). While ATAC-seq data from TNBC cell lines showed peaks overlapping with and close to cg19724470 in *PD-L1*, indicating open chromatin at those positions, a more distinct pattern of ATAC-seq peaks overlapping cg07211259 in *PD-L2* was evident for the cell lines that showed elevated *PD-L1/PD-L2* mRNA expression (BT-549, HCC1937 and MDA-MB-231) (Fig. 5e). Public single cell RNAseq (scRNAseq) data for BT-549, MDA-MB-231, HCC1937, and SUM185PE confirmed bulk RNAseq patterns regarding the relational *PD-L1/PD-L2* expression patterns seen in cell lines, but also demonstrated considerable heterogeneity in expression of both genes among cancer cells, with proportionally only few cancer cells showing detectable expression even in the cell line with highest expression (MDA-MB-231) (Supplementary Fig. S5i, j).

Finally, to explore if these correlation-based observations made from bulk tumor tissue and in vitro cultured cell lines corresponded to differences in actual tumor *PD-L1* expression in situ, we performed immunohistochemistry (IHC) using the 22C3 PD-L1 antibody on matched tissue microarrays (TMAs). This allowed us to classify each tumor (*n* = 153 of 175 Basal tumors evaluable) according to the PD-L1 tumor positive score (TPS), which represents the estimated proportion of tumor cells expressing PD-L1 on their cell surface. Stratification of TPS scores by the Basal1-3 subgroups demonstrated markedly elevated scores in Basal3 tumors (Fig. 5f). Together, these results indicate a putative link between epigenetic states and PD-L1-mediated immune escape in immune-warm Basal3 tumors.

## Two epigenetic subgroups within the nonBasal epitype

To characterize epigenetic subgroups in nonBasal tumors we performed an analogous distal-ATAC (*n* = 5000 most varying CpGs) NMF analysis, restricted to a two-group solution due to lower number of samples (*n* = 60). Clustering identified two subgroups of 29 (termed nonBasal1) and 31 (nonBasal2) tumors, and differential analysis identified 978 CpGs differentially methylated between the nonBasal subgroups (Fig. 6a, Supplementary Data 4a). Of these 978 CpGs, 935 showed higher methylation and 43 lower methylation in nonBasal2 tumors compared to the nonBasal1 subgroup. Moreover, CpGs with higher beta values in nonBasal2 tumors were strongly enriched for higher CpG density (Fig. 6b).

A notable feature of the nonBasal2 subgroup was a very high enrichment of TNBCtype LAR tumors (28 of 30 subtyped tumors in the cluster, representing 78% of all LAR tumors in the nonBasal epitype). In contrast, nonBasal1 comprised a mix of different TNBCtype and PAM50 subtypes. Both subgroups presented an immune-cold TIME (Fig. 6c) and there were no differences between them in patient outcome (DRFI log-rank test, *p* > 0.05, both overall for all patients and for chemotherapy-treated patients only), HRD status, WGS-estimated tumor purity, nor tumor proliferation as assessed by metagenes (Supplementary Fig. S6a–c). The nonBasal subgroups showed additional clinicopathological, genetic, and transcriptional differences including: i) slightly higher age at diagnosis in nonBasal2 patients (Fig. 6d), ii) borderline more genetic alterations in the PI3K pathway genes in nonBasal2 tumors (Fig. 6a, two-sided Fisher's test, *p* = 0.07), iii) more frequent *TP53* mutations in nonBasal1 tumors (Fig. 6a, two-sided Fisher's test, *p* = 0.02), iv) higher scores of the basal gene expression metagene in nonBasal1 tumors, v) higher scores of the steroid response expression metagene in nonBasal2 tumors, and vi) higher *AR* mRNA expression in nonBasal2 tumors (Supplementary Fig. S6c). Based on CIBERSORTx cell deconvolution data, no differences in computed scores for epithelial, macrophage, stroma, B-cell, endothelial, or T-cell types were observed between the nonBasal subgroups (two-sided Wilcoxon test, *p* > 0.05).

Differential analysis identified 1740 genes differentially expressed between nonBasal subgroups, 60% of which had higher expression in the nonBasal2 LAR subgroup (Supplementary Data 4b). Pathway analysis of these 1740 genes identified strong enrichment of genes involved in fatty acid, xenobiotic, and bile acid metabolism hallmarks, as well as estrogen and androgen response signatures (Supplementary Data 4c). To further analyze the differentially expressed gene set, we

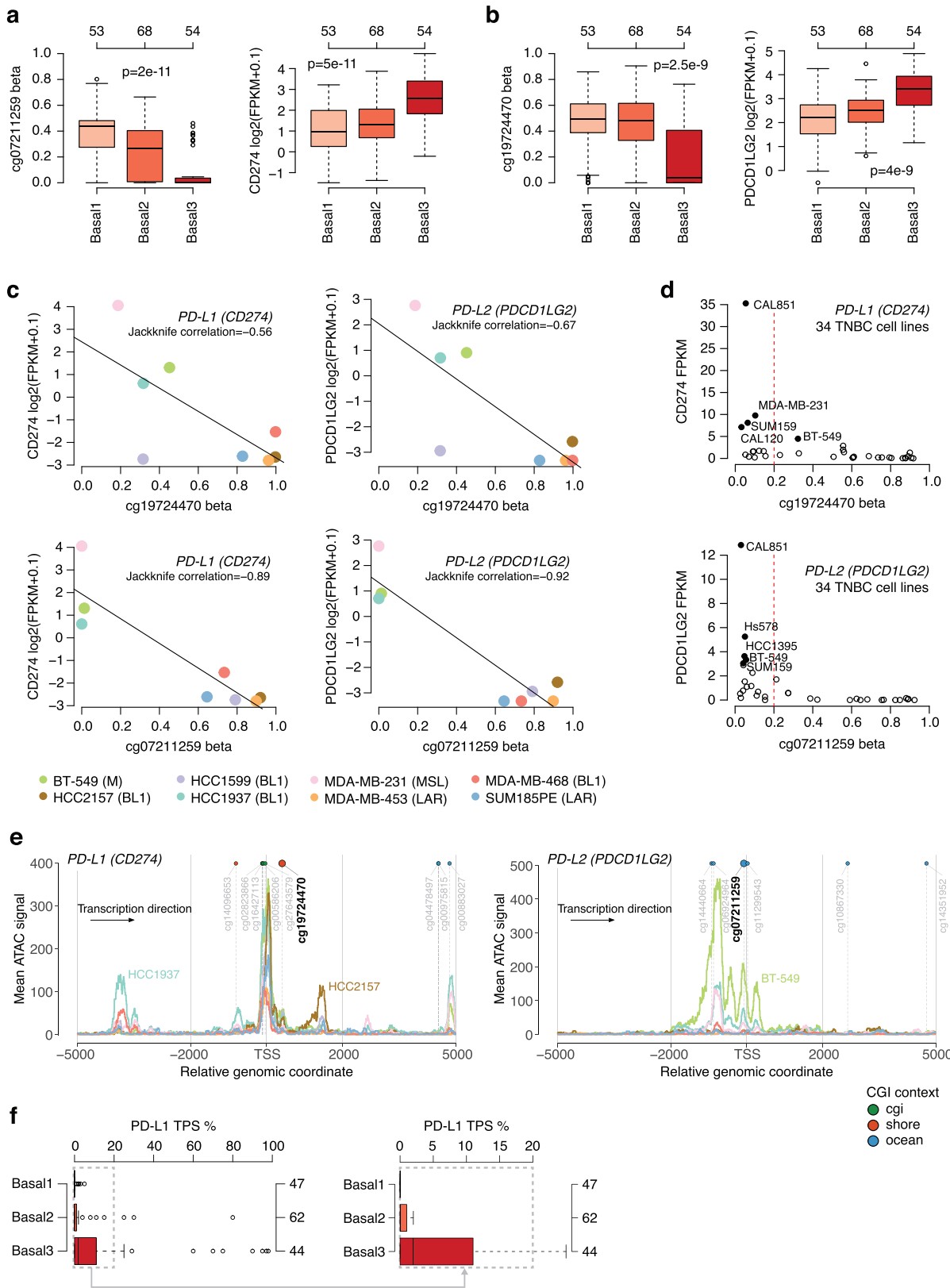

performed gene network analysis as originally described by Fredlund[33]. Three distinct networks were identified (network1-3). Gene set analysis aligned these with interferon signaling (network1, $n = 17$ genes), proliferation (e.g., mitotic spindle and G2M checkpoint, network2, $n = 83$ genes) and metabolism (xenobiotic, fatty acid, and bile acid), estrogen and androgen response, adipogenesis and cholesterol

homeostasis (network3, $n = 291$ genes) (Supplementary Data 4d, e). When compared across Basal and nonBasal subgroups, the metabolism network (network3) appeared highly specific to nonBasal2 tumors (Fig. 7a), tumors that also showed the highest *FOXA1* expression (Fig. 7b). To provide a broader view of the metabolism network in breast cancer, we computed similar network3 expression scores for

**Fig. 5 | PD-L1 epigenetic and expression patterns in Basal1-3 tumors. a** Beta values and FPKM expression for the significantly correlated pair cg07211259 - *CD274* (*PD-L1*). Two-sided *p*-value calculated using Kruskal-Wallis test for Basal tumors. Top-axis reports group sizes. **b** Same as (**a**) but for the pair cg19724470 - *PDCD1LG2* (*PD-L2*). Top-axis reports group sizes. **c** Beta values and mRNA expression for the cg07211259 - *CD274* and the cg19724470 - *PDCD1LG2* pairs in eight TNBC cell lines. **d** Scatter plots of beta values for CpGs specific to *PD-L1* and *PD-L2* versus FPKM gene expression in 34 TNBC cell lines. The five cell lines with the highest expression per gene are identified. **e** Mean ATAC-signal in eight TNBC cell lines in the promoter region of *PD-L1* and *PD-L2*. TSS = transcription start site of gene. The correlated CpGs cg19724470 and cg07211259 are highlighted. Cell line annotations (colors) are the same as in panel (**c**). **f** PD-L1 tumor positive score (TPS), representing the estimated proportion of tumor cells expressing PD-L1 on their surface, for all Basal1-3 samples, and in detail for those with lower values (excluding outliers). Right-axis reports group sizes. Boxplot elements correspond to: (i) center line = median, (ii) box limits = upper and lower quartiles, (iii) whiskers = 1.5x interquartile range. Source data are provided as a Source Data file.

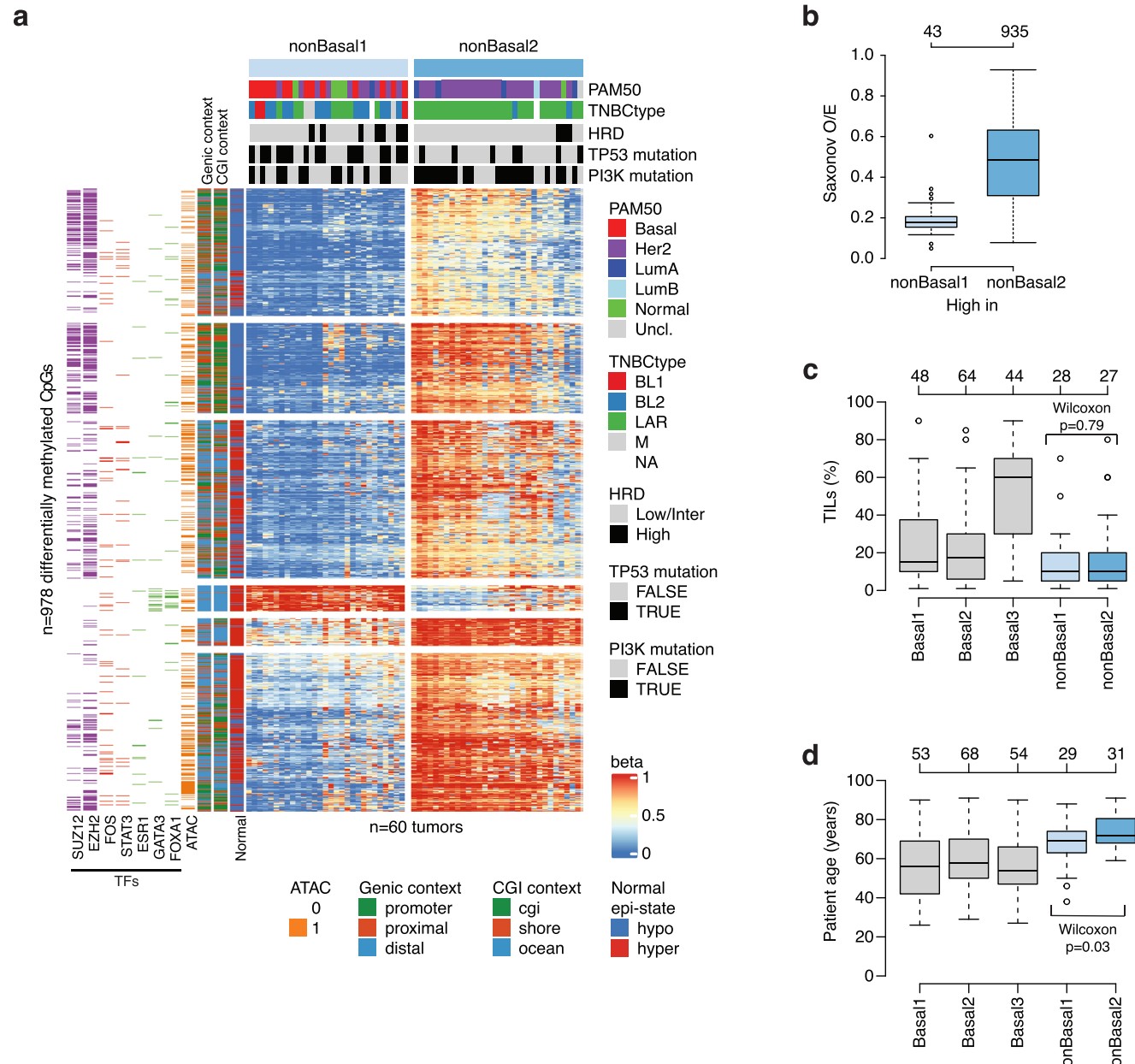

**Fig. 6 | Epigenetic subgroups within the main nonBasal epitype. a** Heatmap of DNA methylation beta values for 978 CpGs differentially methylated between the nonBasal1 and nonBasal2 subgroups. Sample annotation tracks show PAM50 subtypes, TNBCtype, homologous recombination deficiency (HRD) status, *TP53* mutations, and PI3K pathway mutations. Clustering of CpGs (rows) was performed using Euclidean distance and ward.D linkage. CpG annotation tracks show if CpGs are located within transcription factor (TF) binding sites (color = yes, white = no), different CpG contexts, and beta values seen in normal breast samples. **b** Saxonov observed/expected (O/E) values for the 978 CpGs divided by whether they have higher beta values in the nonBasal1 or nonBasal2 subgroup. **c** Boxplots of tumor infiltrating lymphocyte (TIL) percentages for tumors in the Basal and non-Basal subgroups. **d** Boxplots of patient age at diagnosis versus NMF subgroups. All reported *p*-values from statistical tests are two-sided. Boxplot elements correspond to: (i) center line = median, (ii) box limits = upper and lower quartiles, (iii) whiskers = 1.5x interquartile range. Top-axis in boxplots reports group sizes. Source data are provided as a Source Data file.

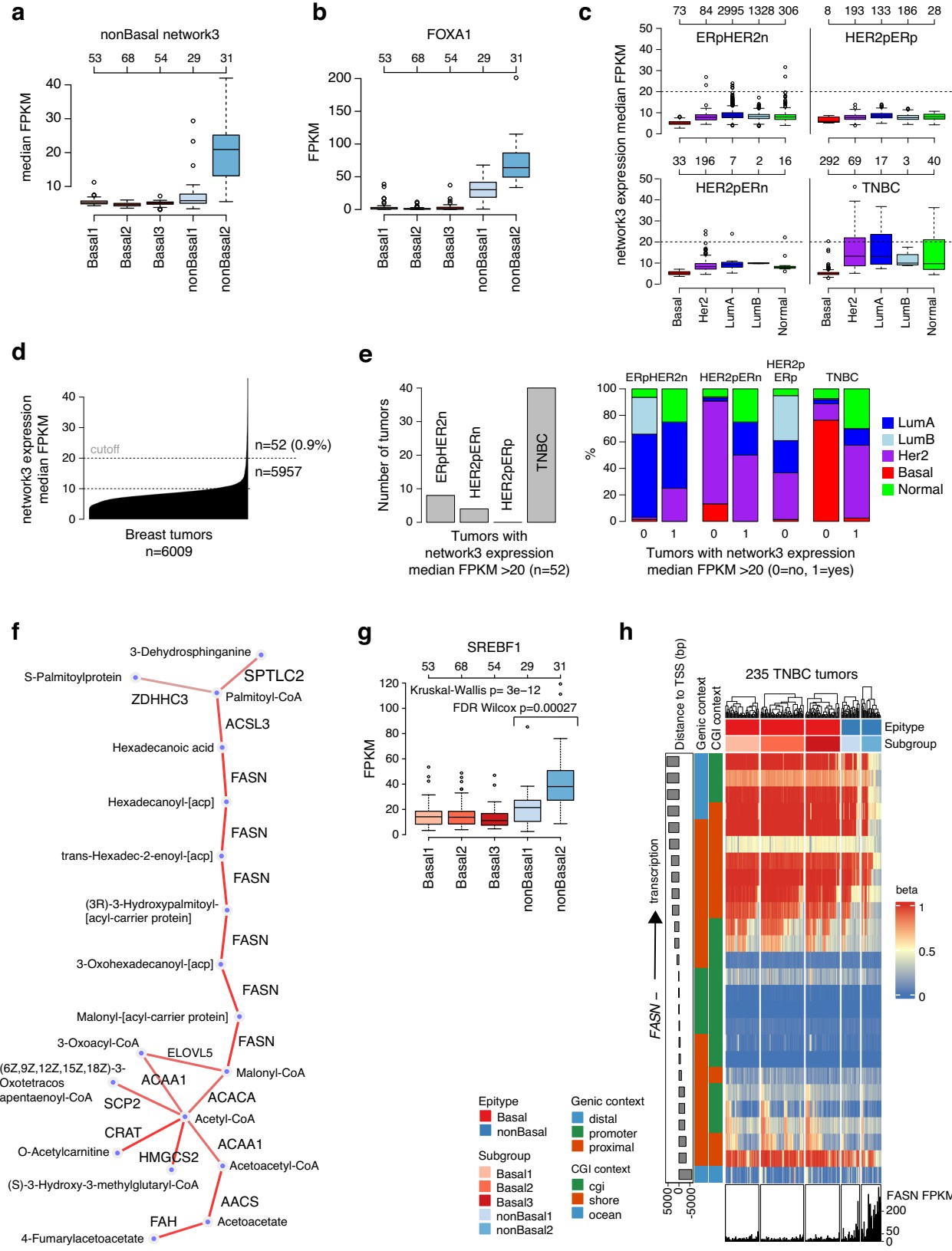

6009 SCAN-B tumors not present in the discovery cohort, representing all clinical subgroups and molecular subtypes. Firstly, across clinical subgroups defined by ER, PR, and HER2-status and further subdivided by PAM50 subtypes, expression of the metabolism network was consistently lowest in PAM50 Basal tumors (Fig. 7c). Secondly, high expression of the network was rare in breast cancer

(Fig. 7d). Thirdly, tumors with high expression were predominantly non-Basal TNBC, consistent with the TNBC-specific findings in the SCAN-B discovery cohort (Fig. 7e).

To investigate which cell types express network3 genes, whether non-malignant cells in the TME or putative cancer cells, we used the Single Cell Portal[37] and the TNBC study by Wu et al.[38]. For most genes in

**Fig. 7 | A metabolism gene network active in nonBasal2 tumors. a** Median FPKM expression for network3 genes (n = 291) across Basal and nonBasal subgroups. **b** FPKM expression of *FOXA1* versus Basal and nonBasal subgroups. **c** Median FPKM expression for network3 genes across 6009 primary SCAN-B breast cancers divided by clinical subgroups (p = positive, n = negative) and PAM50 subtypes. **d** Ordered network3 expression for the 6009 tumors shows a low number of high expressing cases with a median FPKM cutoff of >20 identifying 52 tumors with high network3 expression. **e** Clinical subgroup composition of the 52 high-expressing tumors from (**d**) and PAM50 subtype composition of low- and high-expressing tumors within clinical subgroups including all 6009 tumors from (**d**). **f** GAM[36] metabolite network based on atoms for genes in network3. Genes are in capital letters and metabolites

in lower case letters. **g** FPKM expression of the SREBP-1 (*SREBF1*) transcription factor versus Basal and nonBasal subgroups. Two-sided Kruskal-Wallis p-value computed for an all group analysis, two-sided Wilcoxon's test p-value for the difference between nonBasal1 and nonBasal2 groups. **h** DNA methylation heatmap of CpGs in the *FASN* promoter region for 235 SCAN-B TNBC tumors and corresponding *FASN* FPKM values. Arrow indicates direction of expression for the gene (minus strand). All reported p-values from statistical tests are two-sided. Boxplot elements correspond to: (i) center line = median, (ii) box limits = upper and lower quartiles, (iii) whiskers = 1.5x interquartile range. Top-axis in boxplots reports group sizes. Source data are provided as a Source Data file.

network3 with some expression reported by the Single Cell Portal, we found highest and most frequent expression in epithelial basal cycling and/or epithelial luminal mature cells, and not immune cells, endothelial cells, nor cancer-associated fibroblasts (Supplementary Fig. S6d). Using the GAM web service for network3 genes, we identified a set of metabolites related to e.g. synthesis of palmitate from acetyl-CoA and malonyl-CoA into long-chain saturated fatty acids through e.g. fatty acid synthase gene (*FASN*), for which important enzymes present in network3 are highly upregulated in nonBasal2 tumors (Fig. 7f, Supplementary Fig. S6e).

To analyze epigenetic regulation of network3 genes we reperformed the differential methylation analysis in nonBasal tumors, requiring no significant p-value but instead an absolute beta difference >0.5 between the two subgroups (compared to >0.25 in Fig. 6a). This reanalysis identified 25523 CpGs, again with a distinct *GATA3/FOXA1*-enriched CpG cluster (n = 2954 CpGs) showing hypermethylation in nonBasal1 tumors (Supplementary Fig. S6f). Based on these 25523 CpGs, we performed a CpG-gene correlation analysis, filtering the output to 1882 significant CpG-gene pairs with negative correlations and CpGs within 5 kbp of TSSs (Supplementary Data 4f). Of the 1882 CpG-gene pairs, 211 CpGs connected to 147 unique genes were present in the extended *FOXA1/GATA3*-enriched CpG cluster shown in Supplementary Fig. S6f. Two of these genes, *FASN* (fatty acid synthase) and *ACSL3* (Acyl-CoA Synthetase Long Chain Family Member 3), are present in the GAM network shown in Fig. 7f. Moreover, significant CpG-*FASN* and CpG-*ACSL3* pairs were only found in the *FOXA1/GATA3*-enriched CpG cluster hypomethylated in nonBasal2 tumors. *FASN* is pivotal for the increased production of fatty acids and is suggested to promote cell proliferation, cell invasion, metastasis, angiogenesis, and to have a role in immune escape in cancer (see ref. 39 for review). This gene appears to be predominantly expressed in epithelial basal cycling and epithelial luminal mature cells (Supplementary Fig. S6d), and scRNAseq data for four TNBC cell lines demonstrated *FASN* expression in nearly all cells (93%) of the SUM185PE LAR cell line (Supplementary Fig. S6g).

*FASN* is proposed to be connected to cell proliferation via activation of the PI3K/AKT/mTOR pathways that govern the expression of the Sterol Regulatory Element-Binding Protein 1 (SREBP-1) (*SREBF1*) transcription factor that induces *FASN* expression[39]. Consistently, *SREBF1* was among the 1740 genes differentially expressed between the nonBasal subgroups with significantly higher expression in nonBasal2 tumors (FDR-adjusted Wilcoxon test, p = 0.00027) and very low expression in Basal tumors (Fig. 7g). Additionally, mutations in the PI3K-AKT pathway are particularly frequent in nonBasal2 tumors (Fig. 6a). Promoter methylation maps suggest that the background tissue state for *FASN* shore CpGs is hypermethylated outside of a promoter-overlapping, and presumably methylation-resistant, CGI region (Supplementary Fig. S6h). *FASN* expression patterns are therefore consistent with epigenetic regulation by hypomethylation in many nonBasal2 tumors in the SCAN-B discovery cohort (Fig. 7h) and in LAR TNBC cell lines (Supplementary Fig. S6i). Investigation of ATAC signals in the *FASN* promoter region in TNBC cell lines showed, in contrast to *FOXA1*, a more homogeneous pattern between cell lines,

with no clear differences in chromatin accessibility between LAR and non-LAR cell lines (Supplementary Fig. S6j). In contrast to *FASN*, we found no support for epigenetic regulation of *ACSL3* or *SREBF1* based on similar investigations.

### Epigenetic regulation of other metabolism network3 genes
The directed analysis described above identified *FASN* as a likely epigenetically regulated gene part of the metabolism gene network (network3) expressed in the nonBasal2 subgroup in both the SCAN-B discovery and validation cohorts. To investigate if additional metabolic genes showed evidence of epigenetic regulation, we revisited the list of 1882 significant CpG-gene pairs with negative correlation and proximity to a TSS in nonBasal tumors (Supplementary Data 4f). Given the apparent specificity of the metabolism network to nonBasal2 tumors, and that these are relatively few in the respective SCAN-B cohorts, we merged the SCAN-B discovery and validation cohorts to gain power (n = 371 tumors in total). From the list of 1882 CpG-gene pairs, we identified 48 metabolism network genes with negative correlation to at least one CpG located <5 kbp away from the TSS. Based on promoter methylation maps and matched FPKM expression in the 371 tumors, a set of 12 candidate genes (*APMAP*, *ATP13A4*, *MUCL1*, *KYNU*, *IQGAP2*, *IDI1*, *ECHDC2*, *EAF2*, *CLDN8*, *PHYHD1*, *FAM174B*, and *LONP2*) was identified with elevated expression linked to epigenetic regulation primarily in nonBasal2 tumors. Not all these genes appear to be specifically associated with metabolism, and they may have been detected because of correlated expression patterns rather than functional similarity. To further refine the number of candidates, we analyzed the mRNA and protein expression for each gene in the eight TNBC cell lines using RNAseq and global proteomic data, finding support of expression for *APMAP* and *IQGAP2* in LAR cell lines (Supplementary Fig. S6k, l).

### Validation of TNBC epitypes and their subgroups
To validate our identification of two major DNA methylation groups in TNBC, we performed two-group NMF clustering of tumor purity-adjusted DNA methylation data from 136 TNBCs using the 5000 most variant distal-ATAC CpGs in a validation cohort also from SCAN-B (Supplementary Data 1b). This de novo approach divided the cohort into a Basal epitype (n = 89 tumors, 91% PAM50 Basal) and a nonBasal epitype (n = 47 tumors, 30% PAM50 Basal). NMF clusters derived instead using the 28565 differentially methylated CpGs identified in the discovery cohort demonstrated highly overlapping sample classification (92% epitype agreement, two-sided Chi-square test, p < 2e−16). Corroborating findings from the discovery cohort, TNBCtype LAR tumors were predominantly in the nonBasal de novo epitype (81% of 32 LAR tumors), and the Basal epitype was a mix of all TNBCtype subtypes (Fig. 8a). Additionally, we observed higher expression of the basal metagene in the Basal epitype (two-sided Wilcoxon's test, p = 3e−12) and higher steroid response metagene scores in the nonBasal epitype (two-sided Wilcoxon test, p = 8e−14). Finally, the epigenetic patterns of *FOXA1* hypermethylation and *FOXC1* hypomethylation in Basal tumors were also reproduced in the validation cohort (Supplementary Fig. S7a-b).

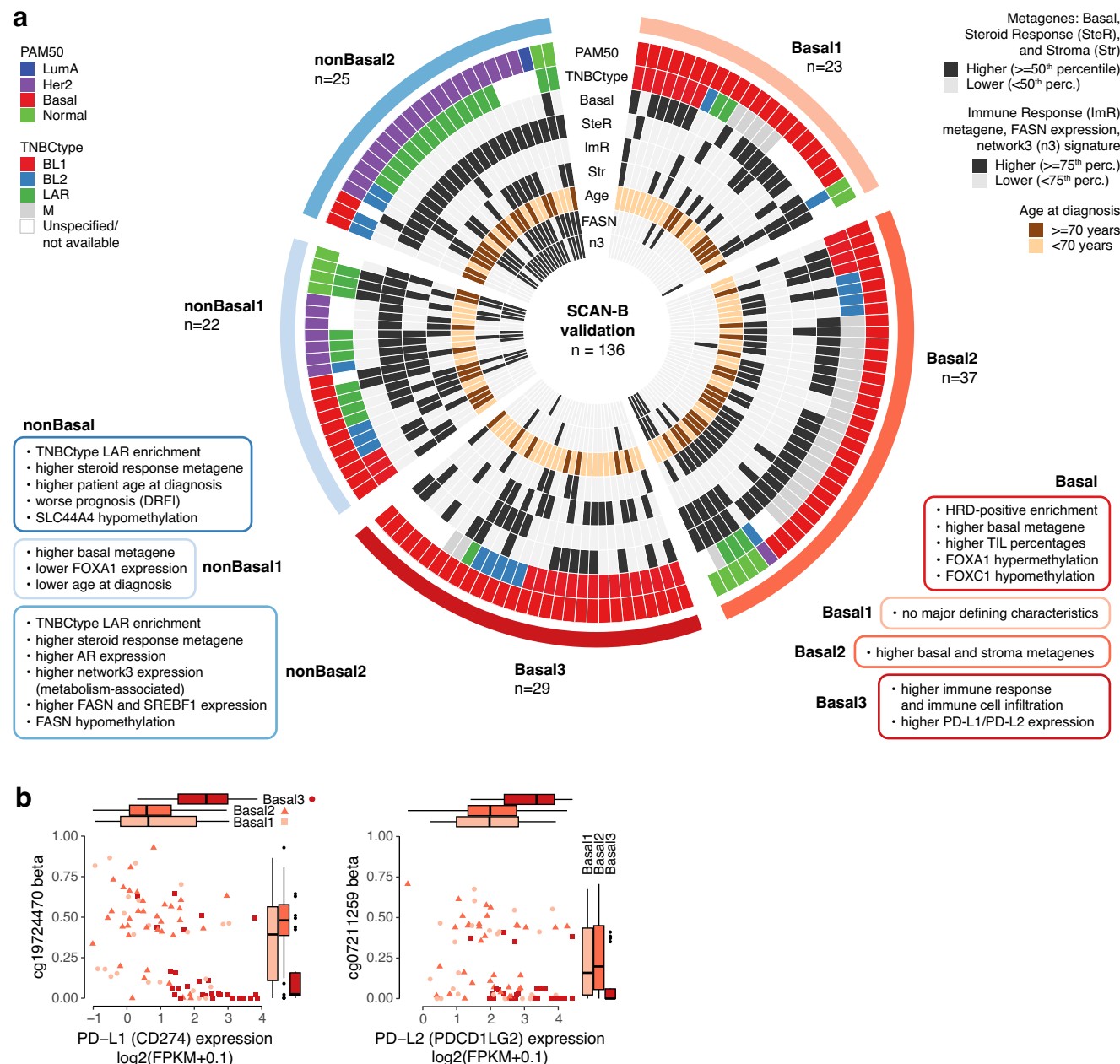

**Fig. 8 | Epigenetic subgroups in the SCAN-B TNBC validation cohort. a** Circular heatmap showing 136 samples classified into five DNA methylation subgroups as columns and the following characteristics as rows: PAM50 and TNBCtype subtypes, different expression metagenes (basal, steroid response, immune response, stroma), patient age at diagnosis, *FASN* expression, and gene network3 (metabolism-associated) expression. **b** *PD-L1* and *PD-L2* expression against beta values of the correlated cg19724470 and cg07211259 CpGs respectively located close to their TSSs, divided by Basal subgroups. Boxplot elements correspond to: (i) center line = median, (ii) box limits = upper and lower quartiles, (iii) whiskers = 1.5x interquartile range. Source data are provided as a Source Data file.

To validate the epigenetic subgroups within the Basal epitype, we chose to perform three-group NMF clustering of Basal tumors using tumor purity-adjusted DNA methylation data for the 2845 CpGs identified as differentially methylated within this epitype in the discovery cohort. Interestingly, despite significant agreement between NMF strategies (two-sided Chi-square test, $p = 4e-09$), sample classification with a de novo approach (using instead the 5000 most varying distal-ATAC CpGs) showed discrepancy in class agreement on an individual tumor level (Supplementary Fig. S7c), suggesting that subdivision of the Basal epitype may be less distinct with limited sample size. Like in the SCAN-B discovery cohort, TNBCtype subtypes were distributed in all Basal subgroups (Fig. 8a). Based on metagene expression patterns, Basal3 tumors demonstrated higher immune response (two-sided

Wilcoxon test, $p = 8e-09$) and Basal2 tumors showed higher stroma expression (two-sided Wilcoxon test, $p = 7e-05$), also in line with the discovery findings. Differential analysis identified 1060 genes differentially expressed specifically in Basal3 tumors that were highly associated with immune response, confirming all significant hallmarks in the discovery cohort's Basal3 subgroup (Supplementary Data 3b). Genes differentially expressed in the Basal2 subgroup ($n = 1006$) were associated with the EMT, G2M checkpoint, E2F and MYC target hallmarks. No associations with any hallmarks were observed for the 342 Basal1 genes in the validation cohort. Finally, Basal3 tumors showed elevated *PD-L1* and *PD-L2* mRNA expression and matched hypomethylation of the cg19724470 and cg07211259 CpGs located close to the respective genes' TSSs (Fig. 8b).

To validate the epigenetic subgroups within the nonBasal epitype, we applied NMF clustering as described for the Basal epitype but using the 978 CpGs differentially methylated in the nonBasal discovery cohort and a two-group solution. Using the approach with distal-ATAC CpGs would bring similar results as class agreement was high between NMF strategies (83%, two-sided Chi-square test, $p = 2e{-}05$). Similar to the discovery cohort, the nonBasal2 subgroup ($n = 25$ tumors) showed enrichment of the TNBCtype LAR subtype (73% of tumors) while the nonBasal1 subgroup ($n = 22$ tumors) comprised a mix of TNBCtype subtypes (Fig. 8a). Also consistent with the discovery cohort, nonBasal1 tumors showed higher basal expression (two-sided Wilcoxon test, $p = 0.03$) and nonBasal2 tumors showed a trend towards higher steroid response (two-sided Wilcoxon test, $p = 0.12$) based on expression metagenes. NonBasal2 tumors also showed higher age at diagnosis, higher expression of the metabolism metagene (network3), and higher FPKM levels of *FOXA1*, *AR*, *FASN* and its transcription factor *SREBF1* compared to nonBasal1 tumors (Fig. 8a, Supplementary Fig. S7d). Finally, promoter region analysis of *FASN* confirmed a putative epigenetic regulation of this gene by hypomethylation of shore/CGI CpGs primarily in nonBasal2 tumors from the validation cohort (Supplementary Fig. S7e).

## Discussion

In this work we have profiled the DNA methylation landscape of primary TNBC with two methodological modifications compared to previous studies[18–20], namely the use of purity-adjusted DNA methylation data and the consideration for the effect of CpG context on sample clustering. One of the most common ways of selecting CpGs for cluster analysis is by filtering for high beta variance without considering CpG density or genomic context. Our results suggest that this, at least in the context of TNBC, can lead to differences in clustering dynamics, potentially due to fundamentally and functionally different or stochastic processes being intermingled.

Several mRNA subtyping schemes have been proposed for TNBC[5,8,10], some with associated epigenetic characteristics (e.g., TNBCtype subtypes)[40]. Except for perhaps the TNBCtype LAR subtype enriched in the nonBasal2 DNA methylation subgroup, our study provides less support for the association between TNBCtype and DNA methylation patterns as illustrated by, e.g., stratified expression of *FOXA1* and an extensive one versus rest (e.g., BL1 vs non-BL1) differential methylation analysis of TNBCtype subtypes in all tumors, Basal tumors, and nonBasal tumors only (Supplementary Fig. S8). Our observation is not surprising as bulk mRNA analyses merge the expression patterns of all cells present in a tissue sample and analyzed together. Consequently, TNBC mRNA subtypes represent a mix of combined gene expression patterns from non-malignant (TME/TIME) and malignant cells. In this setting, tumor methylome profiles can both provide refinement of existing mRNA subtypes and be an important part of integrative multi-omics subtyping studies in TNBC. A small patient subset gaining importance in breast cancer comprises patients with ER-low tumors (1–9% positive ER staining) with increasing support emerging that such patients should be considered for clinical trials focused on TNBC[41]. TNBC status in SCAN-B tumors is defined according to Swedish guidelines (<10% ER-positivity), allowing us to investigate the distribution of tumors with 1-9% staining versus NMF clusters ($n = 29/235$, 12%). These tumors cannot be distinguished from other TNBCs in the discovery cohort, and we found no enrichment of these in the epigenetically defined Basal/nonBasal subgroups (two-sided Chi-square test, $p > 0.05$).

Based on our methodological approach, we postulate the existence of two major epigenetic backbones in TNBC, a Basal and a nonBasal epitype, that should represent distinct DNA methylation ground states of the disease. These epitypes align well with the intrinsic PAM50 subtypes in both the SCAN-B discovery and validation cohorts. In previous DNA methylation studies in TNBC that have used different CpG selection strategies, not considered CpG context, nor

used tumor purity-corrected DNA methylation data, we find both partial agreement and discordance to our proposed ground states. In the study by Lin et al., two of three subgroups show clinicopathological characteristics that could align with a Basal and nonBasal division, although sample numbers are small and mRNA profiling is lacking[19]. In the study by DiNome et al. reporting four subgroups, while one subgroup was enriched for LAR tumors, 69.2% of tumors in that subgroup were classified as PAM50 Basal-like[20], unlike in our subgroups. Notable features of the Basal and nonBasal epitypes were their strong association with clinical parameters like patient age, mRNA subtypes, HRD status, and TIMEs, consistent with what has been reported in non-epigenetic based studies for this major subdivision. Notably, the epigenetic stratification also divided TNBC tumors subtyped as PAM50 Normal-like into the Basal/nonBasal context with aligned expression and epigenetic patterns to that of other tumors in these epitypes, indicating that the underlying PAM50 subtype of these specific tumors might be masked in bulk RNAseq data.

The division into the Basal and nonBasal main epitypes aligned strongly with mRNA expression of prototypical pioneering factors and steroid response genes in breast cancer like *FOXA1* and *GATA3*, demonstrating concordant DNA methylation patterns at transcription factor binding sites across differentially methylated CpGs in our analyses. Based on differential methylation analysis and correlation with RNAseq data, we identified that *FOXA1*, a key transcription factor essential for ER and AR attachment to chromatin and the subsequent transcriptional induction of luminal genes in breast cancer cells (see ref. 42 for a review), shows a strikingly selective correlated pattern between specific hypermethylation in the CpG dense promoter region and mRNA downregulation in Basal-like breast cancer, irrespective of ER, PR, and HER2 status. Consistently, specific CpGs in the *FOXA1* promoter have previously been reported to display differences between PAM50 subtypes[43]. Notably, differential methylation around the TSS was also noted for the Choline Transporter-Like Protein 4 gene, *SLC44A4*, also included in the steroid response metagene, for which pan-cancer studies have demonstrated promoter region hypomethylation and expression in multiple tissue types[44].

In contrast to *FOXA1*, the transcription factors *ELF5* and *FOXC1* presented an opposite pattern with promoter region hypomethylation and higher expression in Basal tumors compared to nonBasal tumors. Overexpression of *FOXC1* in Basal tumors is well-established, and it has been implicated as a driver of cancer development, progression, metastasis, and drug resistance in other cancer forms (see ref. 45 for review). Taken together, while larger tumor sets and more detailed epigenetic profiling methods will likely identify additional genes, our findings indicate regulation of key pioneering and transcription factors involved in steroid signaling and organismal development as underpinning the major epigenetic subdivision in TNBC.

The Basal epitype was further subdivided into three subgroups (Basal1, Basal2, Basal3) and the nonBasal epitype into two subgroups (nonBasal1, nonBasal2) using NMF clustering within each epitype. To the methylation-defined subgroups we mapped the phenotypic landscape defined by TIME features, genetic alterations, transcriptional patterns, metabolic processes, and characterized potentially differing immune evasion mechanisms between them. Within the Basal epitype, the epigenetic subgroups were not associated with specific differences in somatic driver alterations or mutational processes, and they comprised a mix of TNBCtype mRNA subtypes. A finding of high interest was the strong immune association of the Basal3 cluster, and perhaps in particular an intriguing association of PD-L1 expression (mRNA and protein) with differential DNA methylation in putative gene-regulatory elements in both tumors and cell lines. Co-expression of PD-L1 and PD-L2 has been noted in many malignancies[46], and correlation between certain CpGs and PD-L1/PD-L2 expression has been reported in bulk TNBC tumor tissue[47]. In cell lines, including MDA-MB-231, it has been proposed that a super-enhancer element located between *PD-L1* and

*PD-L2* may be involved in synchronous transcription[46], possibly through a new topologically associating domain (TAD) that would allow transcription factors such as STAT3 and IRF1 to recruit to *PD-L1*[48]. Interestingly, a hallmark signature enriched in the gene set differentially expressed specifically in Basal3 tumors was IL6 JAK STAT3 signaling. However, of the CpGs present on the EPIC platform and located in the proposed super-enhancer region, none were significantly correlated with *PD-L1* or *PD-L2* FPKM levels in our cell lines or discovery tumor cohort. It thus remains to be proven whether the proposed super-enhancer and TAD change has a role in primary tumors. Unfortunately, the resolution of current Illumina platforms limits the ability to more closely resolve the methylation status of the *PD-L1/PD-L2* promoter region warranting a more in-depth characterization by alternative methods. Still, our observations may suggest that epigenetic regulation is necessary, but not sufficient, for the prototypical immune evasion mechanism of elevated PD-L1 expression on the tumor cell surface in TNBC. In this context, the observed heterogeneity in both scRNAseq expression of *PD-L1/PD-L2* and PD-L1 IHC TPS scores also suggest high intratumor heterogeneity. In contrast, tumors in the Basal2 subgroup appear stroma rich, enriched for immunosuppressive patterns of EMT, hypoxia and TGFB signaling[49], and express multiple genes known to be immunosuppressive like Transforming Growth Factor Beta Receptor 1 (*TGFBR1*), galectins (e.g., galectin 1, *LGALS1*) and caspases (e.g., *CASP2, CASP6, CASP10*). Tumors in Basal1 appear more mixed with respect to the TIME and show less distinct mRNA signatures, although a weaker enrichment for the specific metabolism pathways was observed in the SCAN-B discovery cohort.

Compared to the weak metabolic association of the Basal1 subgroup, the division of the nonBasal epitype into a highly LAR-enriched and distinct nonBasal2 subgroup and a more mixed nonBasal1 subgroup was strongly associated with both mRNA expression of a metabolism network highly specific for nonBasal2/LAR tumors and a tumor-hypermethylated phenotype targeting high CpG density contexts compared to other TNBC subgroups. While the complete regulatory mechanisms of this seemingly specific metabolic state in nonBasal2 tumors remain to be elucidated, the finding of some support for epigenetic regulation of a set of network genes like the scaffold protein IQ motif-containing GTPase activating protein 2 (*IQGAP2*), Adipocyte Plasma Membrane Associated Protein (*APMAP*), and *FASN* (a pivotal gene for fatty acid production) in nonBasal2 tumors is intriguing. *IQGAP2* has been proposed as a tumor suppressor gene that could participate in metabolism, based on *IQGAP2*-null mice having impaired uptake of long-chain fatty acids[50]. *APMAP* enables arylesterase activity and has been proposed to promote cholesterol-induced EMT in prostate cancer[51], and to protect cancer cells against phagocytosis[52]. Notably, the overexpression and methylation patterns of *APMAP*, *IQGAP2*, and *FASN* in LAR cell lines support a hypothesis of epigenetically regulated tumor cell expression that is retained despite in vitro culturing. There is increasing evidence of involvement of *FASN* in several hallmarks of cancer, including cell proliferation, metastasis, and angiogenesis (see ref. 39 for a review). Interestingly, *FASN* is proposed to be connected to cell proliferation by the PI3K-AKT and mTOR pathways[39], and the former is highly targeted by mutations in nonBasal2 tumors. Finally, FASN has also been proposed to be involved in immune escape by repressing the activation of pro-inflammatory cells and promoting the recruitment of M2 macrophages and regulatory T cells in the tumor microenvironment[39]. In lack of in situ M1/M2 macrophage characterization, it remains to be determined if FASN overexpression represents a potential immune evasion mechanism directed in part by epigenetics in nonBasal2 tumors. In contrast to the nonBasal2 subgroup, nonBasal1 tumors appear more mixed regarding transcriptional and mutational patterns, while still showing higher age at diagnosis, hypomethylation and clear expression of *FOXA1*, and a general steroid response phenotype compared to Basal tumors. It remains to be determined if nonBasal1 tumors represent an intermediate developmental step on the way to a nonBasal2 phenotype from an unknown primary cell type, although an argument against this hypothesis would be the more frequent *TP53* driver alterations observed in nonBasal1 compared to nonBasal2 tumors.

Patients with systemic therapy included in this study were all treated according to national Swedish guidelines during inclusion years, which for the discovery cohort meant mainly a FEC-based (combination of 5-fluorouracil, epirubicin and cyclophosphamide) chemotherapy regimen[6]. Survival analyses demonstrated better outcome for patients with Basal compared to nonBasal tumors, a trend of better outcome in patients with Basal3 tumors compared to Basal1 and Basal2 patients, but no difference in outcome between nonBasal1 and nonBasal2 patients. The observed better prognosis for patients with immune-warm Basal3 tumors aligns with the established prognostic role of the TIME in TNBC[53]. In metastatic TNBC, high TILs/PD-L1 expression appears predictive of immune checkpoint inhibitor efficacy, while in early-stage pre-operatively treated disease pathological complete response rates are higher for immune-warm tumors with or without checkpoint inhibition (reviewed in ref. 53). While it may be speculated that patients with immune-warm, *PD-L1* expressing, Basal3 tumors could be the ones benefitting the most from immune checkpoint inhibitors like pembrolizumab, now part of clinical management of early-stage TNBC[54], it should be noted that patients with high TIL levels have as a group an intrinsically good prognosis even without adjuvant treatment[55]. Other characteristics of the DNA methylation subgroups aligning with recent or emerging treatment options in TNBC include for nonBasal1 and nonBasal2 patients potential androgen receptor inhibition therapy (based on increased *AR* expression), PI3K/AKT inhibitors (based on frequent PI3K pathway alterations including *PIK3CA* mutations), or combinations thereof as outlined in different studies[56–58]. Interestingly, for the mainly older patients with nonBasal2 tumors a potential future treatment option may be inhibition of FASN, for which multiple inhibitors are under development[39]. While TROP2 antibody-drug conjugates are available for treatment of metastatic TNBC, we found no significant differences in mRNA expression of the TROP2 gene (*TACSTD2*) between the five DNA methylation subgroups (two-sided Kruskal-Wallis test, $p = 0.1$) suggesting no informative value for this therapy. Similarly, while 93% of all HRD-positive tumors (primarily driven by *BRCA1*/*BRCA2*-deficiency) were found in the Basal epitype, these did not match a distinct Basal subgroup (two-sided Chi-squared test, $p = 0.94$) suggesting that the latter would not be informative for PARP-inhibitor usage.

Limitations in the current study are that we only cover one aspect of epigenetics, DNA methylation, and that we rely on ATAC-seq regions delimited in breast cancer, not specifically in TNBC. To address these limitations, we made extensive use of epigenetic TNBC cell line data to determine which tumor methylation patterns could be viewed as intrinsic, as well as incorporated histone modification data from normal cells for contrast. While we can reproduce key findings for the Basal/nonBasal, Basal1-3, and nonBasal1-2 divisions in the SCAN-B validation cohort, we acknowledge that even larger multi-omics cohorts will be needed to provide further support to our findings and elucidate the specific cancer cell phenotypes linked to distinct TME/TIME contexts of the Basal epigenetic subgroups. While the discovery cohort has been shown to be population representative[6], a generally positive characteristic with respect to generalizability, it is representative of a specific demographic context of patients (Western Europe/Nordics). Consistently, it should be investigated whether the proposed DNA methylation patterns observed in the current study also hold true in other ethnic contexts. Moreover, the potential epigenetic regulatory finding for *PD-L1/PD-L2* in immune-inflamed TNBC should ideally be experimentally confirmed in vitro, and the metabolic alterations in nonBasal2 tumors should be quantified by comprehensive metabolomic methods to provide a high-resolution characterization of the metabolomic landscape in TNBC, as well as additional

drivers of this phenotype. Importantly, the current study cannot address the intriguing questions of why and when both broader and highly specific DNA methylation patterns, like the one revealed for *FOXA1* and *FOXC1*, emerge in Basal and nonBasal tumors. Currently we are also unable to answer whether the observed DNA methylation patterns that define the immune-warm Basal3 subgroup represent epigenetic immunoediting as proposed in glioblastoma by Gangoso et al.[59], and whether this occurs in response to the immune infiltration or is naturally occurring and selected for. Hypotheses to be tested are that the origin of these two main epitypes lies in either different cell types and/or connections to the molecular and physiological changes associated with normal breast development and menopause.

In summary, in this study we provide a tumor-focused DNA methylation analysis of TNBC demonstrating the importance of considering the influence of the tumor microenvironment as well as CpG context when interpreting DNA methylation data in solid cancer tissue. Our findings connect previously described genetic and mRNA subtypes of TNBC to a two-group main epigenetic backbone of Basal and nonBasal disease that may in turn be further subdivided into subgroups with seemingly distinct metabolic and immune-evasive strategies. Our analyses highlight plausible epigenetic regulation of important pioneering factors in breast cancer that may help to explain key transcriptional differences between Basal and nonBasal breast cancer irrespective of clinical subgroups. Based on our results, further research should be directed towards understanding the cell of origin of the epigenetic backbones, links between the epigenetic phenotypes and immune evasion strategies, and metabolic differences within the transcriptional and epigenetic subtypes of TNBC.

## Methods

### Inclusion and ethics statement
Patients were enrolled in the Sweden Cancerome Analysis Network – Breast (SCAN-B) study (ClinicalTrials.gov ID NCT02306096)[60,61] approved by the Regional Ethical Review Board in Lund, Sweden (registration numbers 2009/658, 2010/383, 2012/58, 2013/459, 2014/521, 2015/277, 2016/541, 2016/742, 2016/944, 2018/267) and the Swedish Ethical Review Authority (registration numbers 2019-01252, 2024-02040-02). All patients provided written informed consent prior to enrolment, including to publish information about sex and age. All analyses were performed in accordance with patient consent and ethical regulations and decisions.

### Patient cohort
Based on our recently reported Swedish population-representative TNBC cohort of 237 patients[6], we assembled 235 cases with complete whole genome sequencing (WGS), RNA sequencing, *BRCA1* hypermethylation status, global DNA methylation profiles (performed for this study and merged with DNA methylation data from[16]), in situ immunohistochemistry data, morphological assessments, and extensive treatment and clinical follow-up data. Clinicopathological characteristics and molecular data for patients are summarized in Supplementary Data 1a with additional details concerning inclusion/exclusion criteria, clinical end point definitions, measurements and details of clinicopathological variables, and molecular data provided in ref. 6. This patient set is hereon referred to as the SCAN-B discovery cohort. In Sweden, the definition of TNBC is a tumor with <10% of cells with IHC-staining for ER and PR (thus including tumors with 1–9% stained cells) and an IHC HER2-staining score of <2, or for patients with IHC 2+, a non-amplified ISH status. All SCAN-B data for ER, PR, and HER2 status were obtained from clinical routine analyses performed in regional pathology departments.

### Global epigenetic profiling and CpG annotation
Global epigenetic profiling of SCAN-B discovery tumors was performed for this study using Illumina Infinium MethylationEPIC v1.0

BeadChip (interrogating ~800000 CpGs) according to manufacturer's instructions at the Center for Translational Genomics, Lund University and Clinical Genomics Lund, SciLifeLab, using DNA extracted from tumor tissue preserved in RNAlater (Qiagen, Hilden, Germany). Basic processing of beadchip data was performed as described in ref. 31. Briefly, raw idat files were processed for the discovery cohort using the function preprocessFunnorm[62] as implemented in the R-package minfi[63] (v1.44). Default parameters were used with patient gender estimated using the built-in function getSex in minfi. Additionally, a platform-related effect on CpG methylation beta values between the two utilized probe chemistries was adjusted for using the method of Holm et al.[64]. Processed CpG beta values for the discovery cohort were further corrected for tumor purity using the approach described by Staaf and Aine[31] using tumor cell content estimations from WGS with the ASCAT[65] software obtained from[6]. For each CpG we built models that were subsequently used to adjust the original beta values and derive an inferred in silico normal estimate. After basic data processing and CpG beta correction, 760405 CpGs remained for analysis, referred to as purity-adjusted beta values. These were further filtered to exclude chrX/Y localization and non-CpG probes, leaving 741145 CpGs for analyses.

We compiled a custom feature annotation set for each CpG probe on the Illumina EPIC methylation platform using the same methodology described for the Illumina HumanMethylation450K array in Staaf and Aine[31]. This included assigning CpGs to a gene-centric context defined as promoter (+/- 500 bp centered on gene transcription start site, TSS), proximal (+/- 5 kbp centered on TSS and excluding the promoter window), or distal ( > 5 kbp from TSS) based on their genomic coordinates (referred to as genic context). For the gene-centric annotations, a consensus transcript model based on GENCODE v27 protein coding genes matching SCAN-B RNAseq data was built for each gene by collapsing of exons. The 5' most base was assigned as the consensus TSS and the 3' most base as the consensus transcription termination site. For the TCGA cohort profiled on Illumina 450 K, gene annotations were derived using the same method except the transcript models were based on the GENCODE 22 TCGA reference file gencode.gene.info.v22.tsv matching the TCGA RNAseq data (https://gdc.cancer.gov/about-data/gdc-data-processing/gdc-reference-files).
Probes were also assigned to a CpG-centric context defined as CpG island (CGI), shore, or ocean (referred to as CGI context)[26]. Local CpG density metrics (e.g., O/E) and contextual classifications for each probe were obtained using the methods of Saxonov et al.[23] for high (HCG) and low (LCG) CpG content and of Weber et al.[24] for HCP, ICP, and LCP. EPIC probe overlaps with ATAC-seq peak data generated on 74 TCGA breast cancer samples by Corces et al.[43] were calculated and used as a proxy for differentially open chromatin in breast cancer. Additionally, ENCODE candidate cis-regulatory elements[66] and ENCODE ChIP-seq peak overlaps for 340 transcription factors in 130 cell lines[67] were used to assess the regulatory potential of EPIC CpGs.

### DNA methylation clustering and differential analysis
Clustering of purity-adjusted DNA methylation data was performed using Non-negative Matrix Factorization (NMF) through the R package NMF (v0.28)[68] using nrun = 100 and seed = 221027. Heatmaps were produced using the pheatmap (v1.0.12) or the ComplexHeatmap (v2.20.0) R packages[69]. For identifying differentially methylated CpGs, beta values were compared using the Wilcoxon test for two group comparisons or the Kruskal-Wallis test if more than two test groups. For each CpG, if the standard deviation in a comparison was 0, then the $p$-value was set to 1. When filtering CpGs, we also used the absolute difference between mean beta per test group. When comparing adjusted tumor beta for a group to the inferred normal beta (using the method by Staaf and Aine[31]), we computed the mean tumor beta for the group in question and the corresponding mean inferred normal beta for the same samples. For each CpG we also classified its inferred

normal methylation state as either hypomethylated (hypo, mean beta across all analyzed tumors ≤0.5) or hypermethylated (hyper, mean beta across all analyzed tumors >0.5).

## Gene expression analyses

Gene expression profiling was performed using RNAseq and has been reported previously as fragments per kilobase per million reads (FPKM) values[6,70]. PAM50 classification was obtained from[70]. Other transcriptional classifications, including the TNBCtype-4 subtypes[8], were obtained as described in refs. 6,16. More specifically, TPM converted data for the SCAN-B discovery cohort's tumors to profile were provided to the TNBCtype webtool[71]. If the webtool returned an error due to inclusion of TNBC tumors proposed as "ER-positive", these cases were removed from the data set and then the reduced set was again provided to the web tool. The webtool returns correlations for each sample to the original 6 centroids described by Lehmann et al.[8]. For each tumor, the subtype class was assigned as the centroid with the highest correlation to the tumor's values. To transform the subtype class to the later proposed TNBCtype-4 classification, which does not include the IM and MSL subtypes, the subtype with the second highest correlation was assigned as the main subtype for tumors with a primary IM or MSL subtype. Cell type deconvolution results were obtained from Glodzik et al.[16] for CIBERSORTx[72]. Based on FPKM data, gene expression-based rank scores for eight biological metagenes in breast cancer originally defined by Fredlund et al.[33] were calculated as described in Nacer et al.[73]. Rank scores were computed individually for each tumor from FPKM data without any further normalization or data centering. For the SCAN-B validation cohort, FPKM data obtained from Staaf et al.[70] were used.

Gene network analysis was performed as originally outlined by Fredlund et al.[33]. Briefly, the strategy aims to identify genes with positively correlated expression (FPKM) in a sample set by computing all possible combinations of correlations. Here we used a Spearman correlation cutoff of 0.7 and required at least 8 connected genes when delineating networks. The output of the analysis is a SIF file of interconnected gene pairs that was further analyzed using Cytoscape[74] (v3.9.1). In Cytoscape we defined the final networks (lists of genes) based on that correlated genes reside as non-connected spheres of genes (see Fredlund et al.[33] for methodology). We performed pathway analyses of defined networks to identify enriched pathways as outlined below. For each tumor we computed a network metagene score as the median FPKM expression of all genes in the network.

Differential gene expression analysis was performed using FPKM data for 19675 genes from ref. 70. Before any statistical tests, an offset of 0.1 was added to all values, followed by log2 transformation. Differentially expressed genes were identified using the Wilcoxon or Kruskal-Wallis test on FPKM data depending on the number of groups to compare (Kruskal-Wallis for more than two groups, Wilcoxon's test for two groups). $P$-values were adjusted for multiple testing using the p.adjust R function with an FDR cutoff of $p < 0.05$. For each sample group tested, median expression was used to identify the directionality of significant gene expression. Specifically for three group comparisons, we also computed pairwise tests for each gene: i) compute the overall Kruskal-Wallis $p$-value for groups 1, 2, and 3; ii) compute the Wilcoxon $p$-value between groups 1 and 2; iii) compute the Wilcoxon $p$-value between groups 1 and 3; and iv) compute the Wilcoxon $p$-value between groups 2 and 3. The overall number of significant genes were determined as the set of genes with FDR $p < 0.05$, and with an FPKM difference > 1 between the group with the highest median expression versus the group with the lowest. The latter filtering step was employed to exclude genes called as significant but with very low expression values, and genes that might be significant by $p$-value due to large sample sizes but with modest effect size (difference in FPKM). To identify genes specifically up- or down-regulated in a specific group when more than two groups were tested, we employed the following

scheme: i) only genes with an overall significance (i.e., FDR adjusted Kruskal-Wallis <0.05 and FPKM difference >1) were considered; ii) based on the reduced gene set left, we adjusted each of the pairwise (e.g., group 1 vs group 2) Wilcoxon $p$-values for multiple testing using FDR (via the p.adjust R function). For genes to be significantly different for group 1, we then required: i) a significant Wilcoxon $p$-value (after multiple testing correction) for group 1 vs 2 and for group 1 vs 3; and ii) a non-significant Wilcoxon $p$-value (after multiple testing correction) for group 2 vs 3. Similar procedures were employed to all groups to identify group-specific differentially expressed gene sets.

Pathway analysis was performed using the R ClusterProfiler package (v4.12.1) and the R package implementations of KEGG, Gene Ontology (org.Hs.eg.db, v3.19.1), Reactome (ReactomePA v1.48.0), and Molecular Signatures Database (MsigDB, msigdbr v7.5.1) gene sets associated with Gene Set Enrichment Analysis (GSEA), as outlined in the ClusterProfiler vignette for the respective analysis. An adjusted $p$-value < 0.05 was used as the significance threshold in all analyses. A list including only significant genes was used as input to the analyses. If a gene universe was required, then the full set of 19675 genes was used.

## Correlation between DNA methylation and gene expression

Correlation between CpG beta values and FPKM expression for specific CpG sets was performed by first mapping each CpG in a set to nearby genes based on transcription start sites located within a 500 kb genomic window centered on the CpG. This creates a one-to-many-genes mapping for most CpGs, depending on the genome structure around the CpG. To reduce spurious correlations, only CpGs with a difference >0.25 between minimum and maximum beta values and genes with a difference >1 between minimum and maximum FPKM expression were included. Genes with mean FPKM expression <0.5 in the entire cohort were also excluded. FPKM values were then offset by adding 0.1 and log2 transformed. Next, we computed jackknife Pearson correlation coefficients between Illumina CpG probe methylation levels and FPKM gene expression levels across the tumor set investigated. Jackknife correlation was chosen in order to reduce spurious correlations driven by outliers in the 8-sample TNBC cell line cohort (see below). For methodological consistency this measure was also used in the SCAN-B discovery cohort. To derive significance cutoffs, jackknife correlations between beta values of 50000 random EPIC probes and expression of genes in identical 500 kb genomic windows were calculated separately for each cohort. The 2.5 and 97.5 correlation percentiles were used as cutoffs in the respective tumor and cell line cohorts.

## Whole genome sequencing analyses

WGS data, including mutational calls, mutational and rearrangement signatures, copy number profiles, and HRD classification by HRDetect[15] were obtained from Staaf et al.[6]. A combined annotation of PI3K/AKT pathway activation based on driver mutations was created by merging all WGS driver data for *PIK3CA*, *AKT1*, *PTEN*, *PIK3R1*, and *PIK3R2*. Specifically, an activated status for a tumor was set if any of the following conditions were detected: i) a mutation (not amplification) in codons 542, 545, 546, or 1047 in *PIK3CA*, ii) a frameshift or nonsense mutation in *PTEN*, iii) a c.49 G > A mutation in *AKT1*, or iv) a driver alteration in *PIK3R1* or *PIK3R2*.

## PD-L1 immunohistochemistry and tumor infiltrating lymphocytes (TILs)

PD-L1 was assessed immunohistochemically using the 22C3 clone on a Dako Autostainer Link 48 platform (Agilent, Inc., CA, U.S.) in formalin-fixed, paraffin-embedded tumor samples in a tissue microarray (TMA) where each sample was represented by two TMA cores with a diameter of 1 mm (TMA construction outlined in ref. 75). Preparations and stainings were done according to the manufacturer's instructions (PD-L1 IHC 22C3 pharmDx kit). PD-L1 expression was evaluated as tumor

positive score (TPS) in percentage as described in the manual for the PD-L1 IHC 22C3 pharmDx antibody (Agilent). The TMA core with the highest score was set as the score for the respective tumor. Of all 235 tumors, 207 could be evaluated. Additionally, TIL estimations (% TILs) were obtained from hematoxylin and eosin-stained whole slides from ref. 75 for 211 cases.

## TNBC cell lines and genomic analyses

Eight TNBC cell lines proposed to represent different TNBCtype mRNA subtypes (HCC2157:BL1, HCC1599:BL1, HCC1937:BL1, BT-549:M, MDA-MB-231:MSL, MDA-MB-468:BL1, SUM185PE:LAR, and MDA-MB-453:LAR[8]) were obtained from commercial (HCC2157, ATCC, www.atcc.org) or in-house cell line biobanks (all others, Division of Oncology, Lund University, Sweden) and used for ATAC-sequencing (ATAC-seq), RNAseq, DNA methylation, and proteomic analysis. HCC2157, HCC1599, HCC1937, and BT-549 were cultured in RPMI 1640 medium supplemented with 10% fetal bovine serum (FBS), 100 U/ml penicillin and 100 μg/ml streptomycin (PEST) (all from GE Healthcare HyClone™). SUM185PE was cultured in Ham's F-12 (GE Healthcare HyClone™) supplemented with hydrocortisone (1 μg/ml, Sigma-Aldrich), insulin (5 μg/ml, Sigma-Aldrich), HEPES (10 mM, Sigma-Aldrich) and 5% FBS, but without antibiotics. These cells were all cultured under standard culturing conditions (37 °C, 5% $CO_2$). MDA-MB-231, MDA-MB-468 and MDA-MB-453 were cultured in L-15 Leibovitz medium (GE Healthcare HyClone™) supplemented with 10% FBS and 1% PEST. These cell lines were cultured in 37 °C without $CO_2$. All cell lines were sent for mycoplasma testing (Mycoplasmacheck service, Eurofin Genomics) and confirmed as mycoplasma free; they were also sent for short tandem repeat (STR) profiling (Human cell line authentication service, Eurofins Genomics) to confirm cell line origin. DNA, RNA, and a protein flow-through fraction were extracted using the Qiagen AllPrep DNA/RNA Mini Kit (cat no 80204) following manufacturer's instructions.

ATAC-seq was performed on each cell line and analyzed as outlined by Arbajian et al.[76]. RNAseq libraries were prepared using the Illumina TruSeq stranded mRNA protocol and sequenced on the NovaSeq 6000 system at the Center for Translational Genomics (www.ctg.lu.se) in Lund, Sweden. Demultiplexing was performed using the bcl2fastq2 software (Illumina) with default settings and the quality was checked with FastQC. Reads were mapped to the GRCh38 reference genome using the HISAT2 software and annotation files from release 103. Finally, expression data in FPKM were calculated with StringTie. DNA methylation data were generated for each cell line using the Illumina Infinium MethylationEPIC v1.0 BeadChip (interrogating ~800,000 CpGs) according to manufacturer's instructions by the SNP&SEQ Technology Platform in Uppsala, Sweden (www.genotyping.se). Basic processing of beadchip data was performed as described above for the discovery cohort (without the purity adjustment step).

## Proteomic analysis of TNBC cell lines

RNAlater flowthrough from the eight TNBC cell lines was used for proteomic analysis. Each cell line was analyzed as two biological replicates. The total protein amount was estimated (DC kit, Biorad). Samples were then prepared for mass spectrometry analysis using a modified version of the SP3 protein clean-up and a digestion protocol[77,78], where proteins were digested by LycC and trypsin (sequencing grade modified, Pierce). In brief, around 100 μg of protein from each sample was alkylated with 4 mM chloroacetamide. Sera-Mag SP3 bead mix (22 μl) was transferred into the protein sample together with 100% acetonitrile to a final concentration of 60%. The mix was incubated under rotation at room temperature for 20 min. The mix was placed on the magnetic rack and the supernatant was discarded, followed by two washes with 70% ethanol and one with 100% acetonitrile. The beads-protein mixture was reconstituted in 100 μl LysC

buffer (0.5 M Urea, 50 mM HEPES pH: 7.6 and 1:50 enzyme (LysC) to protein ratio) and incubated for 3 hours. Finally, trypsin was added in 1:50 enzyme to protein ratio in 100 μl 50 mM HEPES pH 7.6 and incubated overnight. The peptides were eluted from the mixture after placing the mixture on a magnetic rack, followed by peptide concentration measurement (Micro BCA, Thermo Scientific, 23235). The samples were then pH adjusted using TEAB pH 8.5 (100 mM final conc.), 30 μg of peptides from each sample were labelled with isobaric TMT-tags (TMTpro 16 plex reagent) according to the manufacturer's protocol (Thermo Scientific), and 280 μg of peptides separated by immobilized pH gradient - isoelectric focusing (IPG-IEF) on 3-10 strips as described previously[79]. Of note, the labelling efficiency was determined by LC-MS/MS before pooling of the samples. For the sample clean-up step, a solid phase extraction (SPE strata-X-C, Phenomenex) was performed and purified samples were dried in a SpeedVac. An aliquot of approximately 10 μg was suspended in LC mobile phase A and 2 μg was injected on the LC-MS/MS system. Online LC-MS was performed as previously described[79] using a Dionex UltiMate™ 3000 RSLCnano System coupled to a Q-Exactive-HF mass spectrometer (Thermo Scientific). Each of the 72 plate wells was dissolved in 20 uL solvent A and 10 uL were injected. Samples were trapped on a C18 guard-desalting column (Acclaim PepMap 100, 75 μm x 2 cm, nanoViper, C18, 5 μm, 100 Å), and separated on a 50 cm long C18 column (Easy spray PepMap RSLC, C18, 2 μm, 100 Å, 75 μm x 50 cm). The nano capillary solvent A was 95% water, 5% DMSO, 0.1% formic acid; and solvent B was 5% water, 5% DMSO, 95% acetonitrile, 0.1% formic acid. At a constant flow of 0.25 μl min$^{-1}$, the curved gradient went from 6–10% B up to 40% B in each fraction in a dynamic range of gradient length followed by a steep increase to 100% B in 5 min. FTMS master scans with 60,000 resolution (and mass range 300–1500 m/z) were followed by data-dependent MS/MS (30,000 resolution) on the top 5 ions using higher energy collision dissociation (HCD) at 30% normalized collision energy. Precursors were isolated with a 2 m/z window. Automatic gain control (AGC) targets were 1e6 for MS1 and 1e5 for MS. Maximum injection times were 100 ms for MS1 and 100 ms for MS2. The entire duty cycle lasted ~2.5 s. Dynamic exclusion was used with 30 s duration. Precursors with unassigned charge state or charge state 1 were excluded. An underfill ratio of 1% was used. Orbitrap raw MS/MS files were converted to mzML format using msConvert from the ProteoWizard tool suite[80]. Spectra were then searched using MSGF+ (v10072)[81] and Percolator (v2.08)[82]. All searches were done against the human protein subset of Ensembl (105) in the Nextflow platform (https://github.com/lehtiolab/ddamsproteomics, vs2.7) built using the workflow tool Nextflow (v20.01.0). MSGF+ settings included precursor mass tolerance of 10 ppm, fully-tryptic peptides, maximum peptide length of 50 amino acids and a maximum charge of 6. Fixed modifications were TMTpro 16 plex on lysines and peptide N-termini, and carbamidomethylation on cysteine residues, a variable modification was used for oxidation on methionine residues. Quantification of TMTpro 16 plex reporter ions was done using OpenMS project's IsobaricAnalyzer (v2.0)[83]. PSMs found at 1% FDR were used to infer gene identities. Protein quantification by TMTpro 16 plex reporter ions was calculated using TMT PSM ratios to the entire sample set (all 16 TMT-channels) and normalized to the sample median. The median PSM TMT reporter ratio from peptides unique to a gene symbol was used for quantification. Protein false discovery rates were calculated using the picked-FDR method using gene symbols as protein groups and limited to 1% FDR[84]. Mass spectrometry-based proteomics analysis was performed by the Clinical Proteomics Mass Spectrometry facility, Karolinska Institutet/Karolinska University Hospital/Science for Life Laboratory, Solna, Sweden.

## SCAN-B TNBC validation cohort

To validate DNA methylation patterns and subgroups derived from the discovery SCAN-B cohort, a set of 136 non-overlapping SCAN-B TNBC

tumors part of the cohort reported by Staaf et al.[70] was analyzed using MethylationEPIC v1.0 or v2.0 BeadChip according to manufacturer's instructions by the SNP&SEQ Technology Platform in Uppsala, Sweden (www.genotyping.se). This patient set is hereon referred to as the SCAN-B validation cohort. Matched clinicopathological and RNAseq FPKM data, including PAM50 subtypes, were obtained from[70]. Beta values, representing the level of methylation, were computed in a sample-by-sample context using the minfi R package (v1.44) function preprocessNoob() and Infinium probe normalized using the approach described by ref. 64. Due to the lack of whole genome sequencing data for the validation cohort we used the recently developed PureBeta pipeline[32] to estimate tumor purity directly from DNA methylation data that could then be used to correct beta values according to the Staaf and Aine[31] method. Specifically, we first used the purity_estimation() function to estimate tumor purity using the provided breast cancer models in PureBeta. Then, we corrected the beta values for each CpG and tumor using the reference_based_beta_correction() function. For the correction we used the CpG regression models derived in the 235-sample SCAN-B discovery cohort setting the refitting option to false. After all processing steps, 701304 CpGs remained for analysis. Patient specific data for the validation cohort is available in Supplementary Data 1b. All reported analyses of the validation cohort were performed as described for the discovery cohort.

## Multi-omics TCGA general breast cancer validation cohort
Matched DNA methylation data from Illumina 450 K bead arrays, RNAseq FPKM data and whole exome sequencing (WES) somatic mutation data for 645 breast cancers of all clinical subgroups from the TCGA consortium were obtained and processed as described in Staaf and Aine[31]. For DNA methylation data, 381355 CpGs remained after annotation mapping, filtering and EPIC platform matching. TCGA tumors were PAM50 subtyped using the nearest centroid classification approach described by Staaf et al.[70], using the same reference sets and code to ensure single sample classification.

## Non-overlapping general breast cancer SCAN-B RNAseq cohort
RNAseq FPKM data from 6233 primary tumors were collected as described by Veerla et al.[85] from data deposited by Staaf et al.[70]. Of the 6233 tumors, 6009 did not overlap with the 235-sample SCAN-B TNBC discovery cohort and were used to contrast gene networks in a general breast cancer population. All patient and tumor annotations were taken from Staaf et al.[70]. RNAseq processing is identical for the 6009 tumors and the SCAN-B TNBC discovery and validation cohorts.

## Normal breast tissue and sorted immune cell epigenetic cohorts
Illumina 450 K DNA methylation profiles for 96 normal breast tissue samples obtained from mastectomies, breast reductions, and prophylactic tissue were obtained from Gene Expression Omnibus (GEO) under accession number GSE67919 as processed beta values[86]. Illumina 450 K DNA methylation profiles ($n = 60$) for 10 different blood cell types/fractions were obtained from GEO under accession number GSE35069 as processed beta values[87].

## Single Cell Portal web analysis
The Broad Single Cell Portal (https://singlecell.broadinstitute.org/single_cell)[37] was used to investigate expression of genes in single cell RNAseq (scRNAseq) data. As basis for the investigation, we used the scRNAseq TNBC study by Wu et al.[38] as available through the portal. Gene expression was explored using the Heatmap function with the 'celltype_final' annotations provided and all other parameters as default. The portal was accessed August 16 2024.

## Genes And Metabolism (GAM) webservice analysis
To analyze metabolic pathways connected to a gene list, we used the Genes And Metabolism (GAM) webservice (https://artyomovlab.wustl.

edu/shiny/gatom/) described by Sergushichev et al.[36]. We used the 'Shiny GATOM: integrated analysis of genes and metabolites' application and created lists of genes to be used as input as required by the application for 'Differential expression for genes'. The webservice was accessed August 21 2024. The following parameters were used for the analysis of nonBasal gene networks: i) Network type: KEGG, ii) Network topology: atoms, iii) Scoring parameter for genes: Number of positive genes, iv) Number of positive genes: 50, v) Solve for optimality: check, vi) Add highly expressed genes: ticked, and vii) Actions with atoms: connect atoms inside metabolite.

## Public cell line data
Genome coordinate conversion to hg19 for EPIC probes was carried out using the V1-version of EPIC.hg19.manifest.tsv.gz available through the Zhou-lab GitHub page (https://github.com/zhou-lab) and used instead of hg38 annotations where needed. H3K27Ac bigWig tracks for CD4 (ERS208296 and MS028001), CD8 (ERS206373 and ERS358697), Macrophage (ERS206577 and ERS791875) and Breast luminal progenitor cells (CEMT0191 and CEMT0195) were downloaded (2024-02-28) from the International Human Epigenome Consortium (IHEC) data portal (https://epigenomesportal.ca) and processed using rtracklayer (v1.62.0)[88] and GenomicRanges (v1.54.1)[89] in R.

RNAseq data and matched Illumina 450 K DNA methylation profiles for 34 TNBC cell lines were obtained from the study by Jovanovic et al.[90] as processed FPKM expression values and beta values ($n = 372551$ CpGs) (GEO, GSE202770), respectively. From the same study we collected single cell RNAseq raw count data for 19 TNBC cell lines (GEO, GSE202771)[90]. Data for four of the eight TNBC cell lines included in our study (HCC1937, MDA-MB-231, BT-549, and SUM185PE) were available among these 19 cell lines. Single cell RNAseq data were read and processed using seurat (v5)[91] in R. A seurat object was created using the function CreateSeuratObject and parameters "min.cells=10", "min.features"=200. Mitochondrial reads were quantified using the function PercentageFeatureSet and "pattern = '^MT-'". A subset of the seurat object was created using the function subset and parameters "subset = nFeature_RNA >= 750 & nFeature_RNA <= 6000 & percent_mt <= 20 & nCount_RNA >= 5000". The subset was then processed using the functions "NormalizeData", "FindVariableFeatures" with parameters "selection.method = vst" and "nfeatures = 2000", ScaleData with all genes as features, FindNeighbors with "dims = 1:30", FindClusters with resolution 0.1, and RunUMAP with "dims = 1:30". The functions DimPlot and FeaturePlot were used for visualization. The function FetchData was used to obtain gene-level counts for each cell. Values were further dichotomized using the cutoff value of ≥1 prior to visualization using tidyverse (v2.0.0) in R.

Lastly, a cancer cell line set comprising 869 cell lines of different malignancies with matched processed Affymetrix mRNA expression and Illumina 450 K DNA methylation profiles were collected from Iorio et al.[92] (GEO, GSE68379).

## Survival analyses
Survival analyses were performed in R (v4.2.2) using the survival package (v3.5.8) with distant relapse-free interval (DRFI) as endpoint defined according to the STEEP guidelines[93]. Survival curves were compared using Kaplan-Meier estimates and the log-rank test. Full details on the exclusion criteria for outcome analysis and individual patient treatment data are available in Staaf et al.[6].

## Statistics & reproducibility
Calculated $p$-values from statistical tests are two-sided if not explicitly stated otherwise. If not stated otherwise, boxplot elements correspond to: (i) center line = median, (ii) box limits = upper and lower quartiles, (iii) whiskers = 1.5x interquartile range. No statistical method was used to predetermine sample size. No data (patients) were excluded from the analyses. The experiments were not randomized.

The Investigators were not blinded to allocation during experiments and outcome assessment. Patient sex was self-reported and not considered in the study design, as all patients were female.

## Reporting summary

Further information on research design is available in the Nature Portfolio Reporting Summary linked to this article.

## Data availability

The previously reported SCAN-B WGS data from Staaf et al.[6] used in this study are available from [https://data.mendeley.com/datasets/2mn4ctdpxp/3]. The previously reported SCAN-B RNA-sequencing data from Staaf et al.[70] used in this study are available from [https://data.mendeley.com/datasets/yzxtxn4nmd/3]. The DNA methylation data generated in this study for the SCAN-B discovery cohort have been deposited in the Gene Expression Omnibus database under accession code GSE148748 and GSE148906. The DNA methylation data generated in this study for the SCAN-B validation cohort have been deposited in the Gene Expression Omnibus database under accession code GSE290981. Raw RNA sequencing data and ATAC-seq data generated in this study for eight TNBC cell lines are available through the SRA archive at NCBI under BioProject accession PRJNA1189708 and study SRP547133. The DNA methylation data generated in this study for the eight TNBC cell lines have been deposited in the Gene Expression Omnibus database under accession code GSE282347. Raw and normalized proteomic data generated in this study for eight TNBC cell lines are available in Supplementary Data 5, and mass spectrometry data have been deposited to the ProteomeXchange Consortium via the JPOST partner repository with the data set identifier PXD058472. The previously reported breast cancer TCGA data used in this study is available from the GDC data portal [https://portal.gdc.cancer.gov]. The previously reported Iorio et al.[92] cancer cell line data used in this study are available from [https://www.ncbi.nlm.nih.gov/geo/query/acc.cgi?acc=GSE68379]. The previously reported normal breast tissue DNA methylation data from Hair et al.[86] used in this study are available from [https://www.ncbi.nlm.nih.gov/geo/query/acc.cgi?acc=GSE67919]. The previously reported gene expression and DNA methylation data from Jovanovic et al.[90] used in this study are available from [https://www.ncbi.nlm.nih.gov/geo/query/acc.cgi?acc=GSE202770]. The previously reported single cell RNAseq data for TNBC cell lines from Jovanovic et al.[90] used in this study are available from [https://www.ncbi.nlm.nih.gov/geo/query/acc.cgi?acc=GSE202771]. The previously reported sorted immune cell DNA methylation data from Reinius et al.[87] used in this study are available from [https://www.ncbi.nlm.nih.gov/geo/query/acc.cgi?acc=GSE35069]. The remaining data are available within the Article, Supplementary Information or Source Data file. Source data are provided with this paper.

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

## Acknowledgements

The authors would like to acknowledge Rikard Mohlin at RICA pathology for support in pathological evaluations, the Center for Translational Genomics, Lund University and Clinical Genomics Lund, SciLifeLab for support with DNA methylation analyses, all patients and clinicians participating in the SCAN-B study, personnel at the central SCAN-B laboratory at the Division of Oncology, Department of Clinical Sciences Lund, Lund University, the Swedish national breast cancer quality registry (NKBC), Regional Cancer Center South, RBC Syd, and the South Sweden Breast Cancer Group (SSBCG). Methylation profiling of cell lines and the SCAN-B validation cohort was performed by the SNP&SEQ Technology Platform in Uppsala (www.genotyping.se). The facility is part of the National Genomics Infrastructure supported by the Swedish Research Council for Infrastructures and Science for Life Laboratory, Sweden. Financial support for this study was provided by the Swedish Cancer Society (CAN 2021/1407 JS, 2024/3591 J.S.), the Mrs Berta Kamprad Foundation (FBKS–2020-5 JS and FBKS-2024-14 J.S.), the Swedish Research Council (2021-01800 J.S.), the Mats Paulsson Foundation (Å.B.), the Cancera Foundation (Å.B.), the BCF-VÖS Foundation (J.S.), the National Society of Breast Cancer Associations in Sweden (J.S.), and Swedish governmental funding (A.L.F., grant 2022/0021 J.S.). The funders had no role in study design, data collection and analysis, decision to publish, or preparation of the paper.

## Author contributions

Conception and design: J.S. and M.A. Collection and assembly of data: J.S., J.V.C., J.H., D.F.N., A.K., F.R., E.A., M.A. Provision of study material or patients: Å.B., J.V.C. Data analysis and interpretation: J.S., D.F.N., S.V., E.A., M.A., H.J.J. Financial support: J.S. Administrative support: J.S., J.V.C. Manuscript writing: All authors Final approval of manuscript: All authors Agree to be accountable for all aspects of the work: All authors.

## Funding

## Competing interests

The authors declare no competing interests.
