## [Transparent Peer Review file · Nature Communications]

The DNA methylation landscape of primary triple-negative breast cancer

Corresponding Author: Dr Johan Staaf

Version 0:

Reviewer comments:

Reviewer #1

(Remarks to the Author)

Aine and colleagues explore the DNA methylation patterns in triple negative breast cancer from the SCAN-B cohort. They identified two major groups with correlation to expression-based subtypes of Basal and Non-basal and further separated into additional 3 basal groups and 2 non-basal groups. These were validated in a separate validation set of SCAN-B samples as well as in TCGA. By correcting for purity, they were able to identify stronger DNA methylation signatures for TNBC groups. The non-basal group was less methylated at steroid response transcription factors and correlated with the LAR TNBC subtype. The basal3 group showing hypomethylation of CD274 and corresponding increase in expression. Would any of your data allow you to identify additional potential hits that explain the increased immune infiltration in your Basal3? And would this help determine if the hypomethylation of CD274 is in response to the immune infiltration or somehow is the cause?

I think much of this text could be streamlined. I understand the desire to show how you came to the set of DNA methylation features and the purity filter, but it distracted from the main goal of the paper. I think potentially moving more of the details to supplement but retaining the main take aways would help strengthen the reading of the manuscript. The validation data results section seemed to be quite a lot of repeating the prior text and could be streamlined as well. the FOXC1 and FOXA1 results were also shown in TCGA ATAC-seq paper (Corces et al) supplementary figure S6. However, their focus was PAM50 subtypes of all clinical subtypes and had not looked at the TNBC non-basals to show that pattern was more like the luminals.

Reviewer #2

(Remarks to the Author)

Overall impact:

This manuscript presents a compelling and comprehensive analysis of the epigenetic landscape in triple-negative breast cancer (TNBC). By integrating data from large patient cohorts, multi-omics approaches, and advanced computational techniques, the study links epigenetic patterns with clinicopathological and high-dimensional omics data to address intrinsic subtypes and the potential epigenetic regulation of key transcriptional drivers in TNBC. By analysing pure tumor methylome profiles and using unsupervised NMF clustering based on CpG methylation context, the authors classified TNBC into Basal and nonBasal subtypes. The Basal group is further classified into three subgroups: Basal3, characterized by the highest tumor-infiltrating lymphocytes (TILs) and immune-related gene enrichment; Basal2, with low TIL infiltration and the highest expression of stromal metagenes; and Basal1, with an intermediate TIL infiltration and no strong metagene enrichment. The nonBasal groups are classified into nonBasal1 containing a mix of different TNBCtype and PAM50 subtypes and nonBasal2 enriched with the TNBCtype LAR subtype and metabolic gene networks. Both nonBasal subgroups exhibit immune-cold tumor microenvironments (TME). This study represents the first tumor-focused DNA methylation analysis of TNBC, highlighting the significance of CpG context and the TME in interpreting methylation data from solid cancers. The findings provide a novel molecular perspective on TNBC, emphasizing the role of epigenetic alterations in tumorigenesis. Specifically, the manuscript sheds lights on the association of transcription factors (FOXA1, FOXC1, and SLC44A4), metabolic networks, and immune checkpoints (PD-L1/PD-L2) with epigenetic regulation and immune evasion in Basal and

nonBasal TNBC subtypes. While this study offers critical insights into TNBC biology, some aspects of methodology and presentation require further clarification and elaboration.

The manuscript is acceptable for publication after addressing the following concerns for improvement.

Major Comments

1. The manuscript contains numerous grammatical errors and would benefit from thorough editing by a professional editor proficient in English. Additionally, simplifying the technical language would enhance its readability and accessibility for a broader audience.
2. The abstract covers the study's novelty and emphasizes the potential clinical implications of epigenetic subtyping. However, the results section in the abstract should explicitly mention the identified epigenetic subtypes for clarity. Additionally, the conclusion section in the abstract should emphasize the therapeutic impact of the findings. Simplifying technical language in the abstract would further enhance accessibility for a broader readership.
3. In the second paragraph of the introduction section, it is recommended to include a discussion of previous DNA methylation subtyping efforts and their limitations. This will help contextualize the current study and underscore its novelty and contributions to the field.
4. The introduction could be revised to include key background information on methylation dynamics, providing a clearer foundation for readers who may be less familiar with the topic.
5. The authors utilized purity-adjusted beta values for CpG analysis. However, since WGS and ASCAT inherently assess tumor purity, this approach may limit the detection of all tumor cell fractions. The reliability and definitiveness of purity estimates using this method are therefore questionable. The authors should perform a quality check to compare purity-adjusted versus non-adjusted clusters to validate their approach. Additionally, the discussion section should emphasize the importance of purity adjustment by addressing the limitations of prior epigenetic subtyping studies on bulk tumors. Comparing the variability observed in this study with findings from previous studies would further contextualize the results and highlight the advantages of the current methodology.
6. The patient cohort is primarily derived from the SCAN-B study, which may limit the generalizability of the findings. This limitation should be acknowledged and discussed in the discussion section to provide a balanced interpretation of the study's implications.
7. The classification and subclassification of TNBC into Basal and nonBasal subtypes, and further into six subgroups, are well explained. However, the clinical significance of these subgroups is not addressed. Including a discussion of the potential clinical implications of these subgroups would strengthen the argument and provide a more comprehensive understanding of the study's relevance in the discussion section.
8. Most of the findings related to FOXA1 methylation are currently presented in the supplementary material. To enhance the manuscript's impact, it would be beneficial to integrate these findings into the main text or include them in a figure. This would highlight the significance of these results more effectively.
9. The sections assessing overall breast cancer and other cancer-related analyses should be removed, as they are not directly relevant to TNBC. Focusing solely on TNBC will strengthen the manuscript's focus and clarity.
10. In the discussion section, the authors claim that their subtyping method offers an advantage over previous transcriptional and genetic classifications. However, they do not directly compare their classification in terms of therapeutic relevance. To strengthen this argument, the authors should address how their subtyping could inform treatment decisions or improve therapeutic outcomes. A direct comparison with existing subtyping systems and highlighting how their classification may offer more precise insights into treatment response, therapeutic targets, or patient prognosis would provide a more compelling rationale for the claimed advantage.
11. The authors argue that epigenetic regulation could be a potential therapeutic target area. However, they do not provide sufficient literature supports to demonstrate that these novel methylation changes are not merely correlated but actually serve as valuable therapeutic targets. Additionally, while the authors suggest that their epigenetic classification could be therapeutically valuable, the argument appears speculative without concrete evidence linking therapeutic responses to variations in the methylome. To make this claim more compelling, the authors should provide evidence or examples from clinical or preclinical studies showing how these methylation changes might influence treatment outcomes or response to specific therapies.

Minor Comments

1. Abbreviations should be defined when mentioned for the first time in the manuscript and then used consistently throughout the rest of the manuscript.
2. The text is highly technical and densely packed with information, which may be challenging for non-specialists to follow. To improve accessibility, the authors should consider simplifying the language and breaking down complex concepts into clearer, more digestible explanations to make the findings more accessible to a broader audience, including researchers from other fields or clinicians with limited expertise in this specific area.
3. Please use "subgroup" or "subtypes" consistently throughout the manuscript, as "subbranches" is not an appropriate term in this context. This will ensure clarity and consistency in the terminology.
4. On page 2, lines 24-25, the statement "Triple-negative breast cancer (TNBC) is a clinically challenging entity that constitutes 10% breast cancer" is inaccurate. TNBC actually accounts for approximately 15-20% of breast cancers.
5. The authors used distal ATAC peaks to infer regulatory potential but did not provide a clear rationale for this approach. It would strengthen the manuscript if the authors explained why distal ATAC peaks were chosen as a marker of regulatory potential and how they contribute to the overall findings.
6. For the external datasets used in the study, it is unclear how the authors ensured comparability between data generated from different platforms, such as the EPIC and 450k arrays. The authors should clarify the normalization or harmonization methods applied to account for platform-specific differences and ensure consistency across datasets.
7. Eight TNBC cell lines representing different TNBCtype mRNA subtypes (HCC2157: BL1, HCC1599: BL1, HCC1937: BL1, BT-549: M, MDA-MB-231: MSL, MDA-MB-468: BL1, SUM185PE: LAR, and MDA-MB-453: LAR) were used in this

study. Please explain why did not include BL2 and IM cell lines.

8. The authors chose the 1000, 2000, and 5000 most variable CpGs for clustering but did not provide a rationale for selecting these specific numbers. It would strengthen the methodology to explain why these particular thresholds were chosen.

9. In the result section for Fig 2a, on page 17, lines 437-457, the characterization of CpG cluster 2 is missing. While the authors provide a detailed characterization of CpG clusters (hyper- or hypomethylation) in the nonBasal subtype, they should also discuss the CpG clusters in the Basal subtype. Including this analysis would provide a more balanced and complete view of the methylation patterns across both Basal and nonBasal subtypes.

10. The figures are not very non-specialist friendly, and some figures, such as Fig 5, lack clear annotations.

11. On page 4, lines 63-65, the statement "Triple-negative breast cancer (TNBC) is negative for the three current treatment-predictive markers (estrogen, ER, and progesterone, PR, receptors and ERBB2/HER2 gene amplification) and constitutes ~10% of patients with primary disease" is inaccurate and needs to be rephrased. ER, PR, and HER2 are primarily used as treatment targets rather than predictive biomarkers. Additionally, TNBC accounts for approximately 15-20% of breast cancers, not ~10%.

12. The citation format for references should be revised to ensure consistency throughout the manuscript. The authors should follow the appropriate style guide and ensure that all references are cited in the same manner.

13. On page 3, line 74, "androgen (AR)" should be "androgen receptor (AR)".

14. The manuscript inconsistently uses the terms "mRNA subtypes" and "transcriptional subtypes." To improve clarity and consistency, the authors should choose one term and use it consistently throughout the manuscript.

15. On page 6, line 144, where it says "merged with data from 14," and on line 148, where it says "with additional details provided in 4," and so on, the authors should provide context before citing the references.

16. On page 17, lines 450-452, the authors mention mapping the 5000 distal-ATAC CpGs to the enhancer-linked H3K27 acetylation (H3K27Ac) mark in breast luminal progenitor cells and three immune cell types using publicly available data. The authors should provide a rationale for why they specifically focused on these particular cell types: breast luminal progenitor cells, CD4, CD8, and macrophages. Given that this manuscript is focused on TNBC classification, it would also be valuable to explain why the study is limited to these specific immune cell types rather than including all immune cell types, which could provide a more comprehensive view of the immune landscape in TNBC.

Reviewer #3

(Remarks to the Author)

Reviewer #4

(Remarks to the Author)

In this study, Aine et al. stratified TNBC patients samples into epigenetic subgroups following the adjustment of Illumina EPIC beadchips-based global DNA methylation profiles for tumor cell content and the accounting for genomic context of CpG methylation. Epigenetic subgroups were validated in several independent cohorts and extensively characterized through integration of whole genome sequencing, RNA-sequencing, immunohistochemistry, and clinical data. They identified the epigenetic regulation of key steroid response genes and developmental transcription factors as important drivers of this subclassification. They further divided the Basal and non-Basal subgroups into several subbranches, identifying a distinct Luminal Androgen Receptor (LAR) phenotype among the non-Basal subgroups that is characterized by the putative epigenetic regulation of a specific metabolic gene network. Basal subbranches were characterized by differences in immunogenicity and immune evasion strategies including a subgroup with epigenetically-regulated expression of the immune checkpoint gene PD-L1. They conclude that the subdivision of TNBC based on pure tumor methylomes and genomic context of CpG methylation transcends genetic and transcriptional TNBC subtypes by highlighting widely differing immunological microenvironments and putative epigenetically mediated immune evasion strategies.

This study is very interesting and timely and features a wealth of data and well and thoroughly executed data analyses. However, some concerns remain, especially surrounding the leap of knowledge of their epigenetic TNBC subgrouping with respect to gene expression-based TNBC subtyping.

Major comments:

1. The authors mention tumor heterogeneity and tumor purity as source of bias and as an underlying rationale for adjusting DNA methylation data for tumor cell content, irrespective of the well-established role and clinical importance of the TME/TIME. Could the authors better outline their rationale for focusing on TNBC intrinsic subtypes? Further related to this question, the authors state that "Importantly, purity-adjusted CpG data showed consistently higher accuracy for this split compared with conventional unadjusted CpG data (Supplementary Figure S4c), supporting the relevance of using DNA methylation levels adjusted for tumor cell content." We encourage the authors to re-phrase this sentence as both approaches, i.e. focusing on tumour cells only and taking the tumor as a whole, are valid approaches.

2. The authors suggest the use of distal regulatory CpGs for investigating TNBC DNA methylation phenotypes, yet their

results seem to indicate the importance of other classes of CpGs for the distinction between Basal and nonBasal subgroups:

- o CpGs with higher methylation in nonBasal samples were more frequently in high-density CpG island regions
- o CpG shore/CGI methylation for FOXA1 that aligned almost perfectly with the Basal (hypermethylated shore/CGI CpGs) and nonBasal (hypomethylated shore/CGI CpGs) epigenetic stratification

Could the authors comment on that and also advise on how to take into account the genomic CpG context in future studies based on the finding of this study?

3. The authors thoroughly characterize their epigenetic subgroups with respect to gene expression-derived TNBC subtypes, but could the authors better explain what the epigenetic subtyping approach of this study adds on top of what is already known based on gene expression and how this is clinically relevant?

4. It would be interesting to know how the epigenetic TNBC subgroups of this study compare with those of previous studies. Are their overlaps between the different epigenetically-defined TNBC subtypes?

5. I wonder whether the results described in the following statement “Gene set analysis using “hallmark” signatures of differential genes in Basal3 tumors confirmed a strong association with multiple immune pathways (Supplementary Table S3b).” are of intrinsic or extrinsic nature. Could the authors redo this analysis with a purity for Basal 3 that matches the purity of Basal 1 & 2? The authors could use regression to suppress the cell composition in their expression data, like they did with their DNA methylation data. Comparing adjusted and unadjusted gene expression changes could allow to distinguish between intrinsic and extrinsic contributions.

6. Could the authors provide a statistical test, e.g. Chi-squared or hypergeometrical, for the following statements:

o “Sample cluster agreement between solutions was high, and there was a 57% overlap of...”

o “...57% overlap of differentially methylated CpGs between solutions (n=1087 CpGs for k=2, n=2845 CpGs for k=3), indicating that the same biology was captured (Supplementary Figure S7a).”

o “Class agreement was lower between the two different NMF approaches compared to the main NMF split, indicating that subdivision of the Basal branch may be less distinct in a sample size-limited validation cohort (Supplementary Figure S6f).” Please provide a Chi-squared test comparing de novo and directed classification to statistically confirm the validation.

o “Class agreement was high between the two approaches (83%) (Supplementary Figure S6j).”

7. Related to the statement “Except for perhaps the LAR subtype, our study provides less support for the latter as illustrated by, e.g., stratified expression pattern of FOXA1 and an extensive “one versus rest” (e.g., BL1 vs non-BL1) differential methylation analysis of TNBCtype subtypes in all tumors, Basal tumors, and nonBasal tumors only (Supplementary Figure S9). Our observation is not surprising as mRNA bulk analyses merge the expression patterns of all cells analyzed.”, could the authors use purity-adjusted RNA expression data to overcome and then compare with DNA methylation?

Minor comments:

1. The authors state “Importantly, purity-adjusted CpG data showed consistently higher accuracy for this split compared with conventional unadjusted CpG data (Supplementary Figure S4c)”, however, some CpGs (e.g. associated w/ CGI, promoter or HCP) showed higher accuracy when not adjusted. Could the authors comment and adjust their statement?

2. The authors state “PD-L1 was assessed immunohistochemically (...) in formalin-fixed, paraffin-embedded tumor samples in a tissue microarray (TMA) where each sample was represented by two TMA cores (...) The TMA core with the highest score was set as the score for the respective tumor.” Why was this approach chosen over calculating the average of the two cores? Could the authors provide a reference for their chosen approach?

3. Supplementary Table S1a: Could the authors describe in the methods section how TNBC were defined, i.e. the method of measurement and the threshold applied for ER, PR and Her2. In the table, there is no information regarding the PR and Her2 measurements, only the “ERperc_group” showing that all samples were < 10%. There is no explanation of the method used to measure ER expression. Was it IHC?

4. The authors state in the methods section “Basic processing of beadchip data was performed as described 25.” We believe the reader would benefit from a more detailed description of the preprocessing of the discovery cohort similar to the description of the preprocessing of the validation cohort in the Supplemental Methods. Notably, for the validation cohort they used “preprocessingNoob” while ref 25, which was used for the discovery cohort, used “preprocessingFunnorm”. Could the authors clarify this difference?

5. The authors state “This included assigning CpGs to a gene-centric context defined as (...) “proximal” (+/- 5 kbp centered on but excluding TSS) (...)” Did you mean “excluding promoter” instead of “excluding TSS” as TSS is just a single base?

6. Related to the following statement “Based on this, we proceeded to characterize the three-group NMF solution which divided the Basal branch into a group with high immune cell infiltration (IM positivity, Basal3)...”, the authors state in the Suppl. Methods part that IM positivity is defined using the score Lehman for IM subtype with a cut-off at 0.17. Can the authors explain why 0.17?

7. Could the authors clarify how they obtained the 14464 pairwise correlations in the following statement “Out of 14464 pairwise correlations, 142 were significant in both data sets”?

8. For the characterization of their epigenetic TNBC subgroups, the authors rely repeatably on metagene signatures from a publication from 2012 (Fredlund et al.). Could more recently published signatures be considered?

9. Could the authors re-work the following statement as almost all nonBasal are HRD-negative “Similarly, the genetic phenotypes of HRD-positive and HRD-negative tumors shaped by defective DNA repair do not match intrinsic epigenetic subtypes as they fall in all subgroups defined in this work, although less frequently in nonBasal tumors.”?

Version 1:

Reviewer comments:

Reviewer #1

(Remarks to the Author)

The authors have addressed my comments.

Reviewer #2

(Remarks to the Author)

The authors have thoroughly addressed all comments raised by the Reviewers and have revised the manuscript accordingly. They have enhanced relevant sections with additional details, provided further clarifications, incorporated appropriate citations, and included several new figures to strengthen the manuscript. The manuscript is acceptable for publication as an original research article in Nature Communications after implementing the following corrections. References should be cited directly following a statement without using “e.g.,” or “like” before them. For example, citations such as “e.g., 6, 7, 15, 16” and “like 18, 19, 20” should be formatted as “6, 7, 15, 16” and “18, 19, 20”, respectively. Furthermore, formatting adjustments are needed for statistical reporting. For example, in the sentence: “Sample cluster agreement between 350 solutions was high (two-sided Chi-square test, $p < 2e-16$), and there was a 57% overlap of differentially methylated CpGs identified between groups using the different solutions (n=1087 CpGs for two-group, n=2845 CpGs for three-group), indicating that the same biology was captured (Supplementary Fig. S5a).” The following changes should be made: insert a space between “p” and “<” and between “<” and “2e-16”. Additionally, p should be italicized. Insert a space between “n” and “=” and between “=” and the numbers, so “n=1087” should be formatted as “n = 1087”. These formatting adjustments should be consistently applied throughout the manuscript.

Reviewer #3

(Remarks to the Author)

Reviewer #4

(Remarks to the Author)

The revision of “The DNA methylation landscape of primary triple-negative breast cancer”, by Aine et al., led to improvements of the manuscript. We appreciate the efforts made by Aine and colleagues to reply to all of our comments and to follow most of our suggestions. Overall, the revised manuscript appears more clear, we have just one remaining suggestion:

To address our major comment #2, Aine et al. generated crucial data and provided important explanations. While we agree that including all of this would make the manuscript too long and difficult to read, we encourage them to incorporate some of the data and key explanations. This would be beneficial for future readers who may have similar questions.

Detailed point-by-point response to the reviewers' comments on the manuscript NCOMMS-24-73463-T

First, we would like to thank all reviewers for their constructive criticism and suggestions on how to improve and further clarify this manuscript. We are grateful for the feedback and comments, which have helped us to further improve and strengthen our manuscript.

Based on the reviewers' comments, we have revised the manuscript to address all their concerns, added more detail to the relevant sections of the manuscript, further clarifications, added associated citations to reference literature, and included several new figures.

Please find below the detailed point-by-point responses to the reviewers' comments. Reviewers' comments are presented **in bold** and our responses are shown in regular text. Textual alterations made to the revised version of the manuscript are shown in *italics*, with underlined text showing modifications in existing text when deemed appropriate. The text referenced from other studies used only in this rebuttal is shown in a different font type if present. All textual alterations are present in a marked-up version of the manuscript according to journal instructions. Please note that the actual reference number for a specific literature reference may differ between the reference list in this document and the revised manuscript version that has been resubmitted. Any page numbers refer to the revised manuscript (clean file).

We would like to draw the attention of all reviewers to that the layout of the manuscript has been changed to fit the Nature Communications format, with the major change that Methods appears after the Discussion. Moreover, based on allowance of more display items we have added 2 new figures, thereby changing the original Figure and Supplementary Figure order as outlined below:

Figure order:

New Figure 1: based on the previous Supplementary Figure S1

New Figure 2: previous Figure 1 New

Figure 3: previous Figure 2 New Figure

4: previous Figure 3 New Figure 5:

previous Figure 4 New Figure 6: previous

Figure 5 New Figure 7: previous Figure 6

New Figure 8: Data from previous

Supplementary Figure S6 (Validation)

Supplementary Figure order:

New Supplementary Figure S1: previous Figure S2

New Supplementary Figure S2: previous Figure S3

New Supplementary Figure S3: previous Figure S4

New Supplementary Figure S4: previous Figure S5

New Supplementary Figure S5: previous Figure S7

New Supplementary Figure S6: previous Figure S8

New Supplementary Figure S7: previous Figure S6

New Supplementary Figure S8: previous Figure S9

Moreover, a summary cohort table (Table 1) of the SCAN-B discovery and validation cohorts has also been added to the revised manuscript.

Reviewer #1, expertise in breast cancer multi-omics and classification (Remarks to the Author):

Aine and colleagues explore the DNA methylation patterns in triple negative breast cancer from the SCAN-B cohort. They identified two major groups with correlation to expression-based subtypes of Basal and Non-basal and further separated into additional 3 basal groups and 2 non-basal groups. These were validated in a separate validation set of SCAN-B samples as well as in TCGA. By correcting for purity, they were able to identify stronger DNA methylation signatures for TNBC groups. The non-basal group was less methylated at steroid response transcription factors and correlated with the LAR TNBC subtype. The basal3 group showing hypomethylation of CD274 and corresponding increase in expression.

Would any of your data allow you to identify additional potential hits that explain the increased immune infiltration in your Basal3? And would this help determine if the hypomethylation of CD274 is in response to the immune infiltration or somehow is the cause?

Response:

This is a very interesting and important remark raised by the reviewer and does to some extent touch upon a concept of epigenetic immunoediting as a potential immune evasion mechanism (recently proposed in glioblastoma¹). Even the very basic question of strong co-expression of PD-L1 and PD-L2 is interesting, who regulates whom? Whether the hypomethylation is in response to or the cause of the immune infiltration is, however, very challenging for us to speculate about given the data at hand. Unfortunately, we do not have longitudinal tumor specimens taken during treatment (e.g. neoadjuvant treatment) that could indicate if the hypomethylation pattern is plastic. We do, however, as shown find the hypomethylation pattern present even in cell lines established decades ago without any immune cell compartment present in culture (and thus without a conceptual need to “avoid” immune cells). Considering a) the heterogeneity in PD-L1 expression in cell lines based on scRNAseq data, b) the heterogeneity in PD-L1 protein on tumor cell surface (by IHC) in Basal3 tumors despite a nearly complete hypomethylation state of the highlighted CpGs, and c) the semi-hypermethylation state of the presented CpGs in Basal1 and Basal2 tumors (i.e. the corrected DNA methylation levels are not 1 = fully hypermethylated) it would appear that the answer will be complex and probably related to the composition and interaction of different tumor cell populations.

In response to the remark, we have rephrased the following text in the Discussion to reference a proposed concept of epigenetic immunoediting, as well as to include the excellent question posed by the reviewer regarding cause or consequence:

Importantly, the current study cannot address the intriguing questions of why and when both broader and highly specific DNA methylation patterns, like the one revealed for FOXA1 and FOXC1, emerge in Basal and nonBasal tumors. Currently we are also unable to answer whether the observed DNA methylation patterns that define the immune-warm Basal3 subgroup represent epigenetic immunoediting as proposed in glioblastoma by Gangoso et al.¹, and whether this occurs in response to the immune infiltration or is naturally occurring and selected for.

I think much of this text could be streamlined. I understand the desire to show how you came to the set of DNA methylation features and the purity filter, but it distracted from the main goal of the paper. I think potentially moving more of the details to supplement but retaining the main take aways would help strengthen the reading of the manuscript.

Response:

We appreciate the reviewer's suggestion about streamlining an already extensive manuscript to improve readability. Other reviewers have, in contrast, requested us to provide even more background information about previous studies, methylation dynamics, and text about the rationales for specific choices made for the unsupervised clustering to support less experienced readers (see, e.g., Reviewer 2 below). As a compromise we have opted to provide more supporting text as requested by mainly reviewer 2, despite that this increases the volume of the manuscript.

The validation data results section seemed to be quite a lot of repeating the prior text and could be streamlined as well.

Response:

We appreciate the reviewer's comment about possible improvements to the manuscript. The current layout of the validation was to validate each of the main divisions directly after the discovery section (e.g. the Basal/nonBasal division). In response to the reviewer's request, we merged the three validation sections into one at the end of the Results section, creating a new main Figure 8, and made changes to the text so it would be less repetitive and easier to digest.

The FOXC1 and FOXA1 results were also shown in TCGA ATAC-seq paper (Corces et al) supplementary figure S6. However, their focus was PAM50 subtypes of all clinical subtypes and had not looked at the TNBC non-basals to show that pattern was more like the luminals.

Response:

In the revised manuscript we have acknowledged this fact in the discussion section by referencing the Corces et al. study. We thank the reviewer for pointing this out.

Reviewer #2, expertise in breast cancer and TNBC classification (Remarks to the Author):

Overall impact:

This manuscript presents a compelling and comprehensive analysis of the epigenetic landscape in triple-negative breast cancer (TNBC). By integrating data from large patient cohorts, multi-omics approaches, and advanced computational techniques, the study links epigenetic patterns with clinicopathological and high-dimensional omics data to address intrinsic subtypes and the potential epigenetic regulation of key transcriptional drivers in TNBC. By analysing pure tumor methylome profiles and using unsupervised NMF clustering based on CpG methylation context, the authors classified TNBC into Basal and nonBasal subtypes. The Basal group is further classified into three subgroups: Basal3, characterized by the highest tumor-infiltrating lymphocytes (TILs) and immune-related gene enrichment; Basal2, with low TIL infiltration and the highest expression of stromal metagenes; and Basal1, with an intermediate TIL infiltration and no strong metagene enrichment. The nonBasal groups are classified into nonBasal1 containing a mix of different TNBCtype and PAM50 subtypes and nonBasal2 enriched with the

TNBCtype LAR subtype and metabolic gene networks. Both nonBasal subgroups exhibit immune-cold tumor microenvironments (TME). This study represents the first tumor-focused DNA methylation analysis of TNBC, highlighting the significance of CpG context and the TME in interpreting methylation data from solid cancers. The findings provide a novel molecular perspective on TNBC, emphasizing the role of epigenetic alterations in tumorigenesis. Specifically, the manuscript sheds lights on the association of transcription factors (FOXA1, FOXC1, and SLC44A4), metabolic networks, and immune checkpoints (PD-L1/PD-L2) with epigenetic regulation and immune evasion in Basal and nonBasal TNBC subtypes. While this study offers critical insights into TNBC biology, some aspects of methodology and presentation require further clarification and elaboration. The manuscript is acceptable for publication after addressing the following concerns for improvement.

Major Comments

1. The manuscript contains numerous grammatical errors and would benefit from thorough editing by a professional editor proficient in English. Additionally, simplifying the technical language would enhance its readability and accessibility for a broader audience.

Response:

We agree with the reviewer that enhanced readability is of the utmost importance when aiming to reach a broader audience. We have therefore made an effort to choose simpler words when discussing our results, as well as to identify and correct any grammatical errors made previously.

2. The abstract covers the study's novelty and emphasizes the potential clinical implications of epigenetic subtyping. However, the results section in the abstract should explicitly mention the identified epigenetic subtypes for clarity. Additionally, the conclusion section in the abstract should emphasize the therapeutic impact of the findings. Simplifying technical language in the abstract would further enhance accessibility for a broader readership.

Response:

We thank the reviewer for these suggestions. To note, the abstract has been substantially reworked to fit the Nature Communications requirement (a general 150-word abstract) in the revision as the original version did not adhere to this.

3. In the second paragraph of the introduction section, it is recommended to include a discussion of previous DNA methylation subtyping efforts and their limitations. This will help contextualize the current study and underscore its novelty and contributions to the field.

Response:

We thank the reviewer for these suggestions. In response to the remark, we have further expanded the second section of the Introduction with respect to previously reported epigenetic subtyping studies by adding:

Three studies have suggested the existence of either three ^{2, 3} or four ⁴ DNA methylation subtypes in TNBC based on analysis of Illumina DNA methylation data from bulk tumor tissue. Briefly, Stirzaker et al. reported three TNBC subgroups after analyzing 73 tumors from The Cancer Genome Atlas (TCGA) project, and only differing prognoses were reported as group-specific characteristics ². Lin et al. also reported three subgroups based on 44

*analyzed tumors: one subgroup was characterized by older patients with typically low-proliferative tumors with low immune infiltration, and an overrepresentation of PIK3CA mutations; another subgroup comprised younger patients with tumors typically of higher grade, with higher levels of immune infiltration, and an overrepresentation of BRCA1/2 mutations; and the third subgroup lacked defining characteristics*³. Lastly, DiNome et al. reported four subgroups based on 79 analyzed TCGA tumors, two of which showed enrichment of two TNBC mRNA subtypes (mesenchymal and LAR), and another subgroup showed slightly more immune-warm tumors⁴. However, none of these studies provided validation of the proposed DNA methylation subtypes in independent cohorts and there has been no systematic comparison of reported subgroups.

To note, the current study's discovery cohort is 3-times larger than previous studies, and the validation cohort is approximately 2-times larger than previous studies.

4. The introduction could be revised to include key background information on methylation dynamics, providing a clearer foundation for readers who may be less familiar with the topic.

Response:

Epigenetic processes are, like most biological processes, convoluted and composed of several components that can be acting in different cells at different times. It is unclear to us exactly which "methylation dynamics" information the reviewer would like to see expanded upon in the Introduction of our manuscript. Presently, in the original manuscript there is a section about CpG context and the different ways to categorize CpGs in the genome (by gene context and CGI context) for readers that are less used to epigenetic concepts and its peculiarities. Given another comment by this reviewer (see below), we have now added additional text to this section regarding open/closed chromatin states as we make ample use of it in our work. While we acknowledge the importance of the reviewer's comment, text space is unfortunately limited in an already lengthy manuscript. Notably, another reviewer asked us to streamline the manuscript by excluding e.g. the analyses around DNA methylation features (Figure 1) which were included exactly to provide readers with a background and rationale for selecting the specific CpG context used throughout the manuscript. Figure 1 also provides background data to less experienced readers of the composition of the EPIC array and the relationship between variation in DNA methylation (representing potential biological variation) and CpG context, which is why we decided to keep this section of the Results in the revised version of the manuscript.

To strike a balance between reviewers we have in the revised manuscript added the following text to the Introduction about epigenetics and particular DNA methylation:

Histone modifications, chromatin remodeling, and DNA methylation represent fundamental processes underlying epigenetic regulation and are commonly altered during tumorigenesis (see⁵ for review). DNA methylation involves the addition of a methyl group to a cytosine base in the context of a CpG dinucleotide, catalyzed by DNMT enzymes, and can play a role in transcriptional regulation, gene imprinting, X-chromosome inactivation, suppression of repeat elements, and other aspects of gene and genome regulation.

5. The authors utilized purity-adjusted beta values for CpG analysis.

However, since WGS and ASCAT inherently assess tumor purity, this approach may limit the detection of all tumor cell fractions. The reliability and definitiveness of purity estimates using this method are therefore questionable.

Response:

ASCAT and similar methods applicable to WGS have been demonstrated to provide accurate estimates of tumor purity and ploidy at a tumor level when compared to pathologist-based estimations. Moreover, the PureBeta pipeline used for the validation cohort has been shown to estimate tumor purity estimates that very well correlate with measurements from other data sources ⁶.

We interpret this remark as that the reviewer does not feel that the current purity adjustment approach of beta values accounts for the full possible extent of tumor subclonality when present. With respect to the proposed correction method developed by us we agree on this point (Staaf and Aine et al. ⁷). However, in order to achieve an optimal correction that accounts fully for subclonality would require a new approach in identifying or modelling subclones from DNA methylation data, and would probably require carefully annotated training data, possibly derived from controlled experiments, which just is not presently available. Importantly, a limitation for the current purity adjustment algorithm is that a sufficient number of tumors in the cohort used for deriving the individual CpG models need to show difference in methylation in tumor cells compared to background “normal” cells or unaffected tumor cells. Adding the concept of subclonality to this equation would substantially increase the complexity of modelling and the requirement of the training cohort. The concept of subclonality in DNA methylation data is also by its nature difficult to tackle as each CpG position can theoretically only take on two states on the level of the individual DNA strand. Therefore, a subclonal deviation from a given tumor-specific methylation state will be the same as the background state of non-malignant cells, requiring some form of modelling of the tumor and normal cell compartments followed by a second modelling step to quantify heterogeneity within the tumor compartment. Unfortunately, even minor experimental noise from, e.g., the Illumina platform in combination with unavoidable uncertainty in all types of purity estimates could easily overwhelm or confound this type of analysis. With respect to this study, the matched WGS data for our discovery TNBC cohort does not have the necessary sequence depth to support in depth subclonality analysis. Moreover, it would be challenging to assess how well such a correction worked without equally rich validation data. Together, we believe this represents a study of its own where perhaps simulated DNA methylation data accounting also for the impact of copy number could be used as a starting point. So, with respect to “definitiveness” we agree with the reviewer that future improvements can likely be made to our method allowing for more in depth characterization of complete tumor methylation states. This is however by no means a point only applicable to our study and methods but an inherent part of the scientific process wherein incremental refinement and improvement takes us ever closer to fundamental truths, a process which must allow for some degree of compromise along the way as exemplified by, e.g., the TNBCtype paradigm. With respect to the question of reliability however, we show comprehensively and in multiple independent cohorts that our method of beta correction yields reproducible and reliable estimates which can provably both reproduce and validate previously known and published results and more importantly yield significant new insights and testable hypotheses regarding TNBC biology and epigenetic regulation in cancer.

The authors should perform a quality check to compare purity-adjusted versus non-adjusted clusters to validate their approach.

Response:

The estimated tumor purities in the three main tumor cohorts vary considerably as seen in **Rebuttal Figure 1A** below. With respect to this remark, we would like to point the reviewer

to results in Supplementary Figure S4 in the original manuscript demonstrating that purity-adjusted CpG data showed consistently higher accuracy for the PAM50 nonBasal and Basal split compared with conventional unadjusted CpG data in the discovery cohort. This finding is consistent with the results by Staaf and Aine et al. ⁷ in TCGA tumors (see **Rebuttal Figure 1B-D** below). Similar results were also reported in the study by Sasiain et al. developing the PureBeta pipeline ⁶. The study by Sasiain et al. also exemplified how beta adjustment could improve detection of promoter hypermethylation of tumor suppressor genes like *BRCA1* (see **Rebuttal Figure 1E-F** below).

Importantly, the NMF clustering of unadjusted beta value data shown in the original Supplementary Figure S4C is equivalent to the first main split into an epigenetic Basal and nonBasal cluster. Using unadjusted data for clustering means that there is a less distinct separation of PAM50 Basal and nonBasal tumors, which would appear to make less sense biologically. Of note, a less distinct separation of PAM50 Basal tumors is consistent with the high enrichment (70%) of these tumors even in the LAR-subgroup proposed by DiNome et al. ⁴. Moreover, a less distinct first split would propagate to subsequent subclustering analyses, acting as a confounder with respect to both “true identity” and the identification of defining biological traits.

In response to this remark, we have added the following text in the Result section connected to the supplementary figure (original Supplementary Figure S2) showing the effect of purity adjustment in the discovery cohort:

Such adjusted values have been shown to improve the epigenetic separation between the well-established PAM50 Basal and Luminal gene expression subtypes of breast cancer, as well as to enhance DNA methylation levels of individual genes like BRCA1 in samples 6, 7. As shown in Supplementary Figure S1, adjusted tumor data show less intermediate beta values, as would be theoretically expected given a binary methylation state on the level of individual cells.

Rebuttal Figure 1. **A)** Histograms of tumor purity estimates for the SCAN-B discovery cohort (left), SCAN-B validation cohort (center) and the 645-sample TCGA cohort (right). Beta-value correction improves PAM50 Basal vs non-Basal division of TCGA breast cancer samples. Example of hierarchical clustering heatmap of 500 random CpGs from 630 breast cancers from the TCGA consortium before **B)** and after **C)** adjustment for tumor purity. Adjusted clustering displays better separation of Basal and non-Basal samples (hierarchical tree cut at $K=2$). **D)** Purity adjustment of methylation beta values improves the separation of PAM50 Basal vs non-Basal samples in hierarchical clustering analysis in the TCGA cohort for both relative and absolute terms and when compared with dichotomization or InfiniumPurify prior to clustering. Panels B-D adapted from our study Staaf and Aine et al., see the study for more detail. Purity adjustment improves clarity of *BRCA1* promoter hypermethylation. Beta values of CpGs connected to *BRCA1*. **E)** Original beta values (rows) per CpG for 29 CpGs that fall within a 2000 bp window around the transcription start site of the *BRCA1* gene in samples (columns) from tumors in

GSE148748. F) Same CpGs and samples in the same order as in (E) but showing adjusted beta values for tumor cells with four different methods for adjustment as outlined in Sasiain et al. Samples and CpGs are ordered by mean beta value per sample and per CpG respectively. Panels E-F adapted from our study Sasiain et al., see the study for more detail.

Additionally, the discussion section should emphasize the importance of purity adjustment by addressing the limitations of prior epigenetic subtyping studies on bulk tumors. Comparing the variability observed in this study with findings from previous studies would further contextualize the results and highlight the advantages of the current methodology.

Response:

In the original manuscript we highlight that prior methylation studies have not used a tumor purity correction approach at different instances (e.g. Introduction and start of Discussion), mentioning even that cellular composition could explain observed variability in those contexts. In response to other remarks by the reviewer we have added additional text with respect to previous DNA methylation studies (remark 3 above) and their limitations (remark 3 above and minor remark 5 below).

6. The patient cohort is primarily derived from the SCAN-B study, which may limit the generalizability of the findings. This limitation should be acknowledged and discussed in the discussion section to provide a balanced interpretation of the study's implications.

Response:

We agree with the reviewer that generalizability should be a key aspect of all studies. Generalizable findings can, however, be thought of in different ways. In contrast to the reviewer's comment, we believe the SCAN-B study does support generalizability based on the proven aspect of population representativity for the study as a whole and for e.g. the discovery cohort specifically (see 8, 9, ¹⁰). This aspect does not apply to TCGA (which is also not representative of tumor cell content as it is restricted to higher tumor cellularity samples as seen in **Rebuttal Figure 1A** above) or any other DNA methylation-profiled cohorts to date (not even the METABRIC cohort commonly used for mRNA studies). Notably, the study by DiNome et al. did not even include TCGA TNBC tumors with an estimated cellularity <60%. With respect to non-generalizable findings based on ethnicity, this could be a weakness as SCAN-B patients are mainly expected to be European Caucasian as it is a Swedish cohort, a limitation that also applies to several other publicly available large cohorts. Generalizability can also be regarding methodological choices. Another strong feature of the current study lies in the usage of the EPIC platform compared to previous studies relying on the 450K platform (e.g. TCGA-based) which is more limited with respect to CpGs in distal regions. The latter can impact on the ability to represent all aspects and variation related to e.g. CpG contexts. In the original manuscript we do acknowledge limitations about generalizability, in particular arguing for that the proposed more refined Basal and nonBasal subgroups need additional validation, that larger discovery cohorts may reveal additional subgroups, and more.

In response to this remark, we have included the following text in the Discussion of the revised manuscript to stress the question about generalizability and ethnicity:

While the discovery cohort has been shown to be population representative ⁸, a generally positive characteristic with respect to generalizability, it is representative of a specific demographic context of patients (Western Europe/Nordics). Consistently, it should be investigated whether the proposed DNA methylation patterns observed in the current study also hold true in other ethnic contexts.

7. The classification and subclassification of TNBC into Basal and nonBasal subtypes, and further into six subgroups, are well explained. However, the clinical significance of these subgroups is not addressed. Including a discussion of the potential clinical implications of these subgroups would strengthen the argument and provide a more comprehensive understanding of the study's relevance in the discussion section.

Response:

In the original manuscript we provide survival comparisons for all three major DNA methylation subgroup comparisons: i) Basal vs nonBasal (significant difference, Kaplan-Meier plot in Supplementary Figure S5), ii) Basal1-3 (Figure 3C), and iii) nonBasal1 vs nonBasal2 (no significant difference, only a log-rank $p > 0.05$ reported in the manuscript).

We believe that clinical significance and biological subtypes do not necessarily need to completely go hand in hand in TNBC. In this breast cancer subtype, immune response (e.g. measured by tumor infiltrating lymphocytes, TILs) is an acknowledged strong indicator of better prognosis and it is thus of clinical significance. In the discovery cohort, we have previously demonstrated that DNA repair deficiency (homologous recombination deficiency, HRD, primarily caused by *BRCA1/BRCA2* deficiency) also provides independent prognostic information for chemotherapy-treated patients⁸. HRD status would then also be of clinical significance. However, in previously reported mRNA subtypes both immune infiltration and HRD transcend the proposed subtypes. Consistently, the refined TNBCtype mRNA subtypes were not significantly associated with patient outcome in the refined subtype study by Lehmann et al., nor are the four FUSCC subtypes when analyzed in FUSCC chemotherapy treated patients only (log-rank $p = 0.32$, $n = 293$ patients). Still, it is argued that the mRNA subtypes might have clinical implications based on genomic features associated with them. However, several of these key genomic features (such as *PIK3CA* mutations, immune infiltration, HRD, AR expression) are detectable irrespective of whether we know a tumor's mRNA subtype. Thus, it could be argued that mRNA subtypes in TNBC are not very clinically significant. A similar conceptual argument can be made about Ki67 expression and the PAM50 Luminal A / Luminal B separation in ER-positive / HER2-negative disease. Notably, while the survival analyses present in the manuscript could be used to argue clinical importance (just as for the survival plots shown by Lehmann et al., Burstein et al. or Jiang et al. for respective mRNA subtyping scheme) it would not serve as critical support to claim superiority versus e.g. TILs in a clinical setting.

Therefore, we do not for now propose that the epigenetic subgroups represent new relevant prognosticators in TNBC. Instead, in the current study we provide potential insight into why for instance immune-inflamed TNBC shows high PD-L1 or why FASN exhibits high expression (potentially targetable by inhibitors under investigation) in a specific patient subset. Importantly, these findings go beyond what mRNA or epigenetic analyses alone can inform us about – illustrating the benefit of our integrated methylation–mRNA approach of the current study and where we believe the highest clinical and therapeutical value lies.

In the revised manuscript we have added the following paragraph to the Discussion with respect to clinical and therapeutical implications:

Patients with systemic therapy included in this study were all treated according to national Swedish guidelines during inclusion years, which for the discovery cohort meant mainly a FEC-based (combination of 5-fluorouracil, epirubicin and cyclophosphamide) chemotherapy regimen⁸. Survival analyses demonstrated better outcome for patients with Basal compared to nonBasal tumors, a trend of better outcome in patients with Basal3 tumors compared to Basal1 and Basal2 patients, but no difference in outcome between nonBasal1 and

nonBasal2 patients. The observed better prognosis for patients with immune-warm Basal3 tumors aligns with the established prognostic role of the TIME in TNBC¹¹. In metastatic TNBC, high TILs/PD-L1 expression appears predictive of immune checkpoint inhibitor efficacy, while in early-stage pre-operatively treated disease pathological complete response rates are higher for immune-warm tumors with or without checkpoint inhibition (reviewed in¹¹). While it may be speculated that patients with “immune-warm”, PD-L1 expressing, Basal3 tumors could be the ones benefitting the most from immune checkpoint inhibitors like pembrolizumab, now part of clinical management of early-stage TNBC¹², it should be noted that patients with high TIL levels have as a group an intrinsically good prognosis even without adjuvant treatment¹³. Other characteristics of the DNA methylation subgroups aligning with recent or emerging treatment options in TNBC include for nonBasal1 and nonBasal2 patients potential androgen receptor inhibition therapy (based on increased AR expression), PI3K/AKT inhibitors (based on frequent PI3K pathway alterations including PIK3CA mutations), or combinations thereof as outlined in different studies (e.g., 14, 15 and reviewed in¹⁶). Interestingly, for the mainly older patients with nonBasal2 tumors a potential future treatment option may be inhibition of FASN, for which multiple inhibitors are under development¹⁷. While TROP2 antibody-drug conjugates are available for treatment of metastatic TNBC, we found no significant differences in mRNA expression of the TROP2 gene (TACSTD2) between the five DNA methylation subgroups (two-sided Kruskal-Wallis test, $p=0.1$) suggesting no informative value for this therapy. Similarly, while 93% of all HRD-positive tumors (primarily driven by BRCA1/BRCA2-deficiency) were found in the Basal epitype, these did not match a distinct Basal subgroup (two-sided Chi-squared test, $p=0.94$) suggesting that the latter would not be informative for PARP-inhibitor usage.

8. Most of the findings related to FOXA1 methylation are currently presented in the supplementary material. To enhance the manuscript's impact, it would be beneficial to integrate these findings into the main text or include them in a figure. This would highlight the significance of these results more effectively.

Response:

We agree with the reviewer that the results around *FOXA1*, *FOXC1*, and *SLC44A4* are of high interest (and striking). It should be noted (as pointed out by reviewer 1) that specific CpGs in *FOXA1* have been shown to be PAM50 Basal specific (supplements of Corces et al.). The original manuscript was limited to 6 display items as this is a common journal limitation. As informed by the Senior Editor of Nature Communications, we are allowed to have more than 6 main display items in our manuscript. We have therefore opted to improve the overview of the study by creating a new Figure 1 providing a layout of the study as well as a cohort summary table (Table 1). Moreover, based on a request by reviewer 1 we have also merged the three separate validation sections in the original manuscript into one section at the end with a new associated main figure (Figure 8).

Considering the above, and that the reviewer in the next remark requested us to omit non-breast cancer analyses mainly targeting *FOXA1* and *FOXC1*, we have, to strike a balance between reviewers, opted to keep the remaining (mainly cell line based) *FOXA1* analyses as supplementary figures.

9. The sections assessing overall breast cancer and other cancer-related analyses should be removed, as they are not directly relevant to TNBC. Focusing solely on TNBC will strengthen the manuscript's focus and clarity.

Response:

We do believe some of the non-TNBC exclusive analyses are important to keep in the manuscript in order to support specific conclusions about the generalizability of our findings. This relates to:

- Analyses associated with the unrelated 6009-sample SCAN-B RNA-sequencing cohort that support the uniqueness of the metabolism metagene identified in nonBasal2 tumors in breast cancer.
- The TCGA analyses that demonstrate and support TNBC Basal-specific methylation patterns also for PAM50 Basal breast cancers nearly irrespective of underlying ER, PR, and HER2 status.
- The usage of histone marker data for immune cells and luminal progenitor cells to support that the CpG patterns we observe with respect to CpG-mRNA correlations are not solely due to immune cell expression.
- The methylation analysis of flow-sorted immune cells, as these data also add to the support of the specific CpG-gene patterns for PD-L1 and PD-L2 we observed in TNBC.
- The pan-cancer analysis of the specific CpG-gene patterns for PD-L1 and PD-L2. In our tumors and in our limited TNBC cell line set, these findings are less distinct than for e.g. *FOXAI*, partly related to EPIC coverage in the regions of interest. We strongly believe that showing similar methylation-expression patterns in a larger cohort of cancer cell lines, representing different cancer types, all lacking active immune compartments substantially strengthens the observation.

To comply with the remark by the reviewer, we have removed less impactful analyses like:

- Normal cell tissue analyses for *FOXAI* and *FOXCI* that were performed using GTEx data (original Supplementary Figure S7u).
- The pan-cancer cell line analyses for *FOXAI* and *FOXCI* (original Supplementary Figure S7t).

10. In the discussion section, the authors claim that their subtyping method offers an advantage over previous transcriptional and genetic classifications. However, they do not directly compare their classification in terms of therapeutic relevance. To strengthen this argument, the authors should address how their subtyping could inform treatment decisions or improve therapeutic outcomes. A direct comparison with existing subtyping systems and highlighting how their classification may offer more precise insights into treatment response, therapeutic targets, or patient prognosis would provide a more compelling rationale for the claimed advantage.

Response:

In the Result section of the original manuscript, we provide multiple direct comparisons such as requested by the reviewer between the proposed epigenetic subgroups and existing mRNA subtypes (both PAM50 and TNBCtype) and important genetic classifications (HRD). We also provide data on the immune status for the groups, by e.g. TIL levels and an mRNA-based immune metagene as well as RNAseq-based CIBERSORTx deconvolution data. In essence we do therefore compare our proposed epigenetic subgroups to other molecular subtypes, as well as to prognostic features in TNBC (HRD and immune infiltration status), representing what we believe the reviewer is asking for.

Moreover, our response above regarding a previous remark by the reviewer about clinical importance (remark 7), in the original manuscript we report survival analysis for patients treated with standard-of-care adjuvant chemotherapy using distant relapse-free interval (DRFI) as clinical endpoint for all three major DNA methylation subgroup comparisons.

With respect to therapeutic outcomes, it is first important to recognize that the current study is based on analysis of retrospective patient cohorts that lack more directed “modern”

therapies. While TCGA is generally not a good cohort to perform survival analysis in due to recruitment bias and lack of quality treatment data¹⁸, patients in the SCAN-B cohorts used in this study have only been treated with standard-of-care chemotherapy during inclusion years (2010-2015 for the discovery cohort, and mainly 2015-2018 for the validation cohort). Importantly, this renders any discussion about the therapeutic importance of the epigenetic subgroups in actual patients speculative by nature.

In the current manuscript, we discuss that patients with Basal3 tumors would potentially respond more favorably to, for instance, immune checkpoint therapy based on PD-L1 expression and immune status. It is important to note that we do not propose that epigenetic patterns associated with Basal3 should be used as a treatment predictive biomarker for such therapy. Any such claim would involve establishing a new complex assay in clinical routine that would likely not outperform currently approved alternatives like TIL estimates. Instead, importantly, the current study provides likely insight as to why we see the described immune expression/infiltration patterns. Similar reasoning applies to the part in the Discussion around FASN, i.e., there might be more relevant biomarkers for drugs being developed targeting FASN than methylation status of the FASN promoter (e.g. mRNA/protein expression of FASN). What the current study provides are potential insights as to why that gene is expressed and in which patient group specifically. To reiterate a previous response, the current study with its integrated view of DNA methylation and mRNA expression (as well as protein measurements in vitro) provides insights beyond what single layer studies (mRNA or epigenetic alone) could achieve with respect to potential mechanisms behind therapeutic possibilities.

Please see response to remark 7 above for textual changes related to clinical significance and therapeutics.

11. The authors argue that epigenetic regulation could be a potential therapeutic target area. However, they do not provide sufficient literature supports to demonstrate that these novel methylation changes are not merely correlated but actually serve as valuable therapeutic targets. Additionally, while the authors suggest that their epigenetic classification could be therapeutically valuable, the argument appears speculative without concrete evidence linking therapeutic responses to variations in the methylome. To make this claim more compelling, the authors should provide evidence or examples from clinical or preclinical studies showing how these methylation changes might influence treatment outcomes or response to specific therapies.

Response:

In the original manuscript we pointed to two main therapeutic implications: i) TNBC tumors of the Basal3 group could be more responsive to immune checkpoint inhibition based on their PD-L1 expression, and ii) tumors of the nonBasal2 phenotype could potentially benefit from FASN inhibitors under development. Importantly, we do not claim in the manuscript that the observed methylation patterns are treatment targets per se, or that specific methylation subtypes would be treatment predictive for, e.g., methylation inhibitors. Notably, no DNMT or HDAC inhibitors are currently approved for clinical use in breast cancer¹⁹. Please see responses to remarks 7 and 10 with respect to clinical and therapeutic relevance and associated text changes.

Minor Comments

1. Abbreviations should be defined when mentioned for the first time in the manuscript and then used consistently throughout the rest of the manuscript.

Response:

We have checked the revised manuscript for this issue. It should be noted that the layout of the manuscript has been changed to now fit with the Nature Communications format, meaning that Methods appear after the Discussion.

2. The text is highly technical and densely packed with information, which may be challenging for non-specialists to follow. To improve accessibility, the authors should consider simplifying the language and breaking down complex concepts into clearer, more digestible explanations to make the findings more accessible to a broader audience, including researchers from other fields or clinicians with limited expertise in this specific area.

Response:

We agree with the reviewer that keeping information simple allows it to cater to a wider audience. We made an effort to simplify several sentences in the revised version of the manuscript, as well as to better explain concepts that will be needed for result comprehension (such as ATAC-sequencing patterns).

3. Please use "subgroup" or "subtypes" consistently throughout the manuscript, as "subbranches" is not an appropriate term in this context. This will ensure clarity and consistency in the terminology.

Response:

We thank the reviewer for highlighting the need to be consistent here. This has been addressed in the revised manuscript. We now use “subtype” for gene expression phenotypes, “epitype” for the main Basal/nonBasal epigenetic subtypes, and “subgroup” for clinicopathological groups and Basal1-3 and nonBasal1-2.

4. On page 2, lines 24-25, the statement “Triple-negative breast cancer (TNBC) is a clinically challenging entity that constitutes 10% breast cancer” is inaccurate. TNBC actually accounts for approximately 15-20% of breast cancers.

Response:

In the revised manuscript we have updated the TNBC proportion to “~10-20%”, adding supporting references from population representative data in the United States (Seer, previous reviews, and American Cancer Society).

5. The authors used distal ATAC peaks to infer regulatory potential but did not provide a clear rationale for this approach. It would strengthen the manuscript if the authors explained why distal ATAC peaks were chosen as a marker of regulatory potential and how they contribute to the overall findings.

Response:

As we cannot assess e.g. histone markers in our actual tumor tissues, we used ATAC-seq regions as a surrogate to infer regulatory potential, as a hallmark of active DNA regulatory elements is chromatin accessibility. Notably, eukaryotic genomes are compacted in chromatin (a complex of DNA and proteins), and only active regulatory elements are typically accessible by the cell’s machinery such as transcription factors (TFs). ATAC-seq quantifies DNA accessibility enabling the genome-wide profiling of TF binding events that orchestrate gene expression programs and give a cell its identity. With respect to distal ATAC-peaks being associated with cell and lineage specific events, early studies on blood cells which represent the most well understood and characterized cell types and differentiation cascades showed the value of this specific context for e.g. cell type discrimination²⁰. In terms of biological rationale, gene expression has long been known to be regulated through both direct promoter binding and

interactions with distal enhancers. Promoters often reside in a constitutively open state as evidenced by a predominant lack of DNA methylation and ubiquitous ATAC-peak overlaps even in cancer cells, especially in the context of CpG island promoters. This would theoretically allow binding of both general and specific transcription factors as well as promiscuous/spurious binding and thus potentially chaotic and deleterious transcription. Therefore, transcriptional regulation often involves additional input of e.g. enhancers which contribute both specificity and transcriptional signal boosting. In the evolutionary context these enhancers help direct correct spatial and temporal gene expression across embryogenesis, and the input of multiple elements responding to continuous morphogen gradients synergistically regulate precise expression of target genes. Through some measure of evolutionary happenstance and perhaps some intrinsic constraints on physical bending of DNA, the elements conferring cell type specificity tend to be localized in promoter distal contexts (or this is where we can at least clearly delineate them and quantify their effect). Our focus on distal regulatory elements therefore rests not only on our findings of, e.g., increased information content across this feature set and the work cited above, but also a sound biological rationale for why this feature set would be more informative with respect to biological identity. It is our belief that the chosen approach resulted in both novel and interesting insights to TNBC biology which points towards the validity of the approach, even if one were to consider the starting point as a mere hypothesis based on pure biological reasoning. As we cannot perform ATAC-seq on our SCAN-B tumors due to the tissue being freshly preserved in RNAlater, we used the breast cancer ATAC-peaks reported by Corces et al. We do acknowledge this usage as a limitation of our analyses in the Discussion of the original manuscript. As to why we chose specifically CpGs in distal elements in regions of open chromatin (distal-ATAC) to base our analyses on, this is explained in the first section of the Results of the manuscript but also motivated in the revised Introduction (see below). We hope to satisfy this reviewer comment with the discussion above even if we cannot incorporate it fully into the manuscript. We have, however, made small modifications to this section of the revised version of the manuscript in an effort to simplify the language used to improve the reader's experience.

In the revised manuscript we have added the following text to the Introduction in response to the remark:

Moreover, CpG can reside in regions with different chromatin accessibility, normally referred to as open (accessible) or closed (inaccessible) chromatin as measured for instance by assays for transposase-accessible chromatin using sequencing (ATAC-seq). A hallmark of active DNA regulatory elements is open chromatin with specific binding of transcription factors (TFs) and specific histone modifications including, e.g., histone H3 lysine 27 acetylation. DNA methylation at distal enhancer regions has been implicated in gene regulation by, e.g., interfering with TF binding to enhancer regions 21, 22, 23, 24. Together, this provides an additional layer of complexity when analyzing DNA methylation typically not accounted for in previous studies (like 2, 3, 4).

6. For the external datasets used in the study, it is unclear how the authors ensured comparability between data generated from different platforms, such as the EPIC and 450k arrays. The authors should clarify the normalization or harmonization methods applied to account for platform-specific differences and ensure consistency across datasets.

Response:

In the original manuscript these details are provided for all datasets when data was not directly obtained from e.g. GEO. All data sets were pre-processed separately. Briefly, the EPIC processing was performed as detailed in the Methods and Supplementary Methods using the Minfi R package. A difference between the discovery and validation cohorts in the processing was the usage of a single sample approach for the validation cohort (due to that it was experimentally performed at a later stage at another academic service facility as outlined in the Methods), however, entailing the similar steps as for the discovery cohort. Infinium probe normalization was performed for both EPIC cohorts using the approach described by Holm et al.²⁵.

Processing of TCGA was performed similar to the EPIC discovery cohort arrays as originally outlined by Staaf and Aine et al. (referenced)⁷. Again, Infinium probe normalization was performed using the approach described by Holm et al.²⁵.

Finally, tumor purity correction was performed in a similar manner for both the discovery EPIC and TCGA cohort using the approach outlined by Staaf and Aine (please see this study for explicit details)⁷. As the EPIC validation cohort lacks WGS we used the recently developed PureBeta pipeline, which implements the approach by Staaf and Aine et al.⁶, to perform purity correction.

For comparisons in TCGA we used only the CpGs overlapping between the filtered EPIC CpG set and the 450K platform. This is also stated in the Methods section.

For other DNA methylation data sets we obtained preprocessed data directly from GEO.

Taken together, the preprocessing approach taken is described, referenced, and similar between cohorts.

7. Eight TNBC cell lines representing different TNBCtype mRNA subtypes (HCC2157: BL1, HCC1599: BL1, HCC1937: BL1, BT-549: M, MDA-MB-231: MSL, MDA-MB-468: BL1, SUM185PE: LAR, and MDA-MB-453: LAR) were used in this study. Please explain why did not include BL2 and IM cell lines.

Response:

We did not have in-house access to BL2 cell lines and were unfortunately not able to accrue them in due time for this study. Regarding proposed IM cell lines by Lehmann et al., while the IM subtype was one of the six originally proposed subtypes, Lehmann et al. later excluded it and MSL as separate subtypes (see²⁶). Therefore, we did not include any IM subtype cell lines. As cancer cell lines lack an immune compartment by definition, it also remains somewhat unclear what an IM designation actually would represent in a cell line context. To note, we did have access to one cell line in house classified as MSL (MDA-MB-231) by Lehmann and therefore included it in the panel. Notably, this cell line was also extensively used by e.g. Huang et al. and Jovanovic et al., studies we use data from and cite to derive key conclusions (see^{27, 28}).

8. The authors chose the 1000, 2000, and 5000 most variable CpGs for clustering but did not provide a rationale for selecting these specific numbers. It would strengthen the methodology to explain why these particular thresholds were chosen.

Response:

These thresholds were chosen to provide data using different numbers of most varying CpGs for different CpG contexts, as opposed to just selecting a single number without any additional

investigation. The numbers themselves could be seen as arbitrary, but they are appropriate to illustrate differences (or more precisely, the lack of major differences) in clustering results as shown in the original Supplementary Fig S4a. Based on this simple investigation, we proceeded with sets of 5000 CpGs as these provided larger, but still manageable, numbers of informative data points. It should be noted that including too many CpGs (e.g., in the order of tens of thousands) means an increased risk of including low variant probes introducing “noise” to the analyses rather than biological variation depending on context (see e.g., **Rebuttal Figure 2** below for “raw” variance distribution of three contexts). We therefore did our best to evaluate robustness across multiple probe sets. But as for all preceding DNA methylation analyses published to date, we had to in the end make a somewhat arbitrary choice regarding the specific number of probes to be used in clustering.

In the revised manuscript we have in response to the remark rephrased a sentence to include the rationale for the different CpG numbers:

We evaluated two-class clustering similarity between all ten contexts using the 1000, 2500, or 5000 most variant CpGs within each context (allowing for investigation of the impact of different feature numbers on the clustering), as well as stratifying by overlaps with, without, or agnostic to ATAC-peaks (i.e., chromatin accessibility).

Rebuttal Figure 2. Standard deviation of tumor-purity adjusted beta values for distal-ATAC CpGs (left), promoter-ATAC CpGs (center), and all CpGs in the discovery cohort, ordered from high to low standard deviation.

9. In the result section for Fig 2a, on page 17, lines 437-457, the characterization of CpG cluster 2 is missing. While the authors provide a detailed characterization of CpG clusters (hyper- or hypomethylation) in the nonBasal subtype, they should also discuss the CpG clusters in the Basal subtype. Including this analysis would provide a more balanced and complete view of the methylation patterns across both Basal and nonBasal subtypes.
Response:

In response to the remark, we have the revised manuscript added the following text about CpG cluster 2 in the original Figure 2A in the connected Result section:

While CpG cluster 2 showed a similar TF enrichment to CpG cluster 3, the methylation profile of the former was less distinct between Basal and nonBasal tumors.

In response to the remark, we have in the revised manuscript added the following text about the CpG clusters in the Basal subclustering in the associated Result section:

Figure 4a shows the 2845 differentially methylated CpGs clustered into six groups for the three basal subgroups, Basal1 (n=53 samples), Basal2 (n=68), and Basal3 (n=54). CpG clusters 3 and 5 did not show strong enrichment of any tested TF, while CpG cluster 1

(associated with hypermethylation in Basal2 tumors) displayed weaker enrichment of TFs usually associated with promoters and other high CpG-density regions like RNA-polymerase II subunits and members of the polycomb silencing complex. CpG cluster 2 resembled CpG cluster 1 in TF enrichment but appeared more associated with hypermethylation in both Basal1 and Basal2 tumors compared to CpG cluster 1. Finally, CpG clusters 4 and 6 were enriched for FOS and STAT3 but showed different methylation patterns across Basal subgroups, mainly for Basal3 tumors (hypomethylation of cluster 6 while hypermethylation of cluster 4).

10. The figures are not very non-specialist friendly, and some figures, such as Fig 5, lack clear annotations.

Response:

In the revised manuscript we have expanded the number of main figures from six to 8 as allowed by Nature Communications and made an effort to add more annotations to all figures. Additionally, we reworked the original Figure 5 as it was explicitly mentioned by the reviewer.

11. On page 4, lines 63-65, the statement "Triple-negative breast cancer (TNBC) is negative for the three current treatment-predictive markers (estrogen, ER, and progesterone, PR, receptors and ERBB2/HER2 gene amplification) and constitutes ~10% of patients with primary disease" is inaccurate and needs to be rephrased. ER, PR, and HER2 are primarily used as treatment targets rather than predictive biomarkers. Additionally, TNBC accounts for approximately 15-20% of breast cancers, not ~10%.

Response:

We do believe that ER, PR, and HER2 can be referred to as “treatment-predictive markers” in breast cancer, please see for instance ²⁹ or the WHO website (<https://screening.iarc.fr/atlasbreastdetail.php?Index=138&e=>). We have amended the frequency of TNBC cases to “~10-20%” as mentioned in remark 4 above.

12. The citation format for references should be revised to ensure consistency throughout the manuscript. The authors should follow the appropriate style guide and ensure that all references are cited in the same manner.

Response:

All references are cited according to the EndNote Nature Communication style guide.

13. On page 3, line 74, “androgen (AR)” should be “androgen receptor (AR)”.

Response:

We have corrected this issue in the revised manuscript. Thank you for noticing this issue.

14. The manuscript inconsistently uses the terms "mRNA subtypes" and "transcriptional subtypes." To improve clarity and consistency, the authors should choose one term and use it consistently throughout the manuscript.

Response:

We have corrected this in the revised manuscript – we now use mRNA subtypes throughout. Thank you for pointing this out.

15. On page 6, line 144, where it says "merged with data from 14," and on line 148, where it says "with additional details provided in 4," and so on, the authors should provide context before citing the references.

Response:

This issue has been addressed in the revised manuscript.

16. On page 17, lines 450-452, the authors mention mapping the 5000 distal-ATAC CpGs to the enhancer-linked H3K27 acetylation (H3K27Ac) mark in breast luminal progenitor cells and three immune cell types using publicly available data. The authors should provide a rationale for why they specifically focused on these particular cell types: breast luminal progenitor cells, CD4, CD8, and macrophages. Given that this manuscript is focused on TNBC classification, it would also be valuable to explain why the study is limited to these specific immune cell types rather than including all immune cell types, which could provide a more comprehensive view of the immune landscape in TNBC. Response:

Firstly, the H3K27Ac data for cell types are not providing a comprehensive view of the immune landscape in TNBC, as it is based on pure in vitro cell line analyses using cell lines not derived from TNBC per se. These data are only used to demonstrate that specific epigenetic patterns we observe in tumor-purity adjusted DNA methylation data is unlikely to be due to immune cells (CD4, CD8, macrophages), but rather appear to be related to breast or at least epithelial cell biology (luminal progenitors). As such, the three immune cell types were chosen to generally represent immune cell types as an analysis with all cell types would be too cluttered and more than needed to showcase the point made as it is only the most abundant immunological cell types that could even theoretically confound or influence methylation states observed in our TNBC samples. To comprehensively address the immune landscape in TNBC one should typically employ e.g. multiplexed IHC, scRNAseq or spatial transcriptomics on actual tumor tissue, analyses that are becoming increasingly feasible today.

Reviewer #3, ECR (Remarks to the Author):

Response:

No remarks to address.

Reviewer #4, expertise in breast cancer omics and DNA methylation (Remarks to the Author):

In this study, Aine et al. stratified TNBC patients samples into epigenetic subgroups following the adjustment of Illumina EPIC beadchips-based global DNA methylation profiles for tumor cell content and the accounting for genomic context of CpG methylation. Epigenetic subgroups were validated in several independent cohorts and extensively characterized through integration of whole genome sequencing, RNA-sequencing, immunohistochemistry, and clinical data. They identified the epigenetic regulation of key steroid response genes and developmental transcription factors as important drivers of this subclassification. They further divided the Basal and non-Basal subgroups into several subbranches, identifying a distinct Luminal Androgen Receptor (LAR) phenotype among the non-Basal subgroups that is characterized by the putative epigenetic regulation of a specific metabolic gene network. Basal subbranches were

characterized by differences in immunogenicity and immune evasion strategies including a subgroup with epigenetically- regulated expression of the immune checkpoint gene PD-L1. They conclude that the subdivision of TNBC based on pure tumor methylomes and genomic context of CpG methylation transcends genetic and transcriptional TNBC subtypes by highlighting widely differing immunological microenvironments and putative epigenetically mediated immune evasion strategies.

This study is very interesting and timely and features a wealth of data and well and thoroughly executed data analyses. However, some concerns remain, especially surrounding the leap of knowledge of their epigenetic TNBC subgrouping with respect to gene expression-based TNBC subtyping.

Response:

We thank the reviewer for the positive and constructive comments and remarks provided.

Major comments:

1. The authors mention tumor heterogeneity and tumor purity as source of bias and as an underlying rationale for adjusting DNA methylation data for tumor cell content, irrespective of the well-established role and clinical importance of the TME/TIME. Could the authors better outline their rationale for focusing on TNBC intrinsic subtypes? Further related to this question, the authors state that “Importantly, purity-adjusted CpG data showed consistently higher accuracy for this split compared with conventional unadjusted CpG data (Supplementary Figure S4c), supporting the relevance of using DNA methylation levels adjusted for tumor cell content.” We encourage the authors to re-phrase this sentence as both approaches, i.e. focusing on tumour cells only and taking the tumor as a whole, are valid approaches.

Response:

The sentence the reviewer refers to addresses the two-group clustering solutions using different CpGs from different contexts and beta value adjustment (yes/no) versus the PAM50 Basal/nonBasal division of TNBC. In this specific case, we do observe a higher concordance between DNA methylation clusters and PAM50 Basal classification using adjusted beta values. For there to be no future confusion, we have followed the reviewer’s advice and rephrased the mentioned sentence to:

Importantly, purity-adjusted CpG data showed consistently more stable and similar accuracies for this split across tested CpG contexts compared with conventional unadjusted CpG data (Supplementary Fig. S3c).

We agree with the reviewer about the importance and relevance of the TME/TIME in TNBC, and the importance of bulk mRNA profiling in breast cancer research in general. Immune infiltration (estimated for instance by TILs or mRNA patterns) is a well-established prognostic and increasingly predictive marker in TNBC, unlike e.g. the “intrinsic” PAM50 subtypes. Moreover, TILs alone would provide a much more powerful and clinically achievable prognostic stratification of TNBC than the DNA methylation subgroups proposed in our study. Interestingly, PAM50 subtypes (and their predecessor “intrinsic subtypes” first described by Perou et al.³⁰) were aimed to be “tumor intrinsic” in nature, and the 50 genes selected to define the subtypes correlate only weakly to a typical immune gene expression metagene as recently illustrated by us in a large SCAN-B cohort³¹. In contrast, TNBCtype subtypes appear strongly connected with the TME/TIME, exemplified not in the least by the reduction of the original six

subtypes to four finding that the transcriptional signatures from the previously described immunomodulatory (IM) and mesenchymal stem-like (MSL) subtypes were contributed from infiltrating lymphocytes and tumor-associated stromal cells, respectively ²⁶. Similarly, as previous DNA methylation-based subtyping studies did not use tumor purity-corrected beta values (as described in the Introduction and Discussion of the original manuscript), any identified subgroups would be inherently related to the composition of malignant and nonmalignant cells in the analyzed bulk tumor pieces. The latter and the historic lack of accounting for CpG contexts were the main rationale for revisiting DNA methylation subgroups of TNBC using a larger cohort, i.e., our hypothesis was that accounting for these aspects could identify new and hopefully clearer biological insights in TNBC. This does of course not invalidate previous findings or insights but does highlight how different approaches and research goals can lead to somewhat different conclusions and ways of viewing the same disease entity.

To not seemingly undermine the importance of previous and ongoing mRNA bulk analyses, we have made small modifications to our manuscript including adding the following text to the Introduction section in response to the reviewer's remark:

On a general level, bulk tissue mRNA studies represented the key first steps toward a detailed molecular phenotyping of TNBC tumors and their TME/TIME characteristics, and bulk tissue profiling still probably represents the most easily standardizable and clinically amenable approach to tumor profiling.

2. The authors suggest the use of distal regulatory CpGs for investigating TNBC DNA methylation phenotypes, yet their results seem to indicate the importance of other classes of CpGs for the distinction between Basal and nonBasal subgroups:

- CpGs with higher methylation in nonBasal samples were more frequently in high-density CpG island regions

Response:

Here the reviewer touches upon a set of very interesting observations made in the discovery cohort that were not included in the original manuscript due to restricted space and a need for a more streamlined story. Firstly, based on Supplementary Figure 4 one can conclude that a Basal/nonBasal split can generally be achieved irrespective of CpG context and even partly independently of purity adjustment, at least to a high sample overlap between different cluster analyses. This tells us that the Basal/nonBasal split is a main feature of the TNBC DNA methylation landscape. This is however not exclusively linked to the fact that nonBasal tumors exhibit pervasive hypermethylation of high-density regions. In fact, there are at least two distinct and different ways of capturing the nonBasal/Basal split using largely non-overlapping feature sets. The first way is through profiling of high-density regions as noted by the reviewer, this approach captures the strength of a "methylator-like" phenotype which is clearly strongest in LAR-tumors but less striking in nonbasal 1 tumors. On the contrary in distal ATAC regions, which are overwhelmingly low-density positions, it is instead FOXA/GATA-linked sites that most clearly capture the nonBasal/Basal split but with less difference in methylation states between nonBasal1 and 2 tumors. This is in fact largely what drives differentiation between Basal and nonBasal tumors in our top-level split in which high-density CpGs are scarce. The fact that one feature set works on a gradient and the other does not could indicate that these are in fact fundamentally different processes. But as both CpG sets largely capture the same samples in cluster analyses, resulting differential methylation analyses will identify the same set of significant differentially methylated CpGs. These CpGs will in this specific case include

both dynamics highlighted above. These methylation dynamics however are not equally represented in the “high variance landscape” and this in turn influences what becomes highlighted as statistically significant features of nonBasal tumors. To this end, any stratification that captures a sample set sufficiently enriched for a “methylator phenotype” will end up highlighting pervasive high-density methylation as a defining feature because thousands of baseline hypomethylated island CpGs concordantly switch to a hypermethylated state and therefore come to dominate with respect to statistical significance. In turn, hypomethylation of low-density CpGs in distal regulatory sites, which are likely fewer in number (or less well represented on Illumina arrays), become de-emphasized in these specific global analyses as this process is pushed down in the global “variance stack” by the co-existence of CGI hypermethylation.

In contrast to these dominant global themes defining nonBasal vs Basal disease, for the subgroup analyses we reached a partially different conclusion. For the nonBasal division into nonBasal1 and nonBasal2 the reviewer is correct that we could have achieved almost a similar sample division as the distal-ATAC NMF based solution if we would have used only the 5000 most variable CpGs irrespective of context (referred to as “bySD”) (**Rebuttal Figure 3A** included below). This is largely because the LAR/nonBasal2 tumors represent a very strong and distinct epigenetic and transcriptional entity that can be defined using virtually any data type or feature set, although epigenetically most distinctly through CGI hypermethylation, which in turn is enriched for in the high-variance set of CpGs but is also captured in the distal-ATAC set. For the subclustering of the 175 Basal tumors we however reached a different conclusion. Here, using a most variable CpG approach (“bySD”) resulted in a sample division clearly less concordant with the distal-ATAC solution (**Rebuttal Figure 3B** below). There were more differentially methylated CpGs for the bySD k=3 solution compared to the distal-ATAC solution, however with almost zero overlap on the individual CpG level (**Rebuttal Figure 3C** below). Together, this suggests that different aspects of TNBC biology are captured depending on the genomic context of the assayed CpGs and that this effect is more pronounced in Basal tumors.

For both the bySD and distal-ATAC approaches, only a minority of differentially methylated probes were localized in high-density regions (**Rebuttal Figure 4A and B**). Instead, the most pervasive feature of Basal TNBC methylation for both approaches was hypomethylation of baseline methylated low-density CpGs, although the respective analyses captured wholly different CpG sets as noted above. This again highlights how different processes seem to be represented on different levels of the variation landscape and that using a regulation-associated filter such as ATAC-peak overlaps may filter out non-regulatory variation that may otherwise influence sample clustering. Consistently, while the three-group distal-ATAC clustering result was associated with a significant number of differentially expressed genes (n=5005) (original Supplementary Figure S7c), the bySD clustering result was associated with only 447 differentially expressed genes using the same statistical approach, suggesting that the bySD approach captures fundamentally different and less phenotype-related biology. Further supporting the latter hypothesis, we found that the bySD three-group sample clustering captured a pattern of pervasive hypomethylation affecting a large number of distal low-density CpGs, largely devoid of overlaps with ATAC-peaks, enriched for localization in both giemsa-positive cytobands and repetitive elements compared to the total EPIC platform, but with

reduced overlap with regulatory regions based on ENCODE TFBS and CRE overlaps, a polar opposite to our findings for the distal-ATAC solution (**Rebuttal Figure 4** below).

Rebuttal Figure 3. Effects of different CpG usage in NMF clustering of nonBasal and Basal tumors. **A)** Sample overlap between NMF k=2 solutions for nonBasal tumors using either the 5000 most variant distal-ATAC CpGs, or just the 5000 most variant CpGs irrespective of context (“bySD”). **B)** Sample overlap between NMF k=3 solutions for Basal tumors using either the 5000 most variant distal-ATAC CpGs (corresponding to the main manuscript analysis), or just the 5000 most variant CpGs irrespective of context (“bySD”). **C)** Differentially methylated CpGs and their overlap for each of the two k=3 NMF solutions shown in panel B. As seen, there is very little overlap. Differentially methylated CpGs were defined based on a Kruskal-Wallis test, Bonferroni adjusted $p < 0.01$ and an absolute beta-value difference > 0.25 . The distal-ATAC CpGs represent the main manuscript analysis.

Rebuttal Figure 4. **A)** CpG characteristics of differentially methylated CpGs associated with a “bySD” NMF approach in Basal tumors, using the top 5000 most variant CpGs. **B)** Corresponding characteristics for the distal-ATAC approach used in the main original manuscript.

In the end, including all this data meant that the originally planned manuscript became too large and was also deemed to be too challenging to read. We have therefore initiated a new methodological project to more comprehensively analyze the variance structure of DNA methylation data in breast cancer and hope to be able to communicate our findings to a wider audience in future works.

- CpG shore/CGI methylation for FOXA1 that aligned almost perfectly with the Basal (hypermethylated shore/CGI CpGs) and nonBasal (hypomethylated shore/CGI CpGs) epigenetic stratification

Response:

This is correct and represents a striking feature of Basal disease. In fact, one could based on this observation argue that a more refined definition of Basal disease should be based on, e.g., the axis of *FOXA1* and *FOXC1* using DNA methylation status instead of mRNA expression or IHC surrogate markers. Or, that PAM50 Basal subtyping could be refined/retrained based only on these genes (see also **Rebuttal Figure 5** below).

While we could define PAM50 Basal/nonBasal disease by only a handful of promoter region CpGs, the DNA landscape of Basal TNBC clearly differs from nonBasal, as e.g. illustrated by us in the 2016 study by Holm et al. (using 450K illumina data not adjusted for tumor purity)²⁵. While not included in the current manuscript (aligning with the request by reviewer 2 to only focus on TNBC aspects) we do find that the overlap of the 28K differentially methylated CpGs between the Basal and the nonBasal epitype strongly separates (TNBC) Basal tumors from all other breast cancer groups in the TCGA cohort (i.e. similar appearance as FOXA1).

What we believe to be the next major challenges is to better understand the variance structure in DNA methylation data general across breast cancer, but also more specifically to connect the distal CpG methylation patterns we see enriched for *FOXA1/GATA3* TFBS overlaps with the presumed target genes and identify an actual FOXA1 regulome (“who does FOXA1 regulate”) based on epigenetics, an ambition that would require extensive in vitro experimentation.

Could the authors comment on that and also advise on how to take into account the genomic CpG context in future studies based on the finding of this study?

Response: We do hope that publication of this study in a high-impact journal like Nature Communications, together with an ongoing deeper methodological study performed in our group focusing on biological variation (beta-value variance) versus CpG context in breast cancer, will make the scientific community more aware of the risks of just using a “most variable CpG” approach in unsupervised clustering based on beta values. Depending on the underlying biology in the studied sample set different conclusions can be reached, and biological meaning/relevance be lost as described in a previous response (see **Rebuttal Figure 3** and **4** above and the associated response). While our original intention was to portray both the agreement in clustering as well as the difference in subclustering results in the Basal epitype when using different CpG contexts we found that this in the end consumed too much manuscript space and would risk making the manuscript less streamlined and even more advanced for a less experienced reader. A more thorough methodological study is thus warranted (and has been initiated), which can also include sample exclusion experiments similar to those that showed the weakness of the current PAM50 centroid based subtyping (see e.g. Paquet et al.³²). We do in the original manuscript already stress the importance of considering CpG context but also tumor cell content for unsupervised analysis of DNA methylation data. As a general comment however, one could advise future efforts to not treat all CpGs as equivalent in terms of both function and regulatory potential. Firstly, DNA methylation in high- versus low-density regions have polar opposite default states and whereas ATAC-peak overlaps often indicate focal demethylation and function in low-density distal regions, they often merely denote a baseline universally open

state at CGIs. Demethylation (or chromatin accessibility) at distal regions should also be more closely linked with specific rather than general drivers of gene expression. One should also contemplate that different processes are represented by vastly differing numbers of CpGs and that this will lead them to “inhabit” different levels of the variance structure of the data. Cohort composition will also influence what variation exists and comes to dominate statistical significance and given that too few samples of a given subtype are present in the cohort, different variance structures will become more or less prominent in downstream analyses.

3. The authors thoroughly characterize their epigenetic subgroups with respect to gene expression-derived TNBC subtypes, but could the authors better explain what the epigenetic subtyping approach of this study adds on top of what is already known based on gene expression and how this is clinical relevant?

Response:

Firstly, it should be acknowledged that our DNA methylation clustering of the discovery cohort is likely not perfect, given that some PAM50 Basal tumors are in the nonBasal arm, and how the heatmaps appear. Notably, unsupervised analyses are rarely perfect for all individual tumors. With respect to PAM50, the main epigenetic division adds little compared to a general Basal/nonBasal distinction. However, as illustrated by promoter region methylation patterns for steroid response associated genes, *FOXC1*, etc., epigenetics provide plausible explanation to key expression patterns defining the PAM50 Basal subtype (e.g. both *FOXAI* and *FOXC1* are present in the PAM50 gene list) considering that e.g. sample correlations to the PAM50 Basal centroid appear strongly negatively correlated to *FOXAI* mRNA expression (**Rebuttal Figure 5A** below). As responded to earlier, we believe the next challenge lies in more deeply deciphering the distal methylation patterns observed in regions of open chromatin enriched for overlapping TF binding sites. Understanding the combinatorial transcription factor inputs that produce the full spectrum of TNBC phenotypes would move the field forward in a significant way. In addition, as mentioned in the original Discussion we also found that the epigenetic stratification divided TNBC tumors subtyped as PAM50 Normal-like into the Basal/nonBasal context with aligned expression and epigenetic patterns to those of other tumors in these branches. This observation indicates that these specific tumors’ underlying PAM50 subtype might be masked in bulk RNAseq data by e.g. low tumor cell content.

In the original Supplementary Figure S9 we do provide some insight into what the epigenetic stratification implies for the TNBCtype mRNA classification. In this figure it appears, even for the comparisons with low sample numbers, clear that TNBCtype subtypes stratified by the DNA methylation Basal/nonBasal arms corresponds to distinct differences in *FOXAI* expression. For the BL1, M, and LAR subtypes the few tumors in the “wrong” methylation arm may be interpreted as “misclassified” cases (classification is performed by selecting the highest centroid correlation obtained from the online TNBCtype classification tool), suggesting that DNA methylation may actually assist in refining the bulk mRNA-based classification for “borderline” cases. A similar conclusion appears less likely for the BL2 subtype, where 34% of tumors are in the nonBasal methylation arm and 66% in the Basal arm aligning with a significant difference in *FOXAI* mRNA expression. With respect to expression of the Fredlund et al. metagenes, the stratification of BL2 tumors by DNA methylation arm aligned with the general Basal/nonBasal patterns of lower steroid response, higher immune infiltration, and higher basal expression in BL2-Basal tumors compared to BL2-nonBasal tumors (**Rebuttal Figure 5B** below). The reasons for this may be multiple, including that BL2 is “intrinsically” a “weaker/less distinct” mRNA subtype compared to BL1, M, LAR, a mix of tumors with

different tumor microenvironments, mixed tumor cellularity, etc. Notably, findings like these challenges the conception of BL2 as a distinct TNBC mRNA subtype, exemplifying how epigenetics can refine bulk tumor mRNA subtypes on a more fundamental level. However, we believe that refining molecular subtypes of TNBC based on a more integrative multi-omics approach involving mRNA expression, epigenetics, but also DNA alterations and protein expression represents a study of its own. As the reviewer remarked previously, mRNA expression will be of key importance here, as it provides a more holistic view of the tumor ecosystem compared to “tumor intrinsic” DNA methylation patterns alone.

In response to the remark, we have added a summarizing sentence to the second paragraph of the Discussion of the revised manuscript:

In this setting, tumor methylome profiles can both provide refinement of existing mRNA subtypes and be an important part of integrative multi-omics subtyping studies in TNBC.

Rebuttal Figure 5. **A)** PAM50 Basal centroid correlation (obtained from Veerla et al. ³³) versus *FOXA1* FPKM expression in the SCAN-B discovery cohort. **B)** Rank scores for Fredlund et al. metagenes for TNBCtype BL2 tumors stratified by the DNA methylation Basal/nonBasal classification in the discovery cohort.

Concerning clinical relevance, we would, in order not to extensively reiterate responses to other reviewer remarks in the rebuttal, like to point the reviewer to a previous response to a similar remark made by reviewer 2 (please see remarks 7 and 10) and the associated text changes.

4. It would be interesting to know how the epigenetic TNBC subgroups of this study compare with those of previous studies. Are their overlaps between the different epigenetically-defined TNBC subtypes?

Response:

The reviewer raises a highly relevant question. As stated in the original manuscript, there has been no systematic comparison of previously reported TNBC DNA methylation subgroups. Based on a request by reviewer 2 (please see remark 3 by reviewer 2 above), we have expanded the Introduction section of the revised manuscript with additional information about three previous DNA methylation studies for TNBC ^{2,3,4}. It should be noted that all three studies:

- are based on small discovery cohorts (<80 samples);
- are not based on population-representative cohorts;
- do not provide any validation of proposed DNA methylation subtypes in independent cohorts;
- did not explicitly consider CpG context/density;
- did not consider tumor purity (notably, the study by DiNome et al. ⁴ was even based only on TCGA tumors with >60% cellularity).

We believe the text newly added to the Introduction of the revised manuscript illustrates the issues with previous studies as outlined above. Of note, the study by Stirzaker et al. ² only reported a survival difference between the proposed subgroups as a main difference/characteristic between them, making it difficult to map their subgroups to the ones proposed in our study. Based on our understanding of the study by Lin et al. ³, they seem to also identify a likely mainly Basal group and a non-Basal group, considering that the latter was mainly composed of older patients and tumors with lower proliferation, lower immune infiltration, and *PIK3CA* mutations. The lack of mRNA data, however, does not explicitly clarify this. While DiNome et al. ⁴ do report that one of the four subgroups (Epi-CL-B) was enriched for LAR tumors, the Epi-CL-B subgroup was still 70% composed of PAM50 Basal-like tumors. Based on our results shown in the original Supplementary Figure S4 (revised Supplementary Figure S3), we found that a main DNA methylation split separates PAM50 Basal from nonBasal tumors, but that this split is less distinct in non-purity-adjusted data. This, combined with not taking CpG contexts into consideration, might explain the Epi-CL-B subgroup composition. In addition, the Epi-CL-A subgroup showed some resemblance to the EMT-enriched Basal2 subgroup in our study given an EMT-like phenotype. Finally, it is important to point out that our study's discovery cohort is population-representative and approximately three times larger than that of previous studies, and the chosen validation cohort is approximately two times larger.

In the revised manuscript we have updated the Discussion with the following text:

Based on our methodological approach, we postulate the existence of two major epigenetic backbones in TNBC, a Basal and a nonBasal epitype, that should represent distinct DNA methylation ground states of the disease. These epitypes align well with the intrinsic PAM50 subtypes in both the SCAN-B discovery and validation cohorts. In previous DNA methylation studies in TNBC that have used different CpG selection strategies, not considered CpG context, nor used tumor purity-corrected DNA methylation data, we find both partial

agreement and discordance to our proposed ground states. In the study by Lin et al., two of three subgroups show clinicopathological characteristics that could align with a Basal and nonBasal division, although sample numbers are small and mRNA profiling is lacking³. In the study by DiNome et al. reporting four subgroups, while one subgroup was enriched for LAR tumors, 69.2% of tumors in that subgroup were classified as PAM50 Basal-like⁴, unlike in our subgroups. Notable features of the Basal and nonBasal epitypes were their strong association with clinical parameters like patient age, mRNA subtypes, HRD status, and TIMEs, consistent with what has been reported in non-epigenetic based studies for this major subdivision. Notably, the epigenetic stratification also divided TNBC tumors subtyped as PAM50 Normal-like into the Basal/nonBasal context with aligned expression and epigenetic patterns to that of other tumors in these epitypes, indicating that the underlying PAM50 subtype of these specific tumors might be masked in bulk RNAseq data.

5. I wonder whether the results described in the following statement “Gene set analysis using “hallmark” signatures of differential genes in Basal3 tumors confirmed a strong association with multiple immune pathways (Supplementary Table S3b).” are of intrinsic or extrinsic nature. Could the authors redo this analysis with a purity for Basal 3 that matches the purity of Basal 1 & 2? The authors could use regression to suppress the cell composition in their expression data, like they did with their DNA methylation data. Comparing adjusted and unadjusted gene expression changes could allow to distinct between intrinsic and extrinsic contributions.

Response:

This remark partly relates to a comment by Reviewer 1 regarding whether the observed methylation patterns are in response to the immune infiltration or are somehow its cause, and to some extent it also touches upon a concept of epigenetic immunoediting recently highlighted in a Cell study of glioblastoma¹. As responded to Reviewer 1 (please see text above, including changes made to the manuscript), we cannot draw a supported conclusion based on the data at hand.

As for the comment on correcting mRNA expression for tumor purity, we believe that this is quite a challenging task given the lack of reported methods in the literature that can perform adjustments at gene level. We are aware of few such proposed methods (e.g., by Molania et al.³⁴) and we have not tested/applied them ourselves. It is important to point out here that while DNA methylation, copy number, and mutational data follow a hypothetical linear scale reasonably well (based on for instance the number of alleles), mRNA expression has a much larger dynamic range and is thus likely less linear. Additionally, we do not have explicit tumor microenvironment characterization (for both immune and stromal cells) of the tumor pieces used for WGS, RNA-sequencing, and DNA methylation profiling, lacking the possibility to “confirm/validate/inform” the approach suggested by the reviewer. We do however present single- and bulk cell-line evidence for probable “intrinsic” expression of PDL1/2 with coordinated DNA methylation changes similar to those observed in our primary tumor cohort, an observation that could be compatible with a broader intrinsic immunological phenotype, this is however not something we feel we cannot confidently test or substantiate at this point and would therefore not be comfortable proposing. Taken together, we believe the suggested analysis, however interesting, may become too speculative, and the question at hand is likely better addressed by applying e.g., spatial transcriptomics methods to appropriately classified tumor tissue as we would like to do in the future.

6. Could the authors provide a statistical test, e.g. Chi-squared or hypergeometrical, for the following statements:

- **“Sample cluster agreement between solutions was high, and there was a 57% overlap of...”**

Response:

Please see the sentence below.

- **“...57% overlap of differentially methylated CpGs between solutions (n=1087 CpGs for k=2, n=2845 CpGs for k=3), indicating that the same biology was captured (Supplementary Figure S7a).”**

Response:

We have updated the referred Supplementary Figure to include a Venn diagram showing the CpG overlap between the two cited NMF solutions in Basal tumors. In this new figure, we also show the p-value ($p=0$) for the enrichment test implemented in the Rvenn v1.1.0 package based on the “universe” of all CpGs from which the respective CpG lists were drawn from. Notably, a hypergeometric test produces the same p-value. This sentence has been rephrased to (note the different figure numbering):

Sample cluster agreement between solutions was high (two-sided Chi-square test, $p < 2e-16$), and there was a 57% overlap of differentially methylated CpGs identified between groups using the different solutions (n=1087 CpGs for two-group, n=2845 CpGs for three-group), indicating that the same biology was captured (Supplementary Fig. S5a).

- **“Class agreement was lower between the two different NMF approaches compared to the main NMF split, indicating that subdivision of the Basal branch may be less distinct in a sample size-limited validation cohort (Supplementary Figure S6f).” Please provide a Chi-squared test comparing de novo and directed classification to statistically confirm the validation.**

Response:

We thank the reviewer for spotting this poorly phrased sentence. Based on a suggestion from reviewer 1, we have combined the three different validation sections in the Results into just one. This sentence has thus been rephrased to (note the change in figure numbering):

Interestingly, despite significant agreement between NMF strategies (two-sided Chi-square test, $p = 4e-09$), sample classification with a de novo approach (using instead the 5000 most varying distal-ATAC CpGs) showed discrepancy in class agreement on an individual tumor level (Supplementary Fig. S7c), suggesting that subdivision of the Basal epitype may be less distinct with limited sample size.

- **“Class agreement was high between the two approaches (83%) (Supplementary Figure S6j).”**

Response:

As mentioned above, we have combined all validation sections in the Results. We have added the requested test result, and this sentence has been rephrased to:

Using the approach with distal-ATAC CpGs would bring similar results as class agreement was high between NMF strategies (83%, two-sided Chi-square test, $p = 2e-05$).

7. Related to the statement “Except for perhaps the LAR subtype, our study provides less support for the latter as illustrated by, e.g., stratified expression pattern of FOXA1 and an extensive “one versus rest” (e.g., BL1 vs non-BL1) differential methylation analysis of TNBCtype subtypes in all tumors, Basal tumors, and nonBasal tumors only (Supplementary Figure S9). Our observation is not surprising as mRNA bulk analyses merge the expression patterns of all cells analyzed.”, could the authors use purity-

adjusted RNA expression data to overcome and then compare with DNA methylation?

Response:

The reviewer raises an interesting possibility, which if successful could potentially enhance e.g. CpG-gene correlations as well as provide a new way of analyzing bulk tumor expression data. However, purity adjustment of RNA expression derived from bulk tissue analysis appears to be a highly challenging task. As responded to remark 5 above, we know of few such proposed methods that we have not tested/applied ourselves. Notably, many of the genes we highlight as epigenetically regulated demonstrate a typical “on/off” expression pattern as shown for e.g. *FOXAI*, and *FASN* (see Figure 7 in the revised manuscript). Moreover, acknowledging the inherent issue of non-malignant “contamination” in bulk tumor tissue, we have largely used cell line data to support both epigenetic and mRNA findings in our study. While we agree with the reviewer that it would be interesting to try to adjust bulk RNAseq data, we believe this would require specific information from the analyzed samples and substantial pre-validation work to be able to safely use such corrected data in the current manuscript. In addition to the points raised in our previous answer on this topic with respect to the non-linear dynamic of transcripts per cell for mRNA expression in contrast to e.g. DNA methylation, it is somewhat unknown whether total cellular mRNA-content is constant between the different cell types in our biopsy samples. Accounting for the number of currently unknown factors makes model formulation somewhat challenging and results validation more or less infeasible. Building on the latter we have recently illustrated how commonly used immune cell deconvolution methods compare against actual in situ-stained cell phenotype counts with discouraging results regarding “measuring” specific cell types and not just generic “immune infiltration”³⁵.

Minor comments:

1. The authors state “Importantly, purity-adjusted CpG data showed consistently higher accuracy for this split compared with conventional unadjusted CpG data (Supplementary Figure S4c)”, however, some CPGs (e.g. associated w/ CGI, promoter or HCP) showed higher accuracy when not adjusted. Could the authors comment and adjust their statement?

Response:

Firstly, we would like to point out that there is a substantial overlap between CpGs in the three mentioned contexts (promoter, CGI, and HCP). As example, 87% of CpGs annotated as belonging to the CGI context are also annotated as HCP using the Weber classification, and the majority of CGI CpGs are also annotated as promoter using the genic context reflecting the established link between CpG islands and gene promoters. Therefore, the results the reviewer refers to are interconnected. Consequently, all three CpG contexts show similar beta variance structure (please see the promoter plot in **Rebuttal Figure 2** above).

Secondly, these three CpG contexts appear to be the least affected by the tumor purity adjustment, and only marginal differences can be seen between Supplementary Figure S3b (5000 CpGs) and S3c (note that this is the new number for original Supplementary Figure S4). For all other contexts, we believe the mentioned figure panels clearly demonstrate a higher “stability” when dividing PAM50 Basal and nonBasal tumors using purity-adjusted values compared to unadjusted values, especially for CpGs in ATAC regions (where all accuracies are centered around 0.9 in Supplementary Figure S3b – 5000 CpGs).

A possible explanation for the reviewer’s observation concerning unadjusted data is the impact of highly variant promoter regions in TNBC, like *FOXAI*, which would display more

of an on/off pattern between Basal and nonBasal disease, together with a set of CpGs that would align with the promoter hypermethylation phenotype in nonBasal TNBC as illustrated in our study. As noted previously by the reviewer, the task of separating PAM50 Basal-like from nonBasal tumors could be done with high accuracy by only a handful of CpGs in the promoter region of e.g. *FOXA1*.

While it is apparent that a Basal/nonBasal division in TNBC can be achieved through analyzing different CpG contexts and numbers of CpGs, we believe the data presented above in response to remark 2 illustrates that for further subgroup divisions the same conclusion does not need to be true (e.g., Basal subgroup differences when comparing the bySD and distal-ATAC approaches). As also stated in response to remark 2 above, we believe a more comprehensive methodological investigation of CpG context-dependent variation is needed, with and without purity adjustment, to inform the community in how to best approach unsupervised DNA methylation analyses. This extensive analysis, which should also include tumor cohort composition as a variable, was not within the scope of the current work.

In response to the reviewer's remark, we have rephrased the sentence in the revised manuscript to:

Importantly, purity-adjusted CpG data showed consistently more stable and similar accuracies for this split across tested CpG contexts compared with conventional unadjusted CpG data (Supplementary Fig. S3c).

2. The authors state “PD-L1 was assessed immunohistochemically (...) in formalin-fixed, paraffin-embedded tumor samples in a tissue microarray (TMA) where each sample was represented by two TMA cores (...) The TMA core with the highest score was set as the score for the respective tumor.” Why was this approach chosen over calculating the average of the two cores? Could the authors provide a reference for their chosen approach?

Response:

This approach was chosen after discussions with the pathologist performing the assessments. As the analysis was performed on TMA cores (based on tissue availability), we lacked a comparative complete whole-slide overview. TMA cores had been previously selected to include tumor cells, meaning the two cores could have targeted very different tissue contexts. By selecting one core as representative for a sample, values portrayed in the manuscript's plots represent actual measurements from the tumors instead of an average value that could be inconsistent with any area of the actual tumor. In response to the reviewer's question, we created similar boxplots using the average of both cores, as well as individual values per core (thus a tumor might be represented by 2 cores if assessable) (**Rebuttal Figure 6** below). As seen in **Rebuttal Figure 6**, patterns and conclusions are the same as for the figure in the original manuscript.

Rebuttal Figure 6. PD-L1 TPS score (percentage) versus DNA methylation subgroups. **A)** Individual TMA core level. **B)** On a sample level using the core with highest TPS as representative core. **C)** On a sample level using the mean TPS score for cores.

3. Supplementary Table S1a: Could the authors describe in the methods section how TNBC were defined, i.e. the method of measurement and the threshold applied for ER, PR and Her2. In the table, there is no information regarding the PR and Her2 measurements, only the “ERperc_group” showing that all samples were < 10%. There is no explanation of the method used to measure ER expression. Was it IHC? Response: This information is available in the original study describing the TNBC cohort ⁸ and was therefore not reiterated in this submission. In Sweden, the definition of TNBC is a tumor with $\leq 10\%$ of cells with IHC-staining for ER and PR (thus including tumors with 1–10% stained cells) and an IHC HER2-staining score of < 2 (or for patients with IHC 2+, a non-amplified ISH status). All SCAN-B data for ER, PR, and HER2 status were obtained from clinical routine analyses performed in regional pathology departments.

In response to this remark, we have added this information to the Supplementary Methods document of the revised manuscript for ease of access to readers.

4. The authors state in the methods section “Basic processing of beadchip data was performed as described 25.” We believe the reader would benefit form a more detailed description of the preprocessing of the discovery cohort similar to the description of the

preprocessing of the validation cohort in the Supplemental Methods. Notably, for the validation cohort they used “preprocessingNoob” while ref 25, which was used for the discovery cohort, used “preprocessingFunnorm”. Could the authors clarify this difference?

Response:

The preprocessingNoob() function was used for the validation cohort as it allows data processing of one single sample at a time, an approach that has been implemented in our LIMS (Laboratory Information Management System) holding raw SCAN-B data. Additionally, validation data were generated at one academic service facility (in Uppsala, Sweden) while at another for discovery cohort data (in Lund, Sweden) as outlined in the Methods section. Importantly, both preprocessingNoob() and preprocessingFunnorm() include background correction, but the latter also performs between-array normalization, and we employed a similar Infinium probe normalization for both cohorts using the approach described by Holm et al.²⁵ after the minfi pre-processing step. Lastly, we performed the purity adjustment as outlined in the Methods and Supplementary Methods.

In response to the reviewer’s request, despite it being previously published text, we have added detailed pre-processing steps for the discovery cohort to the Supplementary Methods document.

5. The authors state “This included assigning CpGs to a gene-centric context defined as (...) “proximal” (+/- 5 kbp centered on but excluding TSS) (...)” Did you mean “excluding promoter” instead of “excluding TSS” as TSS is just a single base? Response:

We thank the reviewer for noticing this inconsistency. In the revised manuscript we have corrected this sentence of the Methods to:

This included assigning CpGs to a gene-centric context defined as “promoter” (+/500 bp centered on gene transcription start site, TSS), “proximal” (+/- 5 kbp centered on TSS and excluding the promoter window), or “distal” (>5 kbp from TSS) based on their genomic coordinates (referred to as “genic context”).

6. Related to the following statement “Based on this, we proceeded to characterize the three-group NMF solution which divided the Basal branch into a group with high immune cell infiltration (IM positivity, Basal3)...”, the authors state in the Suppl. Methods part that IM positivity is defined using the score Lehman for IM subtype with a cut-off at 0.17. Can the authors explain why 0.17?

Response:

We agree with the reviewer that 0.17 can be viewed as arbitrary, just as any cutoffs applied to IM-centroid correlations or TNBCtype BL1, BL2, LAR, and M correlations. In another ongoing study from our group (currently under review), we have found 0.17 to be a relevant cutoff for prognostic associations of the IM-classification. However, considering the reviewer’s remark, and that we already use both TIL and an mRNA immune response expression metagene data, we have removed the binary IM-classification altogether from the revised manuscript. We thank the reviewer for highlighting this issue.

7. Could the authors clarify how they obtained the 14464 pairwise correlations in the following statement “Out of 14464 pairwise correlations, 142 were significant in both data sets”?

Response:

The details for the CpG-gene correlation analysis are outlined in the Supplementary Methods document. Briefly, each CpG belonging to a specific set was first mapped to nearby genes

based on transcription start sites located within a 500 kbp genomic window centered on the CpG, creating a one-to-many mapping where individual CpGs are linked to multiple genes within the specified windows (yielding 14464 CpG-gene pairs in total for the CpG set in question). Then, for each pair we computed the correlation between DNA methylation beta values and FPKM gene expression values. We used permutation analysis for determining significance. Analyses in tumors and cell lines were performed similarly. In the revised manuscript, we have further clarified the one-to-many aspect of the CpG-to-gene mapping in the Supplementary Methods document.

8. For the characterization of their epigenetic TNBC subgroups, the authors rely repeatedly on metagene signatures from a publication from 2012 (Fredlund et al.). Could more recently published signatures be considered?

Response:

Signatures published more recently could of course be considered. We have found, however, that Fredlund's 2012 metagenes nicely illustrate the broad transcriptional programs present in breast cancer (shown for instance for PAM50 subtypes in clinical subgroups of breast cancer in a 2023 study by Veerla et al.³³). Moreover, a de novo gene network analysis (as presented in the current manuscript for the nonBasal subgroups) identifies gene networks that are highly correlated with the Fredlund metagenes despite comprising different gene sets. As an example, network1 (identified in the set of 60 nonBasal TNBCs) was enriched for interferon signaling hallmarks. In the set of 6009 unrelated breast cancers, the median FPKM score for this network showed a Spearman correlation of 0.69 with Fredlund's immune response metagene. In the same set of 6009 tumors, Ki67 (*MKI67*) FPKM expression showed a correlation of 0.92 with both metagenes related to mitosis, and *ESR1* FPKM expression showed a Spearman correlation of 0.6 with the steroid response metagene (this gene is not included in this metagene).

In response to the reviewer's question, we computed scores for the 8 predominant TNBC gene co-expression modules recently proposed in 2022 by Saunus et al.³⁶ using the same rank approach as in the manuscript for the Fredlund's metagenes. As seen in **Rebuttal Figure 7** below, we find that each proposed gene expression network is strongly correlated to one or more of the Fredlund metagenes.

A)

Modules	Major functional ontologies ^a	Signalling pathways ^a /intrinsic activators ^b	Size (no. genes)	
Tumour-centric	Blue	Mitotic instability	FOXM1, MYBL2	1239
	Green	Multipotency (SOXE)	Wnt signalling	487
	Brown	Primary cilium	ER, FOXA1	1008
Tumour-stromal	Magenta	ECM-1 (structural)	FBN1, RUNX2	186
	Black	ECM2 (regulatory)	-	207
	Red	Fatty acid metabolism	PPAR γ	274
	Tan	Type-I IFN response	STAT1, IRF9	33
Stromal	Yellow	Adaptive immunity (TILs)	CD40L, CD40, IFN γ , IRF1	712

B)

Spearman: SOXE_signatures_PMD35501337 vs Fredlund

Rebuttal Figure 7. A) Definition of the 8 predominant TNBC gene co-expression modules by Saunus et al. (adapted from publication). B) Spearman correlation plot of rank scores between the Fredlund et al. metagenes and similarly computed rank scores for the 8 co-expression modules proposed by Saunus et al.

Together, this illustrates the well-known redundancy in gene correlations as previously described in breast cancer, e.g., with respect to prognosis³⁷.

9. Could the authors re-work the following statement as almost all nonBasal are HRD-negative “Similarly, the genetic phenotypes of HRD-positive and HRD-negative tumors shaped by defective DNA repair do not match intrinsic epigenetic subtypes as they fall in all subgroups defined in this work, although less frequently in nonBasal tumors.”?
Response:

In the revised manuscript we have rephrased the specific sentence to:

Similarly, while 93% of all HRD-positive tumors (primarily driven by BRCA1/BRCA2-deficiency) were found in the Basal epitype, these did not match a distinct Basal subgroup (two-sided Chi-squared test, $p=0.94$) suggesting that the latter would not be informative for PARP-inhibitor usage.

REFERENCES FOR THIS REBUTTAL

1. Gangoso E, *et al.* Glioblastomas acquire myeloid-affiliated transcriptional programs via epigenetic immunoediting to elicit immune evasion. *Cell* **184**, 2454-2470 e2426 (2021).
2. Stirzaker C, *et al.* Methylome sequencing in triple-negative breast cancer reveals distinct methylation clusters with prognostic value. *Nat Commun* **6**, 5899 (2015).
3. Lin LH, *et al.* DNA Methylation Identifies Epigenetic Subtypes of Triple-Negative Breast Cancers With Distinct Clinicopathologic and Molecular Features. *Mod Pathol* **36**, 100306 (2023).
4. DiNome ML, *et al.* Clinicopathological Features of Triple-Negative Breast Cancer Epigenetic Subtypes. *Ann Surg Oncol* **26**, 3344-3353 (2019).
5. Esteller M, Dawson MA, Kadoch C, Rassool FV, Jones PA, Baylin SB. The Epigenetic Hallmarks of Cancer. *Cancer Discov* **14**, 1783-1809 (2024).
6. Sasiain I, Nacer DF, Aine M, Veerla S, Staaf J. Tumor purity estimated from bulk DNA methylation can be used for adjusting beta values of individual samples to better reflect tumor biology. *NAR Genom Bioinform* **6**, lqae146 (2024).
7. Staaf J, Aine M. Tumor purity adjusted beta values improve biological interpretability of high-dimensional DNA methylation data. *PLoS One* **17**, e0265557 (2022).
8. Staaf J, *et al.* Whole-genome sequencing of triple-negative breast cancers in a population-based clinical study. *Nat Med* **25**, 1526-1533 (2019).
9. Staaf J, *et al.* RNA sequencing-based single sample predictors of molecular subtype and risk of recurrence for clinical assessment of early-stage breast cancer. *NPJ Breast Cancer* **8**, 94 (2022).
10. Ryden L, *et al.* Minimizing inequality in access to precision medicine in breast cancer by real-time population-based molecular analysis in the SCAN-B initiative. *Br J Surg* **105**, e158-e168 (2018).
11. Loi S, *et al.* The journey of tumor-infiltrating lymphocytes as a biomarker in breast cancer: clinical utility in an era of checkpoint inhibition. *Ann Oncol* **32**, 1236-1244 (2021).
12. Loibl S, *et al.* Early breast cancer: ESMO Clinical Practice Guideline for diagnosis, treatment and follow-up(dagger). *Ann Oncol*, (2023).
13. de Jong VMT, *et al.* Prognostic Value of Stromal Tumor-Infiltrating Lymphocytes in Young, Node-Negative, Triple-Negative Breast Cancer Patients Who Did Not Receive (neo)Adjuvant Systemic Therapy. *J Clin Oncol* **40**, 2361-2374 (2022).

14. Traina TA, *et al.* Enzalutamide for the Treatment of Androgen Receptor-Expressing Triple-Negative Breast Cancer. *J Clin Oncol* **36**, 884-890 (2018).
15. Lehmann BD, *et al.* TBCRC 032 IB/II Multicenter Study: Molecular Insights to AR Antagonist and PI3K Inhibitor Efficacy in Patients with AR(+) Metastatic Triple-Negative Breast Cancer. *Clin Cancer Res* **26**, 2111-2123 (2020).
16. Xu L, Xu P, Wang J, Ji H, Zhang L, Tang Z. Advancements in clinical research and emerging therapies for triple-negative breast cancer treatment. *Eur J Pharmacol* **988**, 177202 (2024).
17. Vanauberg D, Schulz C, Lefebvre T. Involvement of the pro-oncogenic enzyme fatty acid synthase in the hallmarks of cancer: a promising target in anti-cancer therapies. *Oncogenesis* **12**, 16 (2023).
18. Liu J, *et al.* An Integrated TCGA Pan-Cancer Clinical Data Resource to Drive High-Quality Survival Outcome Analytics. *Cell* **173**, 400-416 e411 (2018).
19. Brown LJ, Achinger-Kawecka J, Portman N, Clark S, Stirzaker C, Lim E. Epigenetic Therapies and Biomarkers in Breast Cancer. *Cancers (Basel)* **14**, (2022).
20. Corces MR, *et al.* Lineage-specific and single-cell chromatin accessibility charts human hematopoiesis and leukemia evolution. *Nat Genet* **48**, 1193-1203 (2016).
21. Aran D, Sabato S, Hellman A. DNA methylation of distal regulatory sites characterizes dysregulation of cancer genes. *Genome Biol* **14**, R21 (2013).
22. Spitz F, Furlong EE. Transcription factors: from enhancer binding to developmental control. *Nat Rev Genet* **13**, 613-626 (2012).
23. Bulger M, Groudine M. Functional and mechanistic diversity of distal transcription enhancers. *Cell* **144**, 327-339 (2011).
24. Shlyueva D, Stampfel G, Stark A. Transcriptional enhancers: from properties to genome-wide predictions. *Nat Rev Genet* **15**, 272-286 (2014).
25. Holm K, *et al.* An integrated genomics analysis of epigenetic subtypes in human breast tumors links DNA methylation patterns to chromatin states in normal mammary cells. *Breast Cancer Res* **18**, 27 (2016).
26. Lehmann BD, *et al.* Refinement of Triple-Negative Breast Cancer Molecular Subtypes: Implications for Neoadjuvant Chemotherapy Selection. *PLoS One* **11**, e0157368 (2016).
27. Huang H, *et al.* Defining super-enhancer landscape in triple-negative breast cancer by multiomic profiling. *Nat Commun* **12**, 2242 (2021).
28. Jovanovic B, *et al.* Heterogeneity and transcriptional drivers of triple-negative breast cancer. *Cell Rep* **42**, 113564 (2023).

29. Calhoun BC, Collins LC. Predictive markers in breast cancer: An update on ER and HER2 testing and reporting. *Semin Diagn Pathol* **32**, 362-369 (2015).
30. Perou CM, *et al.* Molecular portraits of human breast tumours. *Nature* **406**, 747-752 (2000).
31. Veerla S, Staaf J. Kataegis in clinical and molecular subgroups of primary breast cancer. *NPJ Breast Cancer* **10**, 32 (2024).
32. Paquet ER, Hallett MT. Absolute assignment of breast cancer intrinsic molecular subtype. *J Natl Cancer Inst* **107**, 357 (2015).
33. Veerla S, Hohmann L, Nacer DF, Vallon-Christersson J, Staaf J. Perturbation and stability of PAM50 subtyping in population-based primary invasive breast cancer. *NPJ Breast Cancer* **9**, 83 (2023).
34. Molania R, *et al.* Removing unwanted variation from large-scale RNA sequencing data with PRPS. *Nat Biotechnol* **41**, 82-95 (2023).
35. Roostee S, *et al.* Tumour immune characterisation of primary triple-negative breast cancer using automated image quantification of immunohistochemistry-stained immune cells. *Sci Rep* **14**, 21417 (2024).
36. Saunus JM, *et al.* Epigenome erosion and SOX10 drive neural crest phenotypic mimicry in triple-negative breast cancer. *NPJ Breast Cancer* **8**, 57 (2022).
37. Ein-Dor L, Kela I, Getz G, Givol D, Domany E. Outcome signature genes in breast cancer: is there a unique set? *Bioinformatics* **21**, 171-178 (2005).

Detailed point-by-point response to the reviewers' comments on the manuscript NCOMMS-24-73463A

First, we would like to thank all reviewers for their work, constructive criticism and suggestions on how to improve and further clarify this manuscript. We are grateful for all feedback and comments, which have helped us to improve our manuscript.

Please find below the detailed point-by-point responses to the reviewers' final comments. Reviewers' comments are presented **in bold** and our responses are shown in regular text. Textual alterations made to the revised version of the manuscript are shown in *italics*, with underlined text showing modifications in existing text when deemed appropriate. The text referenced from other studies used only in this rebuttal is shown in a different font type if present. All textual alterations are present in a marked-up version of the manuscript according to journal instructions. Please note that the actual reference number for a specific literature reference may differ between the reference list in this document and the revised manuscript version that has been resubmitted. Any page numbers refer to the revised manuscript (clean file).

Reviewer #1 (Remarks to the Author):

The authors have addressed my comments.

Response: We thank reviewer 1 for all constructive comments and are happy to see that the reviewer approves publication.

Reviewer #2 (Remarks to the Author):

The authors have thoroughly addressed all comments raised by the Reviewers and have revised the manuscript accordingly. They have enhanced relevant sections with additional details, provided further clarifications, incorporated appropriate citations, and included several new figures to strengthen the manuscript. The manuscript is acceptable for publication as an original research article in Nature Communications after implementing the following corrections.

References should be cited directly following a statement without using "e.g.," or "like" before them. For example, citations such as "e.g., 6, 7, 15, 16" and "like 18, 19, 20" should be formatted as "6, 7, 15, 16" and "18, 19, 20", respectively.

Furthermore, formatting adjustments are needed for statistical reporting. For example, in the sentence: "Sample cluster agreement between 350 solutions was high (two-sided Chi-square test, $p < 2e-16$), and there was a 57% overlap of differentially methylated CpGs identified between groups using the different solutions (n=1087 CpGs for two-group, n=2845 CpGs for three-group), indicating that the same biology was captured (Supplementary Fig. S5a)." The following changes should be made: insert a space between "p" and "<" and between "<" and "2e-16". Additionally, p should be italicized. Insert a space between "n" and "=" and between "=" and the numbers, so "n=1087" should be formatted as "n = 1087". These formatting adjustments should be consistently applied throughout the manuscript.

Response: We thank reviewer 2 for all constructive comments and explicit detailed suggestions. In the editorial revision we have changed the text in the main manuscript

accordingly. We are happy to see that the reviewer approves publication.

Reviewer #3 (Remarks to the Author):

Response: We thank reviewer 3 for the work performed.

Reviewer #4 (Remarks to the Author):

The revision of “The DNA methylation landscape of primary triple-negative breast cancer”, by Aine et al., led to improvements of the manuscript. We appreciate the efforts made by Aine and colleagues to reply to all of our comments and to follow most of our suggestions. Overall, the revised manuscript appears more clear, we have just one remaining suggestion:

To address our major comment #2, Aine et al. generated crucial data and provided important explanations. While we agree that including all of this would make the manuscript too long and difficult to read, we encourage them to incorporate some of the data and key explanations. This would be beneficial for future readers who may have similar questions.

Response: We thank reviewer 4 for all constructive comments and are happy to see that the implemented changes have improved the reviewer(s) standpoint on the manuscript.

While we do agree with the reviewer about the relevance of reporting these aspects to the community we do still believe it would unfortunately complicate the main manuscript (which is extensive and challenging at present). We do believe that a more thorough methodological paper, grounded in biology, would be able to tackle the question more comprehensively and represent a more appropriate solution in the end for the research field. E.g. in our continued ongoing work we see that we need to move beyond selecting “arbitrary” numbers of CpGs when we are not using specific CpG contexts (like top 5000 bySD shown in the previous rebuttal), as we are likely detecting stochastic DNA methylation processes that have different impact in the variation distribution compared to biological themes. Moreover, we need to more comprehensively also address the question of cohort composition (e.g. number of basal/non-basal tumors), which is of relevance for all unsupervised analyses targeting variance across a cohort of samples. Finally, we are now also moving these analyses to ER-positive disease which may have a very different variance structure due to underlying tumor biology. All in all, these analyses would be more than sufficient for a methodological study of relevance to the field.

Manuscript Title: The DNA methylation landscape of primary triple-negative breast cancer

Comments

The authors have thoroughly addressed all comments raised by the Reviewers and have revised the manuscript accordingly. They have enhanced relevant sections with additional details, provided further clarifications, incorporated appropriate citations, and included several new figures to strengthen the manuscript. The manuscript is acceptable for publication as an original research article in *Nature Communications* after implementing the following corrections.

References should be cited directly following a statement without using "e.g.," or "like" before them. For example, citations such as "e.g.,^{6, 7, 15, 16}" and "like^{18, 19, 20}" should be formatted as "^{6, 7, 15, 16}" and "^{18, 19, 20}", respectively.

Furthermore, formatting adjustments are needed for statistical reporting. For example, in the sentence: "**Sample cluster agreement between 350 solutions was high (two-sided Chi-square test, $p < 2e-16$), and there was a 57% overlap of differentially methylated CpGs identified between groups using the different solutions (n=1087 CpGs for two-group, n=2845 CpGs for three-group), indicating that the same biology was captured (Supplementary Fig. S5a).**" The following changes should be made: insert a space between "p" and "<" and between "<" and "2e-16". Additionally, *p* should be italicized. Insert a space between "n" and "=" and between "=" and the numbers, so "n=1087" should be formatted as "n = 1087". These formatting adjustments should be consistently applied throughout the manuscript.